# Large-eddy simulation of the ice shelf-ocean boundary layer near ice front of Nansen ice shelf, Antarctica

Ji Sung Na[1], Taekyun Kim[2], Emilia Kyung Jin[1*], Seung-Tae Yoon[3], Won Sang Lee[1], Sukyoung Yun[1], Jiyeon Lee[1]

[1]Korea Polar Research Institute, Incheon, 21990, South Korea
[2]Jeju National University, Jeju, 63243, South Korea
[3]Kyungpook National University, Daegu, 41566, South Korea

*Correspondence to*: Emilia Kyung Jin (jin@kopri.re.kr)

**Abstract.** Ice melting beneath the Antarctic ice shelf is caused by heat transfer through the ice shelf–ocean boundary layer (IOBL); however, our understanding of the fluid dynamics and thermohaline physics of the IOBL flow is poor. In this study, we utilize a large-eddy simulation (LES) model to investigate the role of turbulence within the IOBL flow with a sub-ice shelf plume and the ocean dynamics near the ice front. To simulate the varying turbulence intensities, we imposed theoretical profiles of the velocity by varying the power-law index, because turbulence intensity determines the shape of the velocity profile. To reproduce the oceanic environment near the Nansen ice shelf, boundary conditions for the melting at the ice shelf base and freezing at the sea surface were derived based on *in situ* observations. The main forcings for determining the ocean dynamics near the ice front are: the upwelling of the positively-buoyant meltwater generated by the basal melting, downwelling by the concentrated salinity at the sea surface, and shear force of the sub-ice shelf plume. In terms of overturning features near the ice front, we validated the LES simulation results by comparing them with the *in situ* observational data. In the comparison of the velocity profiles, the LES-derived strength of the overturning cells is similar to that obtained from the observational data. Moreover, the vertical distribution of the simulated temperature and salinity, which were mainly determined by the positively buoyant meltwater and sea-ice formation, was also comparable to that of the observations. We conclude that the IOBL flow near the ice front and its contribution to the ocean dynamics can be realistically resolved using our proposed method. In the strong turbulence case, distinct features such as a higher basal melting rate ($0.153$ m yr$^{-1}$), weak upwelling of the positively-buoyant ice shelf water, and a higher sea-ice formation were observed, suggesting a relatively high-speed current within the IOBL because of a large momentum transfer. Because we only considered the sub-ice shelf plume as a parameter for basal melting in this study, we estimated that 12–25 % of the total basal melting rate near the Nansen ice shelf front is due to the sub-ice shelf plume. The findings of this study will contribute toward the understanding the complex IOBL-flow physics and its impact on the ocean dynamics near the ice front.

## 1 Introduction

The Antarctic ice sheet (AIS) is buttressed by floating extensions of land ice and ice shelves (Rignot et al., 2013), which primarily control the AIS mass balance by hindering the flow of inland ice into the ocean to prevent sea level rise (Holland et al., 2020). Internal glaciological stresses, iceberg calving, and surface melting at the sub-ice shelf are the key factors that weaken the ice shelf stability (Liu et al., 2015; Smith et al., 2020). However, directly accessing the ocean-filled cavities beneath the ice shelves from the open ocean via autonomous underwater vehicles (e.g., Jenkins et al., 2010) or through the ice shelf by hot-water drilling (e.g., Stanton et al., 2013) requires challenging, intermediate to large-scale, logistical support depending on the size and thickness of the ice shelves.

The sub-ice shelf oceanic environment can be divided into two classes, namely the "cold-water cavity" (e.g., the Larsen C, Ross, Filchner–Ronne, and Amery ice shelves) and the "warm-water cavity" (e.g., the Getz and Totten ice shelves), depending on the amount of basal melting as well as ocean conditions (Gwyther et al., 2016; Joughin et al., 2012). In the cold-water cavity, shear forces generated by the tides and brine rejection during the sea ice formation (e.g., high salinity shelf water (HSSW)) are the driving forces that cause basal melting (Davis and Nicholls, 2019; Yoon et al., 2020). In contrast, the intrusion of circumpolar deep water and melt-driven circulation near the grounding line mainly cause basal melting in the warm-water cavities (Holland et al., 2020; Jacobs et al., 1992). Investigating the ice shelf–ocean boundary layer (IOBL), which is the boundary layer (meters to tens of meters) right beneath the ice shelf, is a complex problem because the shear forces from the ocean and the stabilizing force of the meltwater both influence the IOBL characteristics (Begeman et al., 2018; Garabato et al., 2017). Therefore, a turbulence resolving model is required to analyze the IOBL flow and its structure and reveal strategic information on the basal melting phenomenon occurring below the ice shelves (Jenkins, 2016; Holland et al., 2020).

Various observational studies on the main ice shelves of Antarctica have been performed to observe the thermohaline characteristics and structures of the IOBLs as well as identify the ocean conditions beneath the ice shelves (Jenkins, 2010; Kimura et al., 2015). In the Larsen C ice shelf which is a cold-water cavity, a well-mixed boundary layer (20–30 m) was observed in both temperature and salinity, induced by a strong tidal forcing and a weak stratification. A moderate melt rate was observed, despite the low thermal driving due to the observed shear-driven turbulence (Davis and Nicholls, 2019). Similar melt rates for weak stratification were also observed beneath the Fimbul and Ross ice shelves (Arzeno et al. 2014; Hattermann et al., 2012). Moreover, the heat entrainment is prevented near the grounding line of the Ross ice shelf because of the strong density gradient between the meltwater and the denser continental shelf water in this region, yielding a low melting. (Begeman et al., 2018).

Positively buoyant sub-ice shelf plumes created near the grounding line can affect the stratification and heat entrainments within the IOBL; they produce shear forces on the IOBL because of the buoyant moving of meltwater near the grounding line and ice shelf front (Hewitt, 2020; Holland and Jenkins, 1999). Since the sub-ice shelf plume modulates the IOBL characteristics with a strong shear mixing, the basal melt rate can be increased by weakening the stratification and enhancing the entrainment

of the seawater from the outer region. At the ice-shelf bottom near the ice front, an increase in turbulent mixing can increase the IOBL thickness and cause variations in the heat exchange and entrainment, thereby modifying the melt rate (Jenkins, 2011).

Despite the various observational studies, the thermohaline physics as well as the turbulent environment for the melting and refreezing processes within the IOBL remain unclear, primarily because of the technical difficulties in accessing and observing the sub-ice shelf ocean cavity, particularly the IOBL, and the complexity of the flow itself, which has strong turbulent and buoyant characteristics with various spatial and temporal scales (Nicholls et al., 2016; Everett et al., 2018). To observe the effects of various forcings separately (e.g. shear, stabilizing force of the meltwater, and buoyant moving of meltwater), an independent numerical approach for these forcings is needed. Therefore, various high-resolution turbulence

models such as the large-eddy simulation (LES) or direct numerical simulation (Gayen et al., 2016; McConnochie and Kerr, 2018; Mondal et al., 2019; Vreugdenhil and Taylor, 2019) have been introduced in recent years.

The LES has been successfully applied to investigate the role of turbulence in the ice shelf–ocean interactions within the IOBL in Greenland (Carsey and Garwood, 1993; Denbo and Skyllingstad, 1996) and the Arctic Sea (Glendening, 1995; Skyllingstad and Denbo, 2001; Matsumura and Ohshima, 2015; Ramudu et al. 2018; Li et al. 2018). However, studies on the

application of LES to the IOBL under a sub-ice shelf environment are limited (Dinniman et al., 2016). This is because the geometry and scales of this ice–ocean interactions are qualitatively different for an ice shelf, particularly at the ice front.

Observational efforts of meltwater behavior and ocean circulation near the frontal region of the ice shelf demonstrate various mechanisms at different locations around Antarctica. In the frontal region of the Pine island ice shelf, Garabato et al. (2017) revealed that the ascent of the meltwater outflow causes vigorous lateral export, affecting the settling of meltwater at

depth. The intrusion of relatively warm surface waters and high basal melting near the ice shelf front was observed in the Ross ice shelf (Hogan et al., 2011). Moreover, Malyarenko et al. (2018) suggested the existence of a "wedge" of fresher water in the Western Ross sea and it is formed from meltwater near the ice shelf front.

In this study, we performed LES experiments for the IOBL and oceanic flow with neutrally buoyant sub-ice shelf plumes near the ice front. To include thermohaline effect by the sea-ice formation at sea surface and basal melting at the ice-shelf base,

surface fluxes in both temperature and salinity were used. The boundary conditions were derived based on the *in situ* observation data—namely the conductivity–temperature–depth (CTD), lowered acoustic Doppler current profiler (LACDP) data, and automatic weather station (AWS) data—collected in front of the Nansen Ice Shelf (NIS), which is cold-water cavity, during the Antarctic expedition led by the Korea Polar Research Institute in January and February 2017. One of the main objectives of this study was to investigate the neutrally buoyant sub-ice shelf plume's impact on basal melting. Therefore, the

target parameter in the LES experiments was turbulence intensity within the IOBL because the sub-ice shelf plume and its heat entrainment are related to basal melting via turbulent shear. Another objective was to validate our proposed methodology (domain configuration and boundary conditions) and the oceanographic properties simulated using the LES. Herein, quantitatively validate the simulated oceanographic properties with a discussion on the ocean circulation. Using the validated three-dimensional LES outputs, we elucidate the distribution of melting at the ice-shelf bottom as well as the associated factors

such as turbulent characteristics and flux changes within the IOBL.

Section 2 of this paper presents the governing equations (i.e., the Navier–Stokes equation and liquidus condition) for the oceanic flow with melting and freezing effects as well as a detailed explanation of the simulations. Section 3 presents the analysis of the LES simulation results to determine the IOBL characteristics—flow velocity, potential temperature, salinity, fluxes, and turbulence statistics. The major findings, future works, and implications of this study are summarized in Sections 4 and 5.

## 2 Methodology

### 2.1 CTD, LADCP observations

During the hydrographic survey by the icebreaking research vessel ARAON operated by the Korea Polar Research Institute, full-depth CTD and LADCP profiles were obtained in one-hour intervals from 14 February 2017 13:00 UTC to 15 February 2017 11:55 UTC at a single site in the Terra Nova Bay in front of the NIS. This survey was conducted to examine the vertical structures of the sub-ice shelf plume with temporal variations. The exact location of the observations was 75.008 °S, 163.617 °E (Fig. 1b), located approximately 1 km away from the ice front. Because the LADCP data was incorrectly observed at the final casting, 24 CTD profiles and 23 LADCP profiles were used in this study. The CTD data were processed following the SBE recommended procedure (Sea-Bird Electronics, Inc., Bellevue, Washington, USA; 2014), and the LADCP data were processed using the methods introduced in Thurnherr (2004). This location for the observations is the coastal polynya region where strong katabatic winds cause extreme heat loss in the ocean, leading to the sea-ice formation. Based on the wind speeds and air temperatures in the AWS data acquired on ARAON, we calculated the sensible heat as well as the amount of sea-ice formation (Thompson et al., 2020). Atmospheric properties such as wind speed, temperature, sensible heat and sea-ice formation were obtained using the AWS instrument on ARAON and are listed in Table 1. The detailed shipboard information and processing methods for the hydrographic data are described in Yoon et al. (2020).

### 2.2 Numerical model

To simulate the oceanic flow incorporating the effects of sea-ice formation or basal melting, the parallelized large-eddy simulation model (PALM, version 6–r4536) developed by Leibniz University was employed (Noh et al., 2009; Raasch et al., 2001). This model solves the non-hydrostatic, Boussinesq-approximated, filtered Navier–Stokes equations with buoyancy force, Coriolis force, and subgrid-scale (SGS) turbulent closure. The Boussinesq approximation can be applied to flows with negligible density variation. Furthermore, in the time integration, the time-difference formulas were computed using the third-order Runge–Kutta method. The 5[th] order upwind scheme was used to solve the flow advection (Wicker and Skamarock, 2002). The pressure was modeled using a Poisson equation, while the mass, momentum, potential temperature, and salinity conservations were governed by Eqs. (1)–(4), respectively.

$$\frac{\partial \overline{u_k}}{\partial x_k} = 0 , \tag{1}$$

$$\frac{\partial \overline{u_\iota}}{\partial t} + \frac{\partial \overline{u_\iota u_k}}{\partial x_k} = -\frac{1}{\rho}\frac{\partial \overline{\pi^*}}{\partial x_k} - \varepsilon_{ijk}f_j\overline{u_k} + \varepsilon_{i3k}f_3\overline{u_{g,k}} + g\frac{(\rho_\theta - <\rho_\theta>)}{<\rho_\theta>}\delta_{i3} - \frac{\partial \tau^r_{ki}}{\partial x_k} , \tag{2}$$

$$\tau^r_{ki} = \tau_{ki} - \frac{1}{3}\tau_{jj}\delta_{ki}, \overline{\pi^*} = \overline{p^*} + \frac{1}{3}\tau_{jj}\delta_{ki},$$

$$\frac{\partial \overline{\theta}}{\partial t} = -\frac{\partial \overline{u_k}\overline{\theta}}{\partial x_k} + \frac{\partial H_k}{\partial x_k} + Q_\theta , \tag{3}$$

$$\frac{\partial \overline{S}}{\partial t} = -\frac{\partial \overline{u_k}\overline{S}}{\partial x_k} + \frac{\partial S_k}{\partial x_k} + Q_s , \tag{4}$$

where $u_k$ is the flow velocity, $\rho$ is the seawater density, $\pi^*$ is the dynamic pressure, $\varepsilon_{ijk}$ is the Levi–Civita symbol, $f$ is the Coriolis force for 75S ($-1.41\times10^{-4}\,\mathrm{s}^{-1}$), $\delta_{ij}$ is the Kronecker delta function, $g$ is the gravitational acceleration, $\rho_\theta$ is the potential density, $T$ is the absolute temperature, $v$ is the dynamic viscosity, $\tau^r_{kj}$ is the Reynolds stress, $\theta$ is the potential temperature, $p$ is the hydrostatic water pressure, $p_0$ is the reference pressure, and $S$ is the salinity (Jackett et al., 2006). Additionally, $Q_\theta$ and $Q_s$ are the external forcing of the source/sink terms—$T$ and $S$, respectively. The overbars indicate that the values have been filtered over the grid volume. Combining these equations, the SGS turbulent kinetic energy ($e$) equation can be derived as follows:

$$\frac{\partial \bar{e}}{\partial t} = -\overline{u_k}\frac{\partial \bar{e}}{\partial x_k} - \tau_{ki}\frac{\partial \overline{u_\iota}}{\partial x_k} + \frac{g}{\theta_0}\overline{u'_3\rho'_\theta} - \frac{\partial}{\partial x_k}\overline{\{u'_k(e + \frac{p'}{\rho_0})\}} - \varepsilon \tag{5}$$

where $= \frac{\overline{u'_i u'_i}}{2}$ , and $\varepsilon$ is the SGS dissipation rate.

The SGS stresses ($\tau_{ki}$, $H_k$, and $S_k$) for the momentum, potential temperature, and salinity are parameterized as follows: (Maronga et al., 2015)

$$\tau_{ki} = \overline{u_k u_\iota} - \overline{u_k}\overline{u_\iota} = -K_m\left(\frac{\partial \overline{u_\iota}}{\partial x_k} + \frac{\partial \overline{u_k}}{\partial x_i}\right) + \frac{2}{3}\delta_{ik}\bar{e}, \tag{6}$$

$$H_k = \overline{u_k\theta} - \overline{u_k}\bar{\theta} = \overline{u'_k\theta} = -K_h\left(\frac{\partial \bar{\theta}}{\partial x_k}\right), \quad K_h = \left(1 + 2\frac{l}{\Delta}\right)K_m , \tag{7}$$

$$S_k = \overline{u_k S} - \overline{u_k}\bar{S} = \overline{u'_k S'} = -K_h\left(\frac{\partial \bar{S}}{\partial x_k}\right), \tag{8}$$

$$\frac{\partial}{\partial x_k}\left[\overline{u'_k(e + \frac{p'}{\rho_0})}\right] = -\frac{\partial}{\partial x_k}v_e\frac{\partial \bar{e}}{\partial x_k}, v_e = 2K_m , \tag{9}$$

$$\varepsilon = C_\varepsilon\frac{e^{3/2}}{l} , \quad C_\varepsilon = 0.19 + 0.74l,$$

$$K_m = C_m l\sqrt{\bar{e}} \quad \text{with empirical value } C_m = \text{constant} = 0.1, \text{ and } l = \min(1.8z, \Delta, 0.76\sqrt{\bar{e}}[\frac{g}{\rho_{\theta,0}}\frac{\partial \overline{\rho_\theta}}{\partial z}]^{-\frac{1}{2}})$$

where $K_m$ and $K_h$ are the eddy diffusivities for momentum and heat; $l$ is the turbulent mixing length which depends on height $z$ (distance from the wall), grid spacing, and stratification; $\Delta = (\Delta x\Delta y\Delta z)^{(1/3)}$ is the length scale of the filter; and $\rho_\theta$ is the potential density. (the variables with prime are SGS variables). Because this SGS model and the coefficients are designed for a stably stratified boundary layer flow, it is suitable for resolving a stratified ocean flow with melting or freezing effect.

However, the SGS fluxes can be incorrectly determined at a region where the flow becomes laminar, as this model assumes only a turbulent flow.

It is necessary to determine the ambient variables ($\theta_f$ and $S_f$) within the IOBL and the interfacial variables ($\theta_b$ and $S_b$) near the ice shelf–ocean boundary to resolve the thermal and salinity changes caused by the freezing effect at the sea surface or the basal melting effect at the ice shelf–ocean boundary. Herein, we determined the ambient variables based on *in situ* CTD observations. To obtain the interfacial variables, we solved the conservation equations of heat and salt, along with the liquidus condition and turbulent transfer coefficients for heat and salt (Beckmann and Goosse, 2003; Vreugdenhil and Taylor, 2019).

$$c_w \rho_w \Gamma_\theta u_* (\theta_f - \theta_b) = \rho_i L_i m \,, \tag{10}$$

$$\rho_w \Gamma_S u_* (S_f - S_b) = \rho_i S_b m \,, \tag{11}$$

$$\theta_b = \lambda_1 S_b + \lambda_2 + \lambda_3 P \,, \tag{12}$$

$$S_b = \frac{-\left(\lambda_2 + \lambda_3 P - \theta_f + \frac{L_i \Gamma_S}{c_w \Gamma_\theta}\right) + \sqrt{\left(\lambda_2 + \lambda_3 P - \theta_f + \frac{L_i \Gamma_S}{c_w \Gamma_\theta}\right)^2 - 4\lambda_1 \left(\frac{L_i \Gamma_S}{c_w \Gamma_\theta}\right) S_f}}{-2\lambda_1} \,, \tag{13}$$

where $m$ is the melting rate at the ice-shelf base or the freezing rate at the sea surface; the subscripts $w$ and $i$ refer to the parameters for water and ice, respectively. The parameters values are listed in Table 1. The friction velocity at the ice-shelf base was calculated from the simulated velocity field. We used 0.026 m s$^{-1}$ as the friction velocity for the calculation of thermal and salinity fluxes induced by the sea-ice formation, although the effect of wind stress on the momentum was excluded to focus on the relationship between the sub-ice shelf plume and the development of overturning cells.

The fluxes for temperature and salinity, $q_{\theta*}$ and $q_{S*}$ at the ice-shelf bottom were formulated by Monin–Obukhov similarity and interfacial values, $\theta_b$ and $S_b$ obtained by resulting equation, Eq (13) (McPhee et al., 1987; Ramudu et al. 2018) below:

$$q_{\theta*} = \Gamma_\theta [\theta(z_1) - \theta_b] u_* \,, \tag{14}$$

$$q_{S*} = \Gamma_S [S(z_1) - S_b] u_* \,, \tag{15}$$

where $u_*$ is the friction velocity which is calculated using logarithmic law of the wall with the velocity at the first node and roughness length; $\Gamma_\theta$ and $\Gamma_S$ are the non-dimensional transfer coefficients of heat and salt, respectively, determined from the near-wall physics. Based on a high-resolution LES study on the heat and salt transfer coefficients, which are described as functions of the friction velocity and thermal driving, these coefficients, $\Gamma_\theta$ and $\Gamma_S$ at the ice-shelf base were found to be $8 \times 10^{-3}$ and $2.6 \times 10^{-4}$ for basal melting, respectively (Vreugdenhil and Taylor, 2019). For the sea-ice formation at the sea surface, the same coefficients used in a previous study on sea ice formation in polynyas were used, because the thermal driving in our study was comparable that in the previous study (Heorton et al., 2017). These melting and freezing effects were applied at the first grid from the ice-shelf base or the sea surface. In this study, the melting or freezing effects at the lateral side of the ice front were not included.

## 2.3 Simulation description

In this study, we conducted four LES simulations of the turbulence intensity within the IOBL to reveal the influence of turbulence on IOBL characteristics. Different turbulence intensities were described in four different vertical profiles of the zonal velocity. Based on the power-law assumption of turbulent boundary layer flow ($U=U_t\times(z/z_0)^{(1/n)}$), different velocity profiles (inset profiles in Figure 2) were composed using different power-law indices, n = 3 (weak turbulence), 4, 5 and 7 (strong turbulence) to simulate the turbulence intensity within the IOBL (Irwin, 1979; Kikumoto et al., 2017) (height $z$ represents the distance from the ice shelf). The freestream (geostrophic) velocity $U_t$ was set as 0.06 m s$^{-1}$ at 572 m depth, based on the *in situ* observations near the ice front. To explore the effects of sub-ice shelf plume on ocean dynamics without wind stress, initial velocity was set to zero at the open ocean region above the ice shelf base at the initial state and zero surface stress at the sea surface were used as the boundary condition, whereas the velocity profiles were used below the ice shelf base (280 m) for the sub-ice shelf plume. These velocity profiles for the four different cases were used for the initialization of the flow field and inlet boundary condition. The simulation dimensions were 3,456 m × 3,456 m × 864 m in the $x$, $y$, and $z$ directions, respectively. For the simulations, a grid of 288 × 288 × 144 cells was used with a 12m horizontal grid and a 6 m vertical grid, and a surface roughness ($z_0$) of 0.005 m (Gwyther et al., 2016). The grid size and surface roughness were adopted based on a numerical study on high shear plume flow (Gao et al., 2019). In addition, we performed a grid sensitivity study to the optimal grid size for higher accuracy and less computational costs (Figure S1 in Supplementary), and found that a moderate grid resolution of 288 × 288 × 144 was suitable for resolving the turbulence with the parameterization of melting and freezing effects. Using the masking method, an idealized ice shelf with a depth of 280 m was described at the upper-left part of the simulation domain to indicate the ice shelf depth near the front of the NIS (Briscolini and Santangelo, 1989; Stevens et al., 2017). For zonal velocity, potential temperature, and salinity, the inlet boundary condition was set to Dirichlet boundary conditions with constant vertical profiles that were the same as the initial profiles. At the start of the simulation, the initial profiles were imposed on the whole domain. The vertical profiles of the initial and inlet boundary conditions for potential temperature and salinity were determined from the 24 CTD observations (Figure 5). The initial potential temperature above 280 m depth and at a depth from 280 m to 570 m were set equal to the temperature at the sea surface (–1.9 °C) and to the average temperature (–2.06 °C), respectively. The averaged salinity values (obtained from CTD observation) were adoped as the initial profile of salinity. The outlet boundary condition was determined to match the radiation boundary condition (extrapolation), which prevented numerical errors without rapid changes in the velocity and scalar properties. Moreover, the radiation boundary condition at the outlet boundary allowed wave-like motions within the domain to pass through the boundary with only a small reflection. The cyclic boundary condition was applied to lateral boundaries, whereas the Neumann boundary condition for momentum was imposed on the top layer. In other words, the wind affects the scalar (temperature and salinity) fluxes but not the momentum fluxes at the sea surface. A random generator for small velocity perturbation was applied at depths from 30 to 800 m to quickly spin up the turbulence. The total simulation time required to reach the quasi-steady state was 96 hours. Details on the simulation domain and the boundary conditions are presented in Fig. 1, which shows the target

study region, observation points, and simulation domain with a schematic diagram for the oceanic flow alongside a sub-ice shelf plume. In the model simulation, we configured two physical domains, i.e., the sub-ice shelf and open ocean regions. In the open ocean, the simulated ocean velocity, potential temperature, and salinity results were validated using the CTD and LADCP observational data. Subsequently, the flow characteristics of the IOBL flow beneath the ice shelf were investigated in detail.

## 3 Results

### 3.1 Quasi-steady ocean environment near the ice front

To confirm that the IOBL and oceanic flows approach a quasi-steady state, we plotted the time series of the the friction velocities at the ice-shelf bottom (Figure 2). In this study, we determined the IOBL region wherein the heat flux was 5 % of the maximum heat flux near the ice shelf base as IOBL is analogous to the atmospheric boundary layer (Derbyshire, 1990). This is, the IOBL region was defined as the region where the thermodynamic changes induced by the basal melting at the ice shelf base are dominant. The detailed analysis of the IOBL bottom is discussed in a later section on the vertical heat flux profile. The total simulation time (96 h) was normalized by the large-eddy turnover time ($t^*$), which was calculated as the scale of overturning large eddy within the IOBL (IOBL depth) divided by the friction velocity. The friction velocities in the four LES cases showed significant fluctuations during the whole period, indicating that large-scale eddies exist beneath the ice shelf. The friction velocity fluctuations due to large-scale eddies shows a repetitive pattern after 14 $t^*$ (Ramudu et al., 2018). The averaged friction velocities for n = 3, 4, 5, and 7 were 0.00169, 0.00208, 0.00255, and 0.00283 m s$^{-1}$, respectively. This difference is due to the different momentum entrainment by the turbulence intensity within the IOBL. Table 2 presents these friction velocities and the large-eddy turnover times for the four cases. We calculated the time-averaged results within the last 3 $t^*$ period for the later analysis to capture the averaged features of the flow without temporal variance.

Figure 3 illustrates the vertical sections (y = 1,728 m, domain center) of the zonal velocity, potential temperature, and salinity which are time-averaged in the 3 $t^*$ period to examine the spatial distributions of the flow structures and variables in two end-member cases (n = 3, n = 7) for turbulence intensity. In the ocean region, the velocities for weak and strong turbulence cases (upper panel of Figure 3) exhibited similar patterns for the two overturning cells in the upper ocean region (0–280 m depth). Since we did not impose the wind effect at the top boundary, we can conclude that the development of the outer overturning cell is mainly induced by the downwelling (negative buoyancy flux) of the locally concentrated salinity as well as the shear stress induced by the momentum difference between the upper region and sub-ice shelf plume. Moreover, the development of the inner overturning cell is mainly due to the upwelling of the buoyant water and the downwelling of the salt flux. Thus, the inner overturning cell is stretched in the vertical direction, whereas the outer overturning cell is stretched in the horizontal direction. The downwelling convection in which the two overturning cells share pushes the sub-ice shelf plume, moving the stratification line (280 m depth) near the ice shelf to a depth of approximately 350 m. Positive buoyancy at the left

side (near the ice front) of the inner circulating cell originated from the meltwater created at the ice-shelf base near the ice front. In this study, we refer to this water as the "positively-buoyant ice shelf water (PISW). In contrast to the sub-ice shelf plume, which has a neutral buoyancy near the ice front, PISW has a strong buoyancy. Consequently, PISW is a major contributor to the formation of the inner overturning cell. Because the temperature of the sub-ice shelf plume is higher than the local freezing temperature at a depth of 280 m, vigorous basal melting occurs (Figure 7) and creates the PISW. At zonal

distances from 1,280 to 1,600 m, this PISW mixes with the outer ocean and exhibits a temperature that is approximately 0.1 °C lower than the surface freezing temperature, affecting the vigorous sea-ice formation near the ice front (middle panel of Figure 3). As shown in the lower panel of Figure 3, the upwelling of PISW causes an upward movement of the isopycnal line (identified by potential density). Except for the open ocean where overturning cells are dominant, a well-stratified feature is observed below a depth of 350 m.

A noticeable difference between the two cases is observed near the ice front and beneath the ice shelf. At depths from 280 to 320 m (IOBL region), relatively high zonal velocity beneath the ice shelf is observed in the strong turbulence case (n = 7). After passing the ice shelf, this relatively high-speed current flows in a direction perpendicular to the ice front. In the weak turbulence case (n = 3), a relatively low velocity is observed beneath the ice shelf and the current rises after passing the ice shelf. These momentum differences in the two cases mainly affect the magnitude and scale of the circulating cells near the ice

front. Another noticeable difference between the two cases is the different temperature and salinity in the upper ocean region. In the strong turbulence case, the increased friction velocity affects the basal melting rate and a large amount of PISW, leading to a low temperature and salinity in the upper ocean region.

To examine the sea-ice distribution and PISW effect, we illustrate the horizontal distributions of the 3 $t^*$ time-averaged freezing rate (sea-ice formation) at the sea surface, as shown in Figure 4. Because the interfacial values, the atmospheric forcing, and friction velocity are the same in all the four cases, the difference in the freezing rate originates only from the ocean

conditions. Although the amount of PISW is small in the weak turbulence case, most of its PISW is rising along the ice front owing to a low flow momentum within the IOBL. Otherwise, some of the PISW in the strong turbulence case is also rising along the ice front, but most of them are advected to the upper mixed layer apart of the ice shelf. As a result, different patterns of the freezing rate are observed in the two cases. In the weak turbulence case, most of the freezing (sea-ice formation) is

concentrated right after the ice shelf, indicating a maximum freezing rate of 8.9 m yr$^{-1}$. Although a maximum freezing rate of 7.16 m yr$^{-1}$ is observed right in front of the ice shelf in the strong turbulence case, the spatial-averaged freezing rate is higher than that in the weak turbulence case. An interesting point in the strong turbulence case is the heterogeneous pattern of the freezing rate in the meridional direction. This feature is strongly related to the PISW layer formed by the PISW upwelling (Figure S2). In the weak turbulence case, the PISW upwelling occurs along the ice front edge, forming a strong and narrow

PISW layer near the ice front with strengthened inner overturning cell. However, heterogeneous patterns of the freezing rate are observed in the strong turbulence case, because the PISW layer near the ice front is wide with a weakened inner overturning cell, permitting the larger baroclinic disturbance caused by sloped isopycnals. This heterogeneous pattern of the freezing rate is comparable to the disturbance scale (2,066 m), as identified from the Rossby radius of deformation which represents the

length scale the rotation effect is dominant. This scale is obtained by depth-averaged buoyancy frequency and depth between the sea surface and IOBL bottom.

## 3.2 Validation of simulation results

The CTD and LADCP observation data shows the existence of sub-ice shelf plumes, which exhibit lower temperatures than the surface freezing temperature. Above a depth of approximately 350 m, a well-mixed feature of potential temperature and salinity are observed, suggesting a strong mixing in this region. Two interesting results were obtained that are difficult to explain via observations. One of them is the existence of relatively low-temperature freshwater (–1.96 °C, 34.65 psu) at a depth of 100 m. Because of the sea-ice formation (latent heat and salt flux) at the polynya of the NIS region in late summer season, we suggest that the low-temperature freshwater is produced from the ice shelf base, and not from the sea surface. This feature can be explained by the PISW upwelling process. It is observed that the PISW is advected to the open ocean region at 100 m depth in the potential temperature distribution at an instantaneous time of 21.8 $t^*$ (not shown). Another is the negative velocity (toward the ice front) above a depth of 300 m. It is a coherent feature because all the 23 LADCP observations indicated a similar feature. Because the direction of the katabatic winds is positive (away from ice front), it is not the cause of the negative velocity. We hypothesis that there is a large-scale overturning cell (counter-clockwise) caused by the shear force of the sub-ice shelf plume and downwelling by the salinity flux at the sea surface.

To validate our LES results and examine our second hypothesis, we plotted the vertical profiles of the stream-wise zonal velocity, potential temperature, and salinity at a distance of 1 km away from the ice front and CTD and LADCP observations to compare the vertical distribution of the momentum and variables related to potential density (Figure 5) quantitatively. In terms of the velocity, the LES results simulate the development of the outer overturning cell well, having vertical profiles similar to those observed in the LADCP data, although the velocity gradient of LES results is less-sharp compared to that observed in the LADCP data. This mismatch between the LES results and the observations arises from the underestimated convection force (underestimated strength of overturning cells). The potential temperature and salinity profiles of these four LES results agree with the 24 CTD profiles, in terms of magnitude and depth of peak values. The difference in the potential temperature above a depth of 350 m also arises from the underestimated strength of the overturning cells. Because the simulated downwelling of the salinity flux pushes the stratification line (isopycnal line) at a depth of 280 m near the ice shelf (Figure 5c), we speculate that the strength of the overturning cells is underestimated because of the exclusion of wind stress. However, in terms of the development of the overturing cells, similar vertical structure of temperature and salinity, we conclude that the LES results are similar to the *in situ* observations of the oceanic environments.

Because the velocity gradient between the freestream velocity at 572 m and the velocity at the sea surface is similar in all the cases, the velocity profiles of the LES results at 1 km away from the ice front are similar. However, the different turbulent intensities affect the different momentum transfer into the IOBL, resulting in different melting rate. Due to a difference in the melting rate, the magnitude of the potential temperature in the upper mixed layer in all the cases are significantly different: in the upper mixed layer, the potential temperature in the four cases are approximately –1.925, –1.93, –1.932, and –1.943°C. This

difference is due to the amount of PISW created from the ice shelf base, not due to the differences in the advection of PISW, because the total potential temperature in the strong turbulence case is lower than that in the weak turbulence case (Figure 4). Similar features with the effect of PISW are also observed in the salinity profiles.

In the LES model, the filtered Navier–Stokes equation is solved with the modelled effect of small-scale eddies to reduce the computational costs. Therefore, the criteria for "small-scale" are important; these criteria are determined by the grid size. To evaluate the grid size and the parameterization of small-scale eddies, it is necessary to confirm that the turbulence characteristics of the LES result are similar to the turbulence characteristics of the inertial subrange in which energy cascading occurs. We obtain the one-dimensional turbulence energy by integrating the inner product of the wavenumber and two-point

correlation calculated along the single line in the y-direction. The one-dimensional turbulence energy spectra at a depth of 291 m (within the IOBL) in weak and strong turbulence cases are plotted in Figure 6. Moreover, we examined different zonal locations (x = 400, 800, 1200, 1600, 2000 and 2400 m) to observe a spatial transition of the IOBL flow. For the high wavenumbers which are comparable to the grid scale, the energy spectra slope of the LES results is slightly smaller than –5/3 slope of the Kolmogorov scaling in the inertial subrange. These are similar to –1 slope of the Batchelor regime, indicating that

the resolved turbulence has anisotropic characteristics in high Schmidt number. For the weak turbulence case, spatial transition of turbulence energy at the IOBL region is observed. As the direction of the flow is away from the inlet boundary, the turbulence energy is gradually increased beneath the ice shelf. In the open ocean, turbulence energy are similar. For the strong turbulence case, the trend of energy spectra is similar to that for the weak turbulence case, except at x = 1,200 m. The highest turbulence energy is observed near the ice front (x = 1,200 m), showing high momentum transfer near the ice front edge.


### 3.3 IOBL characteristics in the sub-ice environment

        The afore-mentioned analysis shows that the LES model adequately resolves the oceanic flow beneath the ice shelf with the thermohaline dynamics, such as IOBL dynamics, PISW upwelling and convective downwelling by the salt flux at the sea surface. Next, we explore the flow characteristics of the IOBL beneath the ice shelf using the validated LES results. Since we

assumed a flat base of the ice shelf in this study, the buoyant force of PISW does not accelerate the PISW at the ice shelf base. Driving forces within the IOBL flow are the shear forces caused by the momentum of the sub-ice shelf plume and the stratification force (stabilizing force) caused by melting. In this section, we comprehensively analyze the oceanic flow characteristics to reveal the relation between the flow physics and melting patterns within the IOBL beneath the ice shelf.

        Figure 7 illustrates the horizontal distributions of 3 $t^*$ time-averaged meting rate at the ice shelf base. Evidently, different

magnitudes of the melting rates are observed in the weak and strong turbulence cases. Except the magnitude of the melting rate, the overall trend in the two different turbulence cases is similar. In the region close to the inlet boundary and ice front, the melting rate is high. Because the inlet boundary conditions are theoretical profiles, the transition for a fully developed IOBL flow is needed. As shown in Figure S3 in supplementary, turbulence intensities in the weak and strong turbulence cases are fully developed after 312 and 336 m distances, respectively. Therefore, we exclude the region with non-developed

turbulence in the analysis. The averaged values in the two different turbulence cases are 0.092 and 0.153 m yr$^{-1}$, respectively, showing a 66 % difference in the melting rate near the NIS front. The melt rates obtained in this study are significantly low compared to those reported by Wray (2019) (0.45–0.95 m yr$^{-1}$) and estimated via the Cryosat-2 satellite observation during 2010–2018 ($1 \pm 0.6$ m yr$^{-1}$) at the NIS ice front region (Adusumilli et al., 2020). However, the melt rate in our LES results is comparable to the observed melt rate, considering the difference in the thermal driving (0.056 °C (2.116 °C – 2.06 °C) in our

LES simulations and 0.14 °C (2.0 °C – 1.86 °C) in the study of Wray (2019)). In this study, only the effect of sub-ice shelf plume (formed by HSSW) was considered, but the observations included the effects of relatively warm Antarctic surface water and sub-ice shelf plume, resulting in a difference in the thermal driving and melt rates. If the melt rate in this study is assumed to be 0.12 m yr$^{-1}$ (averaged value of 0.092 and 0.153), we can estimate that 12–25 % of the total basal melting near the NIS front is due to sub-ice shelf plumes. The rest portion is related to surface water intrusion with wind stress and tide effects.

To observe and quantify the velocity structure within the IOBL in the strong turbulence case and the Ekman layer development, we plotted vertical profiles of the velocities, turbulence intensity and horizontal momentum flux beneath the ice shelf (Figure 8). At 280–400 m depths, the zonal velocities exhibits different structures for all the four cases. The strong turbulence case displays the smallest mean shear gradient but the largest turbulence intensity, whereas the weak turbulence case has opposing features. Since the turbulence kinetic energy production is proportional to the mean shear gradient and

turbulent shear stress (Pope, 2000), the turbulent shear stress is the highest in the strong turbulence case, indicating that the turbulent shear stress produces a large portion of turbulent kinetic energy production. Due to high turbulent shear, a strong frictional Ekman layer with negative meridional velocity (flows to the right of geostrophic flow) develops within the IOBL. Moreover, the frictional Ekman layer depths for the weak and strong turbulence cases are 11 and 17 m, respectively. These depths are comparable to the depths (11.9 and 19.6 m, respectively) estimated based on the friction velocity and Coriolis

parameter (Coleman, 1990).

    Figure 9 shows the vertical profiles of the vertical fluxes of momentum and heat beneath the ice shelf. As shown in Figure S4 which depicts the vertical buoyancy flux profiles, the PISWs in both the weak and strong turbulence cases have a stabilizing effect because of their positive buoyant characteristics. Moreover, the PISW in the weak turbulence case exhibits a larger buoyant force than that in the strong turbulence case. Thus, subsequently, we determine the PISW bottom (297 m). Combining

the regions, where the meltwater and its stabilizing effect dominate, with that where the turbulence-induced heat entrainment is vigorous (5 % of the maximum heat flux), the IOBL bottom is determined. In the strong turbulence case, the vertical momentum flux is negative, and its maximum is located within the IOBL (IOBL bottom: 319 m depth). This implies that the momentum entrainment from the sub-ice shelf plume to IOBL is effective, resulting in a large heat entrainment. However, the depth of the maximum negative flux in the weak turbulence case is located at 347 m, i.e., slightly away from the IOBL. This

difference causes difference in the heat flux magnitude at the PISW bottom. Because the IOBL flow is quasi-steady, a similar temperature within the IOBL is maintained under heat entrainment from the sub-ice shelf plume and cooling effect of the PISW advection. The positive heat flux at 280–319 m depths and negative heat flux at 320–400 m depths are caused by the large-scale turbulence convection, showing the occurrence of heat entrainment from the sub-ice shelf plume and cooling effect

of the PISW advection. The integrated area of heat flux represents the heat entrainment for the basal melting and PISW

formation. The maximum positive heat flux for the weak and strong turbulence cases are 138 and 213 W m$^{-2}$, respectively, with a 54% difference. This difference is comparable with that (66 %) in the melting rate near the ice front, confirming that the basal melting rate is proportional to the amount of heat flux and entrainment, because of the flow advection penetrating the stratified IOBL.

## 4 Discussions

**4.1 Ocean environment near the NIS front**

The main processes of the ice–ocean interactions related with ice mass balance in the cold-water cavities are the basal melting near the grounding zone and refreezing near the ice front (Dinniman et al., 2016; Jacob et al., 1992; Petty et al., 2013). One of these processes is triggered by the intrusion of the HSSW. As the plume generated by the HSSW gets closer to the ice front, it flows outward, losing its buoyancy (Smithie & Jacobs, 2005). In this study, we observed this neutral, sub-ice

shelf plume and simulated the oceanic environment affected by this plume. The main findings and conclusions are summarized in Figure 10. First, we observe the development of two overturning cells near the ice front. The inner overturning cell is caused by the upwelling of the PISW along the ice front slope and downwelling at concentrated salinity flux (local salt maximum) induced by the sea-ice formation and sub-ice shelf plume. To examine this, we set zero velocity at depths from 0 to 280 m, with no wind effect. Moreover, the sea-ice formation (1.38 m yr$^{-1}$) is determined from the sensible heat flux which is calculated

using the wind speed and air temperature data obtained from the AWS. Through this procedure, we resolve the two overturning cells in the upper mixed layer. Second, we identify the role of turbulence within the IOBL. Notably, a higher turbulence intensity results in a large amount of ice melting near the ice front. A high turbulence intensity causes a large momentum transfer, resulting in an increased melting and relatively high-speed currents within the IOBL. The large momentum causes the IOBL current (positive zonal velocity) to flow perpendicular to the ice front after it passes the ice shelf.

The horizontal scales of the overturning cells differ slightly for the weak and strong turbulence cases. The horizontal scale (492 and 408 m for the cases with weak and strong turbulence, respectively) of the inner overturning cell is calculated, based on the locations of the local salt maximum at the sea surface. Because we observed the negative velocity at the sea surface, positive velocity of sub-ice shelf plume, and downwelling of the salinity flux, we can conclude that an outer overturning cell exists. However, we cannot observe the exact horizontal scale of the outer overturning cell owing to the limitation of the

domain scale in this study. Because the flow with the sub-ice shelf plume beneath the ice shelf is a backward-facing step flow (reattachment flow with geometry), we can estimate the horizontal scale of the outer overturning cell based on the reattachment length (Rygg et al., 2011). For the oceanic flow at a high Reynolds number ($2 \times 10^6$), we observe the development of the outer cell (5.3–8.4 $h$ ($h$ is geometry height)) and inner cell (1.3–1.5 $h$). As the ice shelf thickness is 280 m, we suggest a 364–420 m inner overturning cell and 1,484–2,352 m outer overturning cell.

PISW upwelling is important for the formation of modified Antarctic surface water and ocean circulation near the ice front. In a previous study on the freshwater wedge in the Western Ross sea, freshwater layer at sea surface was observed. They hypothesized that this freshwater mainly originates from the sea ice melting, ISW outflow, and frontal ablation. The PISW upwelling and its accumulation observed in this study could evidentially support the "basal melting input" as a feasible wedge formation mechanism proposed by Malyarenko et al. (2018). If we consider the relatively warm Antarctic surface water with

inclusion of frontal ablation in a future study, we can investigate and quantify the contributions of the basal melting and frontal ablation in the freshwater input in the NIS region. In the study of meltwater outflow and its vertical structure by Garabato et al. (2017), they observed that the injection of the high-buoyancy, meltwater-rich glacially modified water triggers overturning via centrifugal instability near the ice front. For the vertical velocity of the PISW (Figure S5), we observed a high positive velocity (0.04 m s$^{-1}$) near the ice front and a negative velocity (0.02 m s$^{-1}$) at the local salinity maximum. Given that

approximately 0.02 m s$^{-1}$ of the vertical velocity is observed at 1 km away from the ice front by Garabato et al. (2017), our findings can support the ascent of meltwater outflow observed in this previously reported study. Differ to centrifugal instability in the Pine island ice shelf, we can demonstrate the symmetric instability in this study (Figure S6 and S7). This difference is caused by the different directions of the current near the sea surface and the exclusion of the katabatic wind effect.

## 4.2 Limitations of the LES experiments

As shown in the previous LES on the ice–ocean boundary (Ramudu et al., 2018; Vreugdenhil and Taylor, 2019), the LES model is a powerful tool for resolving the flow and its three-dimensional structures with the change of mass and energy, with a high accuracy and low computational cost. Therefore, employing the numerical approach such as a regional ocean model and the LES model is one of the best solutions for resolving the three-dimensional structures in the oceanic flow. Under these circumstances, we demonstrated one method for investigating the IOBL physics and ocean dynamics by combining the

numerical approach with observational data and theoretical profiles. In this study, we used the LES model with *in-situ* boundary conditions to expand the one-dimensional observation profile in the open ocean region to the three-dimensional flow-field in the open ocean and sub-ice shelf regions. Additionally, we set the interfacial values ($\theta_b$, $S_b$) in the liquidus condition using ambient values ($\theta_f$, $S_f$) obtained from the CTD observational results. In this study, to evaluate the applicability of our proposed method, we validated our results with observations, in terms of overturning cell characteristics, vertical structures of

temperature and salinity. In a future work, this proposed methodology will be validated using multi-dimensional sub-ice shelf observations made using advanced instruments such as autonomous underwater vehicles, glider, etc. Moreover, the boundary conditions for the sub-ice shelf region, the effect of wind stress and salt flux changes due to precipitation and evaporation will be investigated for simulating the IOBL and overturning cells more realistically.

In Terra Nova Bay, the northeastward katabatic wind is dominant and it drives the along-front current (Guest, 2021;

Malyarenko et al., 2019). If we included the wind effect at the sea surface, the horizontal mixing may be enhanced, advecting the fresh meltwater of the PISW layer near the ice front to the open sea. In terms of overturning cells, the strength and horizontal scale of the outer overturning cell may decrease, as the wind stress reduces the shear stress between the sea surface and the

sub-ice shelf plume. The weakened outer cell, in turn, weakens the inner (secondary effect) cell. However, the strength and scale of the inner overturning cell may be similar to that obtained in the present study, because the wind stress at the sea surface is imposed at the upper region of the inner overturning cell. In summary, if the wind stress effect is included, similar scale of the inner circulating cell and a decreased scale of the outer circulating cell near the ice front can be expected.

In this study, we did not include the melting or freezing effects at the lateral side of the ice shelf. Because thermal driving at depth of 146 m on the lateral side of the ice shelf was almost zero, the effects of melting and freezing occurred below and above 146 m depth, respectively. Because 146 m was almost near the center of the ice front, the total change in the temperature and salinity, because of the melting and freezing effects at the entire ice front, was not significant. However, this feature in the ice mass change at the ice front might be related to the shape of the ice front edge.

In terms of the frazil ice processes, we only considered the latent heat release and salt flux produced by the sea-ice formation at the sea surface. However, suspended frazil ice, formed by adiabatic cooling, can occur in the region of PISW upwelling at the ice front and upwelling of the sub-ice shelf plume (in this study, this upwelling occurred outside the study domain). The effects of the frazil ice dynamics (e.g. crystal growth rate, nucleation, and concentration) on the sea surface or ice shelf water (PISW and sub-ice shelf plume) should be investigated, because the change in plume characteristics as well as the temperature and salinity are strongly related to the frazil ice dynamics (Galton-Fenzi et al., 2012; Rees Jones and Wells, 2018). Because the inclusion of suspended frazil ice affects the increased plume velocity and decreased plume density, the upwelling of the PISW can be strengthened. This increases the strength of the inner overturning cell. Owing to a high plume velocity and decreased plume density, only a few precipitations within the PISW occur at the lateral side of the ice front. However, if the ice-front geometry is modified through an increased frazil ice formation by the freshwater wedge formation, precipitation on the modified geometry might increase, exhibiting a nonlinear effect on the change in the ice-shelf geometry.

In this study, we employed theoretical power-law profiles of the velocity, which exhibit different turbulence intensities, because observational data on the vertical structures beneath the ice shelf are rarely available. In the flux Richardson number in Table 2, the relationship between the stratification and turbulent mixing is not clear, even though we imposed different turbulence intensities. This implies that the negative feedback (enhanced buoyancy fluxes) of the PISW increases as the turbulence intensity and its entrainment increases. In this study, we performed a preliminary assessment of the oceanic structures in a sub-ice shelf environment. The constant model coefficient ($C_m$) in the SGS model is applied in the IOBL flow near the wall and the flow within the sub-ice shelf plume to reduce the computational costs. In the future study, correction in the coefficients will be required, based on the high-resolution study of near-wall turbulence. The main findings of this study can be used for understanding the oceanographic phenomenon near the sub-ice shelf cavity. If direct observational data for the IOBL flow structures and turbulence characteristics in the sub-ice shelf environment can be obtained, then this study can be improved by the comparing LES results with the observations, and by introducing corrections into the ambient values and transfer coefficients. Other important factors that were not included in this study should be addressed and investigated in future studies such as the effect of ice shelf bathymetry (shape and slope) and surface roughness on the turbulence characteristics, temporal variability of the sub-ice water plume, wind stress effect, and consideration of the lateral melting at the ice front. A

better understanding of the effect of important factors on basal melting and its meltwater dynamics will help in improving the parameterizations (e.g. vertical mixing within the IOBL and sea-ice formation and behavior) in the regional ocean model.

## 5 Conclusions


We successfully simulated the IOBL flow with a sub-ice shelf plume, the melting effect beneath the NIS, and the freezing effect at the sea surface using a LES model and boundary conditions based on *in situ* observations. Since the inflow profile beneath the ice shelf are difficult to observe in *in situ* observation, we assumed the theoretical power-law profile to describe the different turbulence intensities. The flow simulated for a period of 96 h reached a quasi-steady state after 14 $t^*$, exhibiting

a large variance and repetitive pattern with temporal convergence. To validate the LES results for the four different turbulence intensities, the simulated zonal velocity, potential temperature, and salinity were compared with the 24-h CTD and LADCP observational data obtained near the NIS in the Terra Nova Bay of the western Ross Sea, during a 2016–2017 shipboard survey. The simulation results agree well with the observations considering the development of two overturning cells and thermohaline properties.


The different turbulence intensities in the IOBL flow beneath the ice shelf resulted in significantly different melting features and flow dynamics. With increased friction velocity, the melting rate in the strong turbulence case increased by 66 % compared to that in the weak turbulence case, maintaining the stratification intensity (similar flux Richardson number). In the strong turbulence case, distinct features such as higher basal melting (0.153 m yr$^{-1}$), weak upwelling of the PISW, and a larger sea-ice formation were observed, suggesting a relatively high-speed current within the IOBL because of the large momentum

transfer. Comparing with the observations, we estimated that 12–25 % of the total basal melting near NIS front is caused by the sub-ice shelf plumes. We observed that the sub-ice shelf plume, PISW upwelling, and downwelling of concentrated salinity flux compose two overturning cells near the ice front.

**Data availability**

The numerical model, PALM 6.0 (Rev:4552M) used in this study is available at (https://palm.muk.uni-hannover.de/trac).

Detailed model configurations with melting effect parameterization are described in detail in method section. The data used are all publicly available and can be found via the relevant citations.

**Acknowledgments**

This work was sponsored by a research grant from the Korean Ministry of Oceans and Fisheries (KIMST20190361; PM21020).

**Competing interests**

The authors declare that they have no conflict of interest.

**Author contributions.**

JSN, EKJ, WSL conceived the study. JSN modified PALM model and the code of surface fluxes for ice melting, and JSN conducted a suite of simulations. JSN, TK, STY, SY, and JL conducted CTD, LADCP observation and its analysis. JSN and EKJ wrote the manuscript with contributions from all authors.

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

**Table 1. List of model parameters and constants**

| | | | |
|---|---|---|---|
| $\lambda_1$ | Freezing temperature salinity coefficient | $-0.0573$ | °C kg g$^{-1}$ |
| $\lambda_2$ | Freezing temperature constant | $0.0832$ | °C |
| $\lambda_3$ | Freezing temperature depth coefficient | $-7.53\times10^4$ | °C m$^{-1}$ |
| $\Gamma_S$ | Salt turbulent transfer coefficient (sea surface, ice shelf bottom) | $2\times10^{-4}$ [a], $2.6\times10^{-4}$ [b] | - |
| $\Gamma_\theta$ | Heat turbulent transfer coefficient (sea surface, ice shelf bottom) | $5.8\times10^{-3}$ [a], $8\times10^{-3}$ [b] | - |
| $c_w$ | Specific heat capacity of pure water | 3974 | J kg$^{-1}$°C$^{-1}$ |
| $L_i$ | Latent heat of fusion | $3.35\times10^5$ | J kg$^{-1}$ |
| $\rho_w$ | Density of water | 1028 | kg m$^{-3}$ |
| $\rho_i$ | Density of ice | 917 | kg m$^{-3}$ |
| $z_0$ | Surface roughness (sea surface, ice shelf bottom) | 0.001, 0.005[c] | m |
| - | Ice shelf thickness | 280[d] | m |
| $\theta_f$ | Local freezing temperature (sea surface, ice shelf bottom) | $-1.9$, $-2.115$ | °C |
| $\theta_a$ | Ambient temperature (sea surface, ice shelf bottom) | $-1.9$, $-2.06$ | °C |
| $\theta_b$ | Interfacial temperature (sea surface, ice shelf bottom) | $-1.879$, $-2.092$ | °C |
| $Sa_a$ | Ambient salinity (sea surface & ice shelf bottom) | 34.69 | Psu |

| | | | |
|---|---|---|---|
| $Sa_b$ | Interfacial salinity (sea surface, ice shelf bottom) | 34.9425, 34.286 | Psu |
| $k_\theta$ | molecular diffusivities of heat | $1.3\times10^{-7}$ | $m^2\ s^{-1}$ |
| $k_S$ | molecular diffusivities of salt | $7.2\times10^{-10}$ | $m^2\ s^{-1}$ |
| $u_*$ | Friction velocity (sea surface, ice shelf bottom) | 0.026, calculated | $m\ s^{-1}$ |
| | Wind speed, air temperature in AWS | 16.23, −7.76 | $m\ s^{-1}$, °C |
| $Q_s$ | Sensible heat flux at sea surface | 164.88[e] | $W\ m^{-2}$ |
| $F$ | Sea-ice formation | 1.38[e] | $cm\ day^{-1}$ |

**a – Heorton et al. (2017)**

**b – based on friction velocity (0.168 m s⁻¹) and thermal driving (refer to Vreugdenhil and Taylor (2019))**

**c – Smooth ice with melting case (C_d = 0.001) in Gwyther et al. (2016)**

**d – Stevens et al. (2017)**

**e – Thompson et al. (2020)**


**Table 2. Main values of four different cases ($Ri_f = \dfrac{\frac{g}{\rho_{\theta,0}}(\overline{\rho'_\theta w'})}{-\overline{u'w'}(\frac{\partial U}{\partial z})}$)**

| Case | Friction velocity at ice shelf (m s⁻¹) | IOBL depth (m) | Large eddy turnover time, t* (hour) | Averaged melt rate (m yr⁻¹) | Averaged freezing rate (m yr⁻¹) | Flux Richardson number, $Ri_f$ |
|---|---|---|---|---|---|---|
| n=3 | 0.001686 | 30 | 4.94 | 0.092 | 2.628 | 0.0464 |
| n=4 | 0.002077 | 33 | 4.41 | 0.109 | 2.72 | 0.0418 |
| n=5 | 0.002548 | 36 | 3.93 | 0.139 | 2.671 | 0.0599 |
| n=7 | 0.002765 | 39 | 3.92 | 0.153 | 3.142 | 0.0378 |

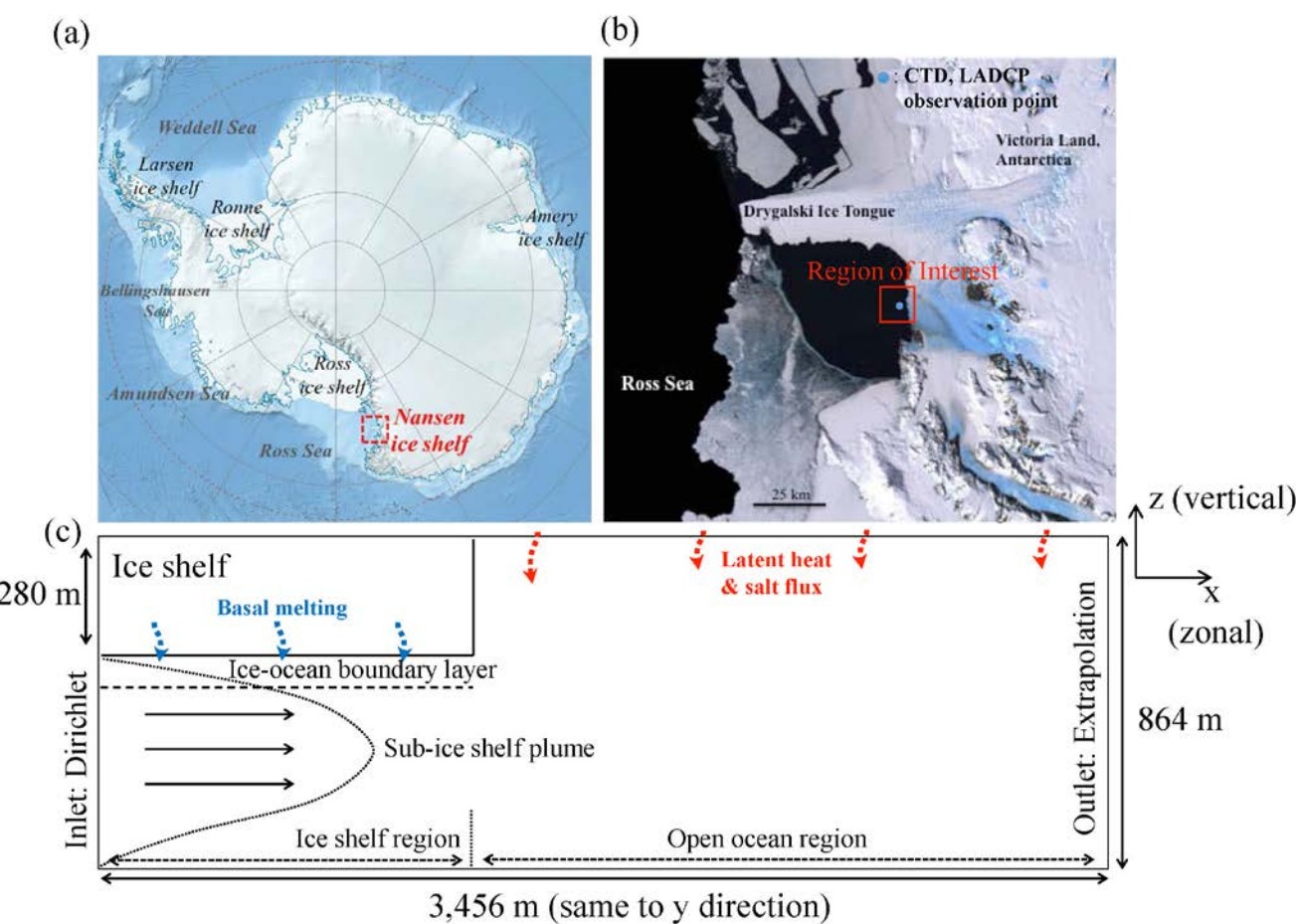

**Figure 1: Region of interest and simulation domain configuration. (a) Map of Antarctica. The red box shows the study are, Terra Nova Bay, in the western Ross Sea. (b) Region of interest where the CTD and LADCP surveys were conducted in the 2016–17 shipboard survey. (c) Simulation domain and boundary conditions.**


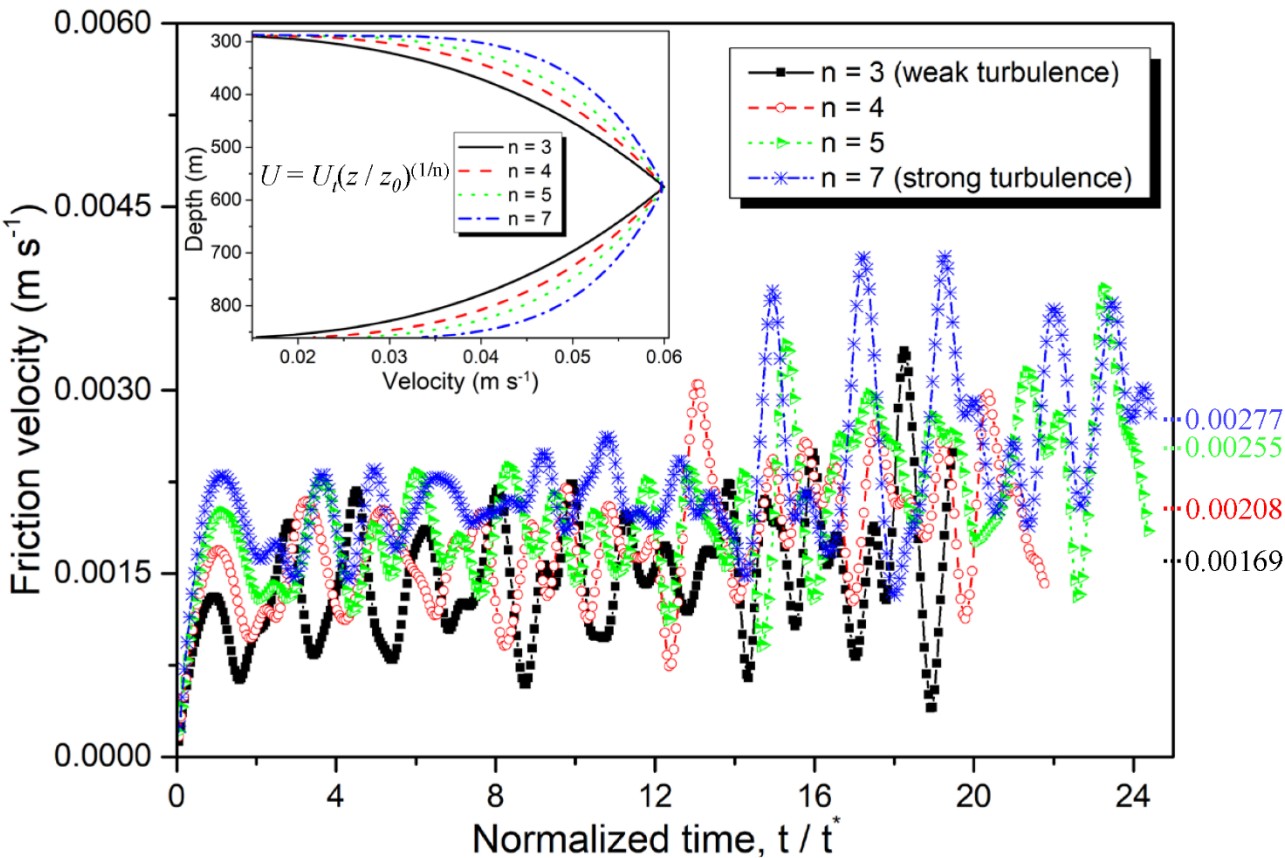

**Figure 2: Time series of the friction velocities in all the four cases. The total time (96 h) was normalized by each large-eddy turnover time, t\*, which was calculated from the overturning eddy scale (ice shelf-ocean boundary layer (IOBL) scale) divided by the friction velocity. The inset figure shows four different theoretical profiles of the velocity.**

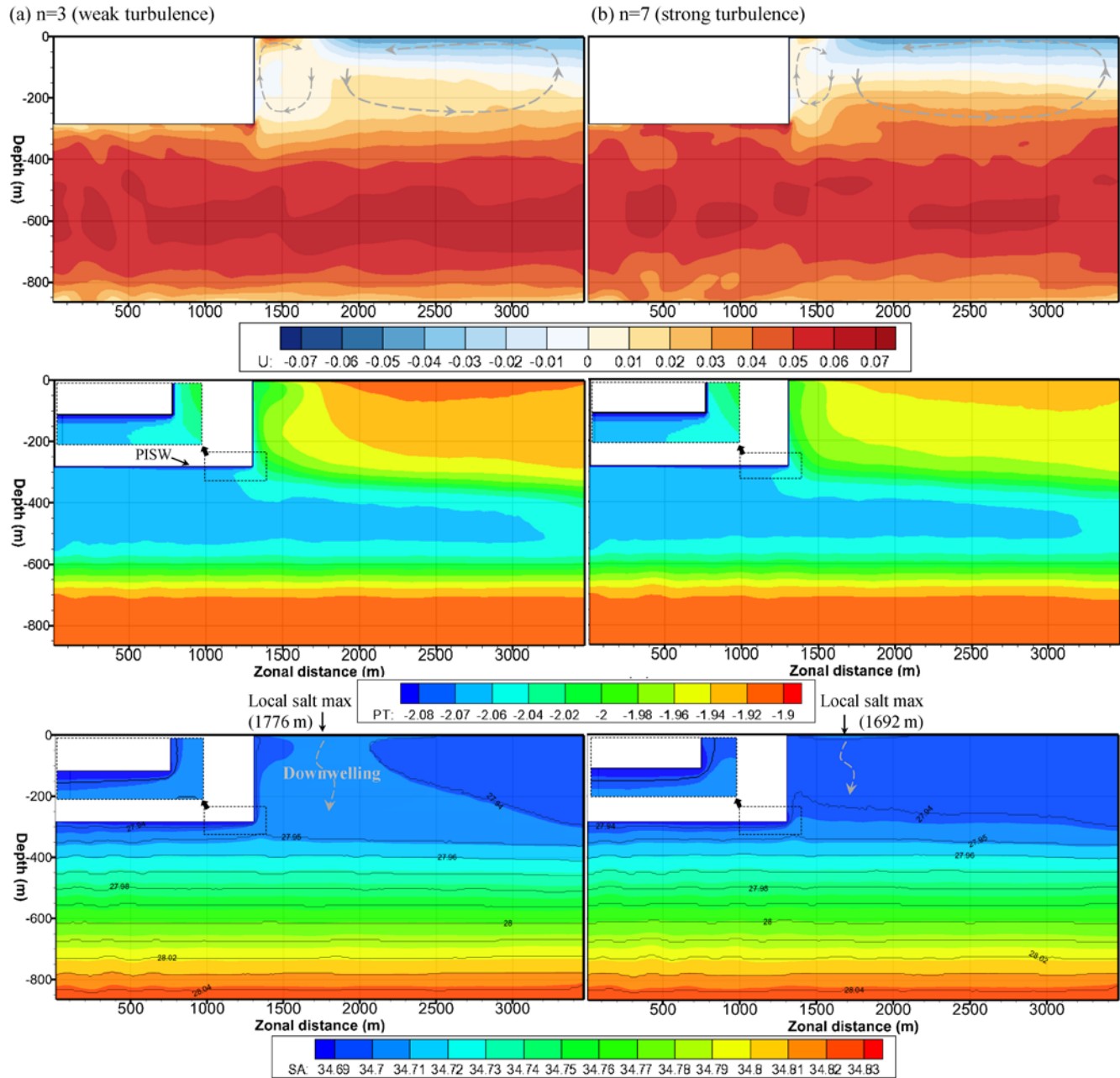

**Figure 3: XZ cross-sectional contours (y = 1,728 m, domain center) of the zonal velocity, potential temperature, and salinity in the n = 3 (left) and n = 7 (right) cases. In these contours, the zonal direction is perpendicular to the ice shelf front. These results are time-averaged in the last 3 t\* period. Upper panel: zonal velocity, middle panel: potential temperature, and lower panel: salinity. In the lower panel, the isopycnal lines are identified by the potential density (kg m$^{-3}$) with 0.01 intervals.**

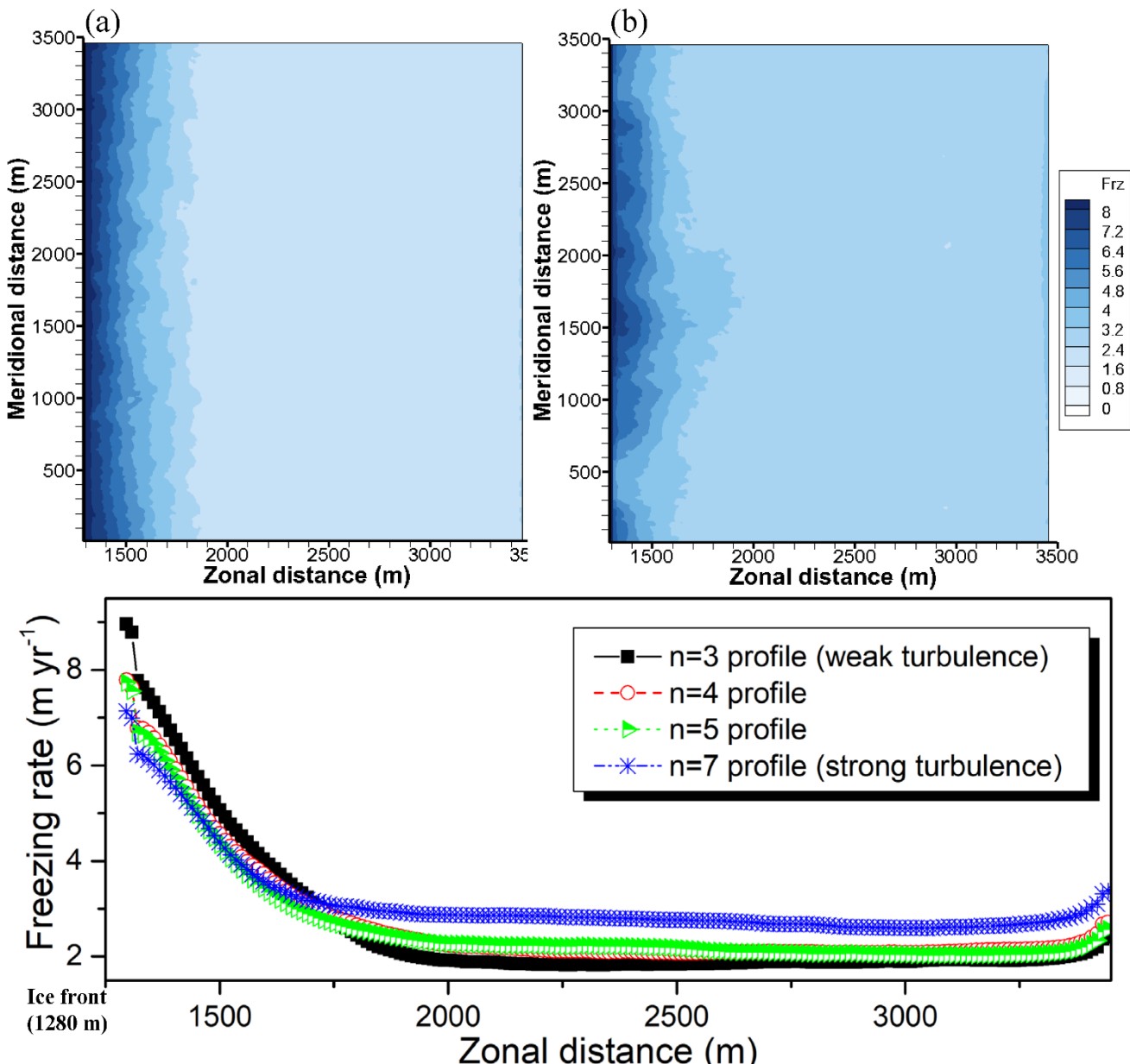

Figure 4: XY horizontal distribution of the 3 t* time-averaged freezing rate (m yr⁻¹) at the sea surface. (a) n = 3, weak turbulence. (b) n = 7, strong turbulence (c) Zonal-spatial distribution of the freezing rate in the four different turbulence cases. These values are averaged along the meridional direction.

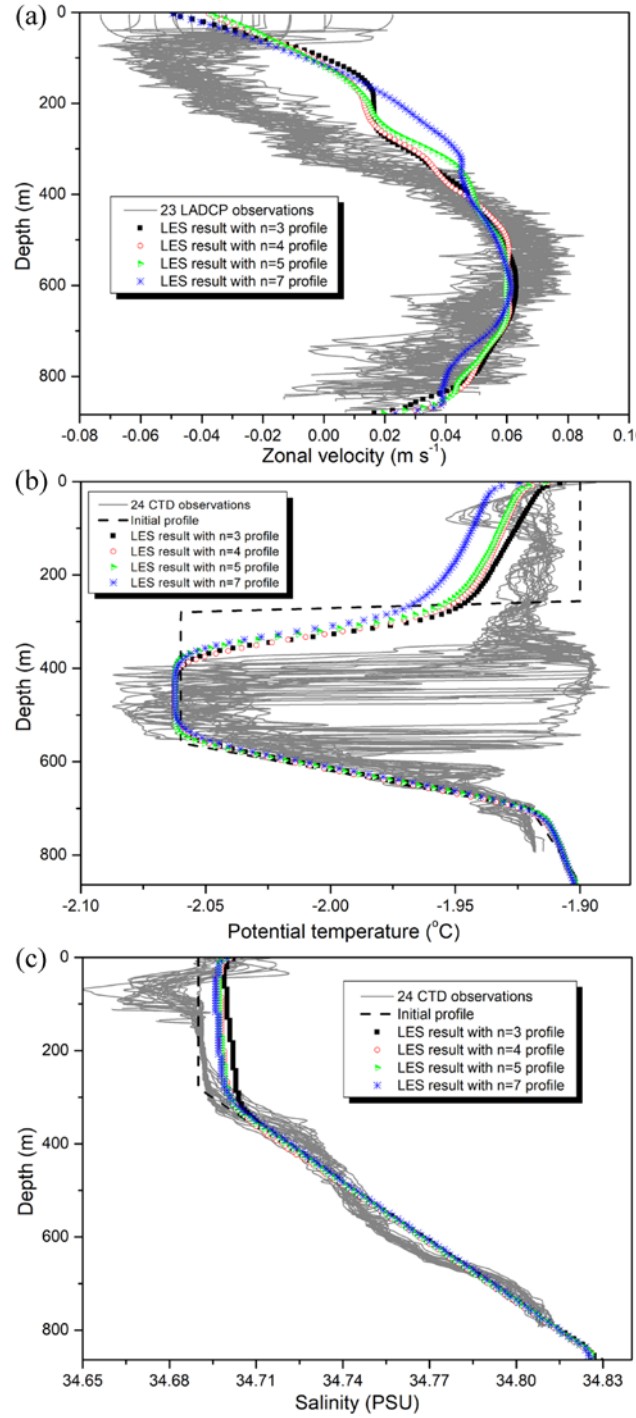

**Figure 5: Vertical profiles of the velocity, potential temperature, and salinity obtained from the CTD and LADCP observational data (solid grey lines), initial profiles (black dashed line), and the results of the large-eddy simulation (LES). (a) zonal velocity; (b) potential temperature; (c) salinity.**

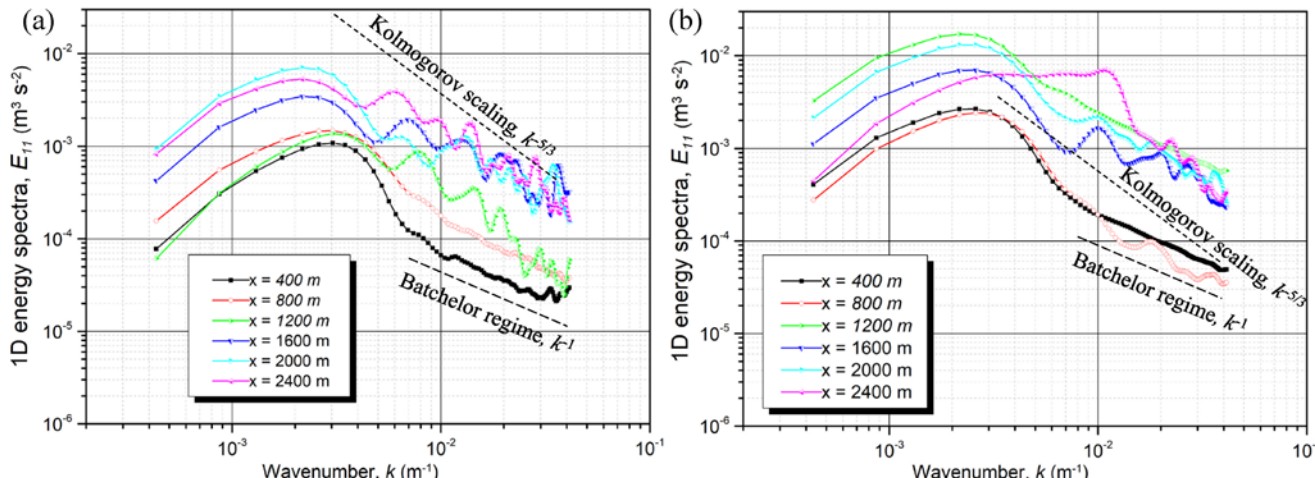

**Figure 6: One-dimensional turbulence energy spectra at a depth of 291 m at the PISW within the IOBL. (a) n = 3 and (b) n = 7.**
**Different shapes and colors represent the values at different zonal distances: 400, 800, 1200, 1600, 2000 and 2400 m. These power**
**spectra are obtained by y-direction (meridional direction) spatial assessment of time-averaged velocity at each x location. The –5/3**
**slope (Kolmogorov scaling) represents the regime of inertial subrange, whereas –1 slope (Batchelor) represents viscous-convective**
**range in high-Schmidt number.**


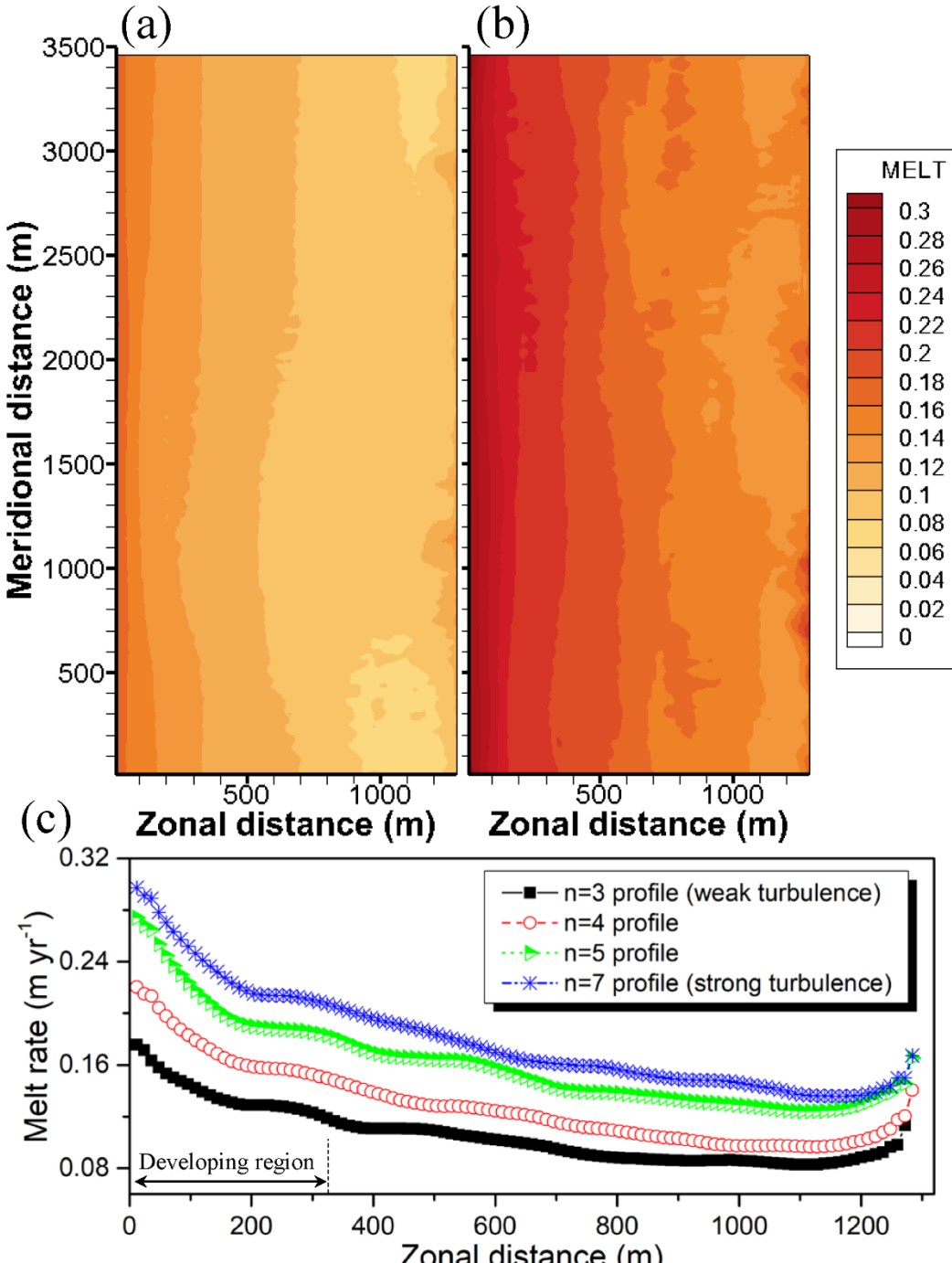


**Figure 7: XY horizontal distribution of 3 t\* time-averaged melting rate (m yr⁻¹) at ice shelf base (280 m). (a) n = 3, weak turbulence. (b) n = 7, strong turbulence (c) Zonal-spatial distribution of the melting rate in four different turbulence cases. These values are averaged along the meridional direction.**

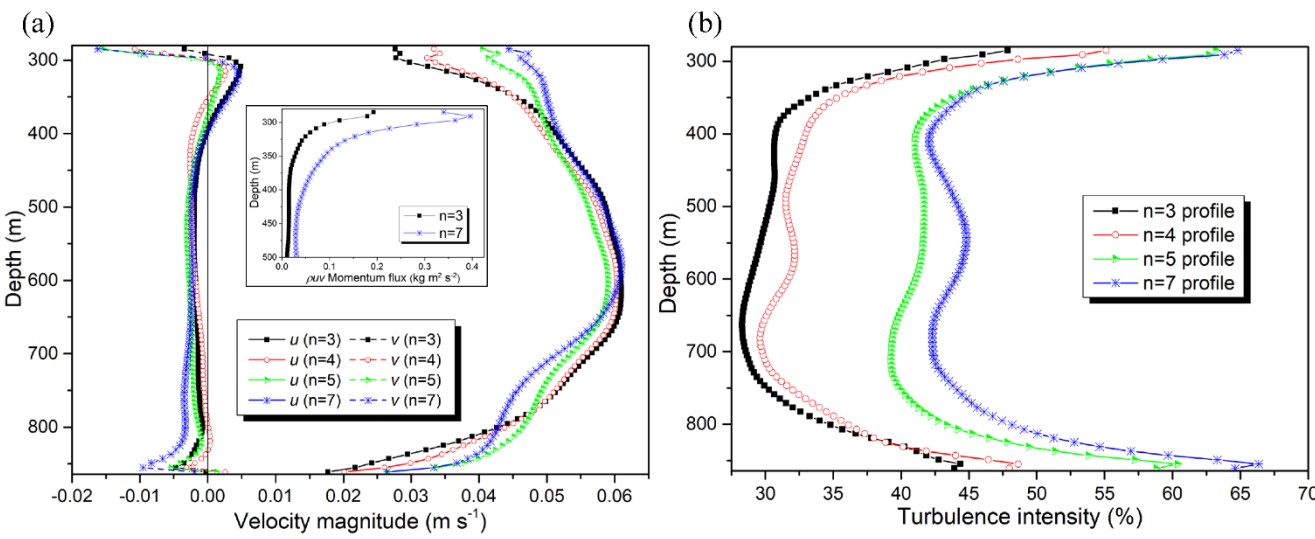

**Figure 8: Vertical profiles of (a) *u*, *v* mean velocities and (b) turbulence intensity. The inset in (a) shows the vertical profile of the horizontal momentum flux for the Ekman layer formation beneath the ice shelf.**



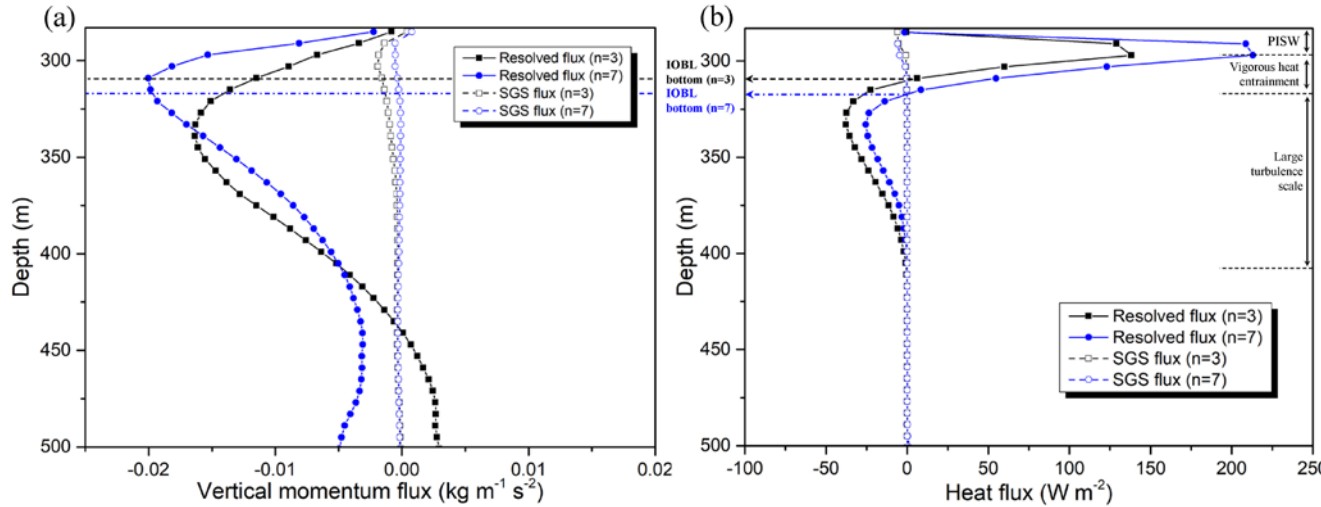

**Figure 9: Vertical profiles of the momentum fluxes and heat fluxes. The fluxes were characterized using resolved flux and subgrid scale (SGS) flux. (a) Momentum fluxes (resolved: $\rho_0\overline{u'w'}$, SGS: $\rho_0 K_m \overline{\partial u}/\partial z$). (b) Heat fluxes (resolved: $\rho_0 c_s \overline{w'\theta'}$, SGS: $\rho_0 c_s K_h \overline{\partial \theta}/\partial z$, where $\rho_0$ (1024 kg m$^{-3}$) is the reference density of seawater, and $c_s$ (4.02 $\times$ 10$^3$ J kg$^{-1}$ K$^{-1}$) is the specific heat capacity of seawater (Sharqawy et al., 2010)).**


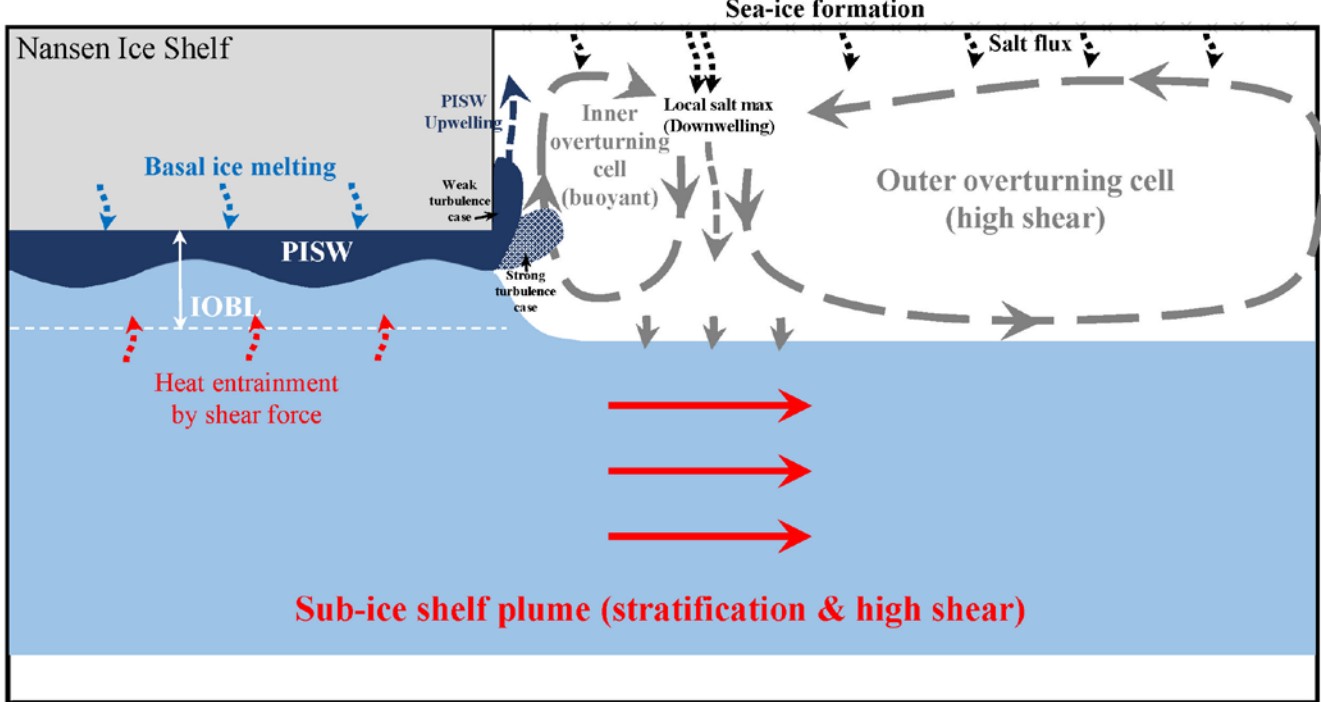


**Figure 10: Final conclusions and schematic diagram representing the oceanographic picture near the NIS front and the various phenomenon that occur within the IOBL, as resolved via LES.**