# Peer review of "Large-eddy simulation of the ice shelf-ocean boundary layer near ice front of Nansen ice shelf, Antarctica"

_The Cryosphere, 2020_

## Referee Comment (RC1) · Anonymous Referee #1 · 31 Jul 2020

General comments

There are important gaps in our knowledge of sub-ice shelf ocean dynamics, particularly in the freezing regime. Na et al. have set their sights high in this study, but the manuscript as submitted does not provide sufficient evidence of model validity. In fact, there are several theoretical reasons to believe that the dynamics are not adequately represented in the model. Their argument for validity chiefly rests on the match between simulated temperature and salinity profiles and observations. Since the model has observations as initial conditions and an inflow boundary condition, it is unclear how far from observations the simulations could evolve. Furthermore, their presenta-

tion of the results and discussion of the turbulent dynamics must be more thorough to substantiate many of their scientific conclusions. This study could be publishable after an expanded discussion of model limitations and scaled-back claims to model realism, more elaboration of simulation results, and an expanded discussion of existing literature on frazil ice dynamics.

Specific comments

The connection to existing literature is inadequate. The readers need to know the physical oceanographic context for the region to assess the strengths and shortcomings of the model and the model setup. What is driving ocean circulation in reality and in the model? We also need to know what observational estimates exist of melting and freezing rates of the Nansen Ice Shelf to determine whether the simulated freezing rates are realistic (Is Mode 3 melting present at Nansen Ice Shelf?). Regarding the result that refreezing rates are high at the ice shelf front, is there any observational basis for this pattern or are you proposing it de novo?

Please provide more context for the observations that are used to initialize the model. There should be a brief presentation of the water masses that are present in the water column and their flow orientation (only zonal velocities are presented). The location of the observations should be shown on the study area figure, and text should indicate their distance from the ice shelf front and the time span over which these observations were collected. It is unclear whether these observations were presented in Yoon et al. (2020) or whether they are published here for the first time. Furthermore, the apparent bimodal temperature distribution at ∼500 m depth also needs to be explained so that it's clear why you try to match the low temperature cases.

I have philosophical concerns about the manner in which the authors validate the model. Authors argue that the model is valid because the LES results match the observations, but the model is initialized to observations and has inflow that roughly matches the observations. Therefore, the argument seems to be that the model does

not drift too far from observations, which may be too weak an argument for model validity. It is also unclear to me where in the LES domain you are evaluating the agreement with observations. There should be a more thorough discussion of model limitations and a discussion of their possible impact on simulation results. There should be an explicit argument addressing whether your LES model can capture freezing dynamics in the absence of frazil ice dynamics. Frazil ice dynamics significantly influence IOBL evolution as documented in previous literature, which also should be cited (including the work of Galton-Fenzi). The lack of ice shelf slope should also be discussed. Furthermore, the parameterization of heat, salt and momentum fluxes at the ice base were developed for the ice melting case by McPhee et al. 1987. The applicability of this parameterization as well as the gamma_T, gamma_S exchange coefficients to the freezing case needs to be discussed.

How do you know that the strong refreezing anomaly at the ice front is not a numerical artifact related to the front geometry?

What physically determines the location of the transition from the inner to the outer region in terms of refreezing rate? How do you know that this transition isn't just characterized by the development of turbulence along-flow from a less-turbulent inflow? Is there any observational evidence for these trends in refreezing rates, either at this ice shelf or any ice shelf?

Line 13: "In particular, it is evident that, when the refreezing effect is considered, the IOBL flow can be more realistically resolved, especially upward advection from the sub-ice shelf plume and the ice front eddy." You don't show that the entrainment and eddying in the freezing case are more realistic, if by realistic you mean closer to observations.

Lines 30-45: The modes of ice shelf melting and the classification of warm and cold water cavities should be presented in the context of your study. This paragraph feels too general and unfocussed.

Line 41: This sentence should state that the basal melt rate of ice shelves is determined by the rate of BOTH heat and salt exchange, as salt exchange plays a key role in the dissolving regime characteristic of most ice shelf settings.

Line 76: "we were able to account for refreezing patterns, detailed flow structures including turbulent characteristics, fluxes and the relationship between refreezing and entrainment of supercooled water from sub-ice shelf plume within the IOBL." This makes it sound as if all of those properties were observed, when in fact you don't present observations that can be linked with those characteristics.

Line 134: I might have missed it but I can't find the value of $z\_0$.

Line 136: It is unclear how the no-refreezing case is configured. Are both heat and salt fluxes set to zero at the top boundary?

Line 145: Argue for the appropriateness of setting a CTD profile that was observed in the open ocean as inflow conditions under the ice shelf.

Line 149: Please specify how the radiation boundary condition is implemented.

Line 150: Further explanation is needed to address how dirichlet conditions at top boundary are appropriate even under the ice shelf and what they represent in the open ocean. Is this consistent with observed winds?

Line 153: With what metric is quasi-steady state evaluated?

Results: Explain in the refreezing case to what degree the increase in boundary layer temperature is due to the release of latent heat and differences in entrainment.

Line 178: "This difference is induced by high momentum exchange by refreezing and its brine rejection." This statement is unclear. Is the high momentum flux related to the destruction of stratification by brine release?

The horizontal velocity orientation throughout the text is unclear. Are zonal velocities aligned with the x-axis of the simulation domain? In some places in the text the zonal

velocity but not the meridional velocity is presented.

Line 185: "negative mean velocity": the velocity vector orientation is unclear.

Line 215: "stream-wise zonal velocity": the velocity vector orientation is unclear.

Line 186: The description of the forces responsible for the ice front eddy are unclear. Can you also describe the eddy structure more clearly? Is it a singular overturning cell spanning the length of the ice front?

Line 189: Are you saying here that convection due to brine rejection inhibits entrainment? It is unclear why this would be the case.

Line 194: "upward advection from sub-ice shelf plume" is unclear. Are you talking about entrainment due to turbulence or advection by the mean flow?

Line 195: "stratification is more dominant than flow shear" You haven't made a strong case for this in Results. Perhaps you could move this statement to the discussion and expand on it there.

Line 200: "upward flow advection" again, is this mean flow or turbulence?

Line 200: "Since there is no downward force due to brine rejection, the upper region of the sub-ice shelf plume expanded to the upward direction immediately after it passed the ice front." It's unclear what you determine the driving mechanism to be. Is it that changes in stratification determine the sub-ice shelf plume extent or changes in the degree of mixing between water masses or something else?

Results: The relationship between refreezing and entrainment of supercooled water at the ice shelf front is unclear in the manuscript. How can we tell that the higher rates of refreezing at the ice shelf front are due to entrainment as opposed to reversed flow of cooler water from the open ocean below the ice front?

Figure 5. Show the inertial subrange in wavenumber space. You mention that the low wavenumber values don't fit the -5/3 slope but aren't these wavenumbers outside the

inertial subrange? Why are these spectra so noisy? Could this indicate insufficient spatial or temporal averaging? You don't specify these details in your description of the methodology. It's also unclear why there are several curves for each case and depth. Furthermore, why are the spectra evaluated in 1-d?

Figure 7,8. There doesn't appear to be a relationship between heat flux at -281 m and refreezing rate. Can you explain why this is the case?

Lines 215-231: You address differences in velocity magnitude but not orientation.

Line 230: This caveat should be explored more deeply in the discussion.

Line 234: "it is shown that the LES model adequately resolves the oceanic flow beneath the ice shelf with the proper refreezing effect." The analysis presented in Section 3.1 does not demonstrate this. You don't show evidence for sufficient resolution (e.g., a comparison of resolved vs. subgrid energy) or the proper refreezing effect (e.g., a combination of the right theory and match to observations).

Line 240: Needs further discussion in text to explain why vertical velocity is not zero given boundary conditions.

Line 257: "strong velocity gradient" in what direction?

Line 264: the definition of IOBL should come at the first mention of IOBL in the Results or Methods.

Figure 11: Is IOBL depth the same for both regions? Inset: Is the arrow for the inset meant to correspond to a certain depth? Is sigma the same for both runs? Are there multiple PDFs for each region overlain, because I was expecting to see a single curve for each region as opposed to the scattered points?

Line 292: "This water plume refreezes..." this statement is only true for some ice shelves.

Line 304: "we used the LES model to expand the one-dimensional observation profile

in oceanic region to the three-dimensional flow-field in oceanic region and the sub-ice shelf region." This statement is imprecise and possibly misleading. It would be more accurate to say that you used the observational profile as initial and boundary conditions and then investigated spatial and temporal variability arising from brine rejection and mixing.

Line 306: "We assumed that the LES results for the sub-ice shelf region are validated if the LES results for the oceanic region are validated." See validation comment above.

Line 307: "Via an evaluation of the refreezing effect" It's not clear what this evaluation consists of and how it supports the validity of the model.

Line 314: You haven't provided much explanation of the causes of heterogeneous freezing rates and what controls the scale of turbulence features.

Line 315: what is the scale of the ice front eddy and what controls it?

Line 316: what determines the IOBL depth and does it match the observations?

Line 319: "this study can be improved by comparing LES results with observations and its feedback" What do you mean by "its feedback"?

Line 321: "If a database for flow physics in various parameters is completed" I can see what you're getting at here, but the wording here is awkward and it's unclear what you mean by "a database for flow physics."

Line 328: "convergence trend in temporal variance" of what quantity?

Technical comments

Make it clear in the introduction why you use the term "refreezing" as opposed to "freezing"

There are a few places in the text where you say that water melts, but I think you mean to say that the water mass has a contribution of meltwater from the ice shelf.

[Figure]

Specify which version of PALM you are using.

Line 17: "high shear impact"?

Line 46, 184: "the shear impact" is confusing if what you mean is more similar to "the direct impact" and does not relate to velocity shear.

Line 107: Sub-grid parameterizations need a citation.

Line 181: include a citation for the original definition of the swirling strength criterion

Line 182: "Due to" to "At its"

Line 182: "apart from the ice" to "below the ice"

Line 206: "with few dissipations" to "with little dissipation"

Line 235: "explores" to "explore"

Line 274: "to IOBL the flow"

Line 277: reference for the flatness factor

Line 281: strange to say that the vertical velocity fluctuations "have" 3 sigma.

Line 302: "the numerical approach including the LES" is confusing. Do you just mean LES on its own?

In all figures, specify whether results are derived from simulation or observations or both.

Figure 1: a. Unclear what blue shading designates. Add "sea" to Ross, Amundsen, Weddell, Bellingshausen labels.b. Show ice shelf boundaries and label sea ice areas. c. label ice shelf length dimension.

Figure 6. b. Show local freezing point, especially given that you claim supercooled water entrainment.

[Figure]

Figure 9. I can't tell the sign of vertical velocity without a reference zero line.

[Figure]

---

## Referee Comment (RC2) · Anonymous Referee #2 · 31 Aug 2020

General comments

This paper has an interesting premise and some potentially significant results, but substantial analysis and revision is required before it can be published. There have been very few LES studies conducted on the ice shelf-ocean boundary layer (IOBL) and I have yet to see one with refreezing included, which makes this work of potentially great interest. However, I am concerned that the set up, validation and analysis of the LES needs considerably more work. I have outlined my major concerns below in the specific comments section, and I would like to see these comments thoroughly addressed.

[Figure]

Specific comments

1. I have concerns about the set up of the LES. The authors describe that the inlet plume boundary condition beneath the ice shelf (and the initial conditions across the whole domain) are set from in situ observations. The in situ observations are measured at the front of the ice shelf, but the inlet plume boundary condition is much further beneath the ice shelf (1000m or so horizontally). According to the authors and the premise of the paper, the IOBL undergoes significant brine rejection, latent heat release and mixing due to refreezing at the ice edge. The resulting profiles of temperature, salinity and velocity in the open ocean are then compared against the same in situ measurements used to force the simulation, to conclude that the refreezing effect is significant.

It does not seem appropriate to force the model with the in situ field measurements (with the inlet forcing at a different region from where the field measurements were taken) and then validate the model against the same in situ measurements (but this time using model results in front of the ice shelf, similar to where the field measurements would have been taken). Perhaps I am missing something in the set up here?

The primary difference between the refreezing and no-refreezing LES depth profiles in front of the ice shelf is then found to be the increase in temperature in the top 400m (Figure 6). I agree that the refreezing case looks closer to the field measurements than the no-refreezing case. But I also question that the initial conditions, and perhaps more importantly the inlet conditions on the plume, have a temperature profile that has values smaller than the average measurement profiles (in particular at 400-550m depth on Figure 6). The authors need to further justify why they have chosen these initial profiles (dashed lines on Figure 6), and any differences that have been made from the field measurements. It is also not clear to me that the solution they have found is truly unique, in that the inlet temperature profile could potentially be tuned to find that the no-refreezing case also gives a good match with the field measurements. I would appreciate a more transparent explanation of the LES set up, including a thorough

discussion on the scientific premise going into the set up.

2. I would like to see further validation of the LES in terms of whether it is resolving the dynamics. The authors currently have only one grid resolution here, so further justification to show why they would expect convergence with a higher grid resolution would be appreciated. The LES resolution validation in the paper was solely focused on the energy spectrum. While it is reassuring to see the -5/3 spectrum, there are other variables that really should be considered, in particular the stress, heat and salt flux. Figure 10 already shows the momentum and heat fluxes with the resolved and SGS terms. The SGS terms appear quite small throughout the whole depth, which is a good sign for LES convergence, but I would like to see this compared against the total values for each depth, along with some discussion. Does the salt flux also look resolved throughout most of the depth (especially near the ice)?

3. There are some really interesting dynamics in the LES output but I am struggling to understand them and put them into context of past work. For example, the eddy at the ice front is very intriguing but there is not much information on its dynamics. I did not know whether the authors were referring to a vertical overturning type eddy, or a horizontal eddy. If the latter, is it a baroclinic eddy? What is setting the size of this eddy (e.g. the domain size or a Rossby length scale)? Is the potential vorticity in the under-ice region important in constraining where the eddy forms?

Other dynamics that come to mind are the brine rejection from refreezing. Is the brine rejection enough to generate convective plumes and a convective region? Is this the cause of the mixing in the refreezing zone? I cannot see any brine rejection influence on the salinity field (Figure 3) but this might be hidden by the colormap scale. Is there a density inversion here? Also are there any double-diffusive effects expected in the water column?

What are the effects of the Coriolis parameter in these simulations? Is the horizontal movement of the plume affected by geostrophic balance (is there a strong velocity

in the y-direction)? Would we expect an Ekman layer to form near the ice base e.g. Jenkins 2016 ("A Simple Model of the Ice Shelf–Ocean Boundary Layer and Current")?

In one of the few previous studies on the exit of meltwater plumes from under ice shelves, Naveira Garabato et al. 2017 ("Vigorous lateral export of the meltwater outflow from beneath an Antarctic ice shelf") concluded that centrifugal overturning instability played an important role in setting the mixing of the meltwater plume. Are the present LES similar to these at all?

The paper would be really strengthened by discussing the dynamics in the context of past work. In the introduction in general, I would recommend discussing some more field observations (e.g. Larsen C by Davis and Nicholls 2019, Ronne ice shelf by Jenkins et al. 2010, Fimbul ice shelf by Hattermann et al. 2012, Ross ice shelf by Arzeno et al. 2014) to really put the Nansen Ice Shelf observations into context. Similarly, I think it would be worthwhile citing some more process-based studies here also, especially ones involving a meltwater plume (e.g. McConnochie and Kerr 2018; Mondal et al. 2019).

4. The ice-ocean boundary condition for refreezing is the commonly used "three-equation" model, but how appropriate is this model for the refreezing case (in particular the frazil ice case)? The turbulent exchange velocity of heat is set to be 10ˆ-4 m/s (please provide a citation for this value) but this could be influenced in some way by the refreezing dynamics, including the brine rejection or latent heat release. This three-equation model is parameterising the heat and salt fluxes mixed towards the base of the ice, so I think more discussion should be included to justify the choice of turbulent exchange coefficients. This is a more minor comment, but could the authors define the friction velocity where it first appears (Eq. 14) and also explain how it is calculated (e.g. is a drag coefficient set?).

Li 26: please provide a citation for the line "…the ice shelf–ocean interaction in cavities is the dominant driving force for ice thinning"

[Figure]

Li 87: I would also appreciate more discussion of the type of LES used here, and why this type of LES SGS parameterisation would work well in this IOBL system. Has PALM has been used in previous studies of ice-ocean flow?

Li 90: Please describe the turbulence closure scheme used here in more detail.

Eqs. (1-9): Please ensure that all variables are defined when first introduced (e.g. u_g, Rd, Cp not defined here). Also try to include the values of any constants (e.g. what value of Coriolis parameter f is used?).

Eq. (6) and Li 113: are K_m and nu_T the same (they are defined similarly)?

Li 113: citation for C_m= 0.1 and more explanation needed for the definition of l (where does 1.8z come from for instance?).

Is Eq. (13) just the rearrangement of Eqs. (10-12)? If not, where does it come from?

Li 129: please include a citation for the turbulent exchange velocity of heat.

Eqs (14-16): please define friction velocity u* when it first appears. Also is tau the boundary stress?

Li 141: Please justify the surface roughness value of 0.07m. Why was this chosen?

Li 143: How was the ice shelf modeled in the simulations? Was it immersed boundary, interface condition, etc?

Li 150: How was the U_top value chosen? Is it based off observations? If so, please cite.

Li 174: the mean velocity of 0.0729m/s – where did this value come from? Also what is gained by putting the time in terms of the overturning time here?

Li 174: I would like more discussion of the fluctuations – it is mentioned that they are because of large-scale eddies beneath the ice shelf, but it would be nice to see what these eddies look a like a little more in terms of velocity, etc.

Li 177: "High momentum exchange by refreezing and its brine rejection..." please explain the physical processes here a little further. Why does refreezing mean high momentum exchange?

Li 178: The time averaged results of 1t* - why not use a longer time-averaging interval? Especially as the fluctuations appear to be on a slightly longer timescale (Figure 2).

Figure 3 and onward: I have some confusion about zonal and meridional velocity here. For some reason I kept thinking that zonal was the y direction, but actually it was the x direction? Please define the velocity directions in terms of x and y, at least initially. This would help me out a lot!

Figure 3: what are the undulations on the plume interface (top and bottom) on the far left hand side of the domain? Why do they form so strongly near the inlet condition? Are these the eddies referred to in Li 174?

Li 181: What is the swirling strength criterion? Please include a one-line explanation.

Li 182: "Due to neutral buoyancy, sub-ice shelf plume is about 100 m apart from the ice shelf and has a high velocity" This information is important and should come much sooner in the paper.

Li 192: "... where salinity stratification is not formed..." this is tricky to make out on Figure 3 colourmap. Perhaps think of using a different colormap here?

Li 194: "This demonstrates that the stratification is more dominant than flow shear near the ice front and play a major role in preventing flow advection from subice shelf plume." Is this really shown here? Please explain more about what is meant by the role of "stratification" here?

Figure 6: Where were the vertical profiles taken in the LES?

Li 226: "However, the multi-layered stratified characteristics of the salinity profile at depth of ice shelf bottom and IOBL top are observed in the case with refreezing effect."

What is "multi-layered stratified characteristics" referring to exactly?

Li 229: "However, it should be noted that, since this is an idealized model, some differences can be expected between the simulated results and observations, in terms of ice shelf bathymetry and surface roughness, the temporal variability of the sub-ice water plume, the drag effect of frazil ice, etc." Please discuss these further. E.g. what effects would each of these processes potentially have on the LES?

Li 245: "Additionally, meridional direction-stretched structures are observed at the interface between the inner and outer regions." Are these referring to the domain-sized undulations on Figure 7?

Figure 8: Is this the vertical heat flux? Similarly in Li 251, "high negative heat flux" is this referring to the vertical or horizontal heat flux?

Around Li 260: So is the supercooled water mixed up from the plume below, or horizontally (by eddy) from waters outside the ice shelf?

Li 263: "Figure 10 shows the vertical profiles of momentum and heat fluxes within the IOBL. As shown in figure, the depth of the IOBL top (438 m) is determined to be the depth where the magnitude of the heat flux is 1% of the maximum heat flux induced by the sub-ice shelf plume." Is this for the inner or outer region?

Li 265: "In the vertical momentum flux in the inner region, negative flux induced by refreezing and stratification is observed, showing that the IOBL flow in the inner region is in a stable condition." I thought that there was little to no refreezing in the inner region?

Li 267: "However, positive flux with large-scale advection (IOBL scale) induced by the ice front eddy is observed in the outer region, showing that the IOBL flow in the outer region is in an unstable condition" What about the top 20m where the flow appears to have negative momentum flux?

The flatness factor is of some interest, but I would be more interested to see the energy

budget of the simulations (e.g. turbulent kinetic energy, buoyancy production term, etc). Have the authors thought about calculating the energy budget?

First paragraph of the discussion. This paragraph is a nice description of the overall flow – it might be helpful to have this earlier, in the introduction or the simulation set up.

Conclusions: This should be more to the point, with a succinct summary of the main findings of the paper.

Figures: please consider using different colourmaps for each of the velocity, temperature and salinity figures. It might be easier to follow and make comparisons.

Technical corrections

Li 32: type "groundling" should be "grounding". Also "...dense and salty water melts the ice..."

Li 34: "tidal pumping melts" – what process is this referring to?

Li 39: Please rephrase the sentence "Even iceberg calving ..." it does not make sense as it stands.

Li 51: "... preventing the heat entrainment..." Please be clear what heat entrainment is being referred to here.

Li 64: Gayen et al. 2016 used DNS not LES.

Li 105: bracket missing in the fourth term on the RHS.

Li 185: "... with negative mean velocity..." mean velocity in which direction?

If possible, try not to start all the paragraphs with "Figure x ..." it is a little clunky.

---

## Author Comment (AC1) · 6 Nov 2020

Dear Referee,

We really appreciate the reviewer's valuable comments on the improvement of model validity and manuscript. We upload the author's response for Referee comment (RC1) as a supplement.

Best regards, Jisung Na.

Please also note the supplement to this comment:

[Figure]

https://tc.copernicus.org/preprints/tc-2020-166/tc-2020-166-AC1-supplement.pdf

---

## Author Comment (AC2) · 6 Nov 2020

Response to comments

We wish to thank anonymous reviewer for their valuable comments, which have helped us to improve our manuscript. We addressed each of the comments in turn below. Our responses are colored by green.

\* New simulation summary

We really appreciate the reviewer's valuable and helpful comments on the improvement of model validity. We performed a suite of new simulations with proper forcing based on the reviewer's comments and physical oceanography of Nansen ice shelf. Before response to each of the comments, we would like to address major changes and summary of new simulation set-up for model validation with proper forcing based on in-situ observations.

[Figure]

**R-Figure 1. Schematic diagram of physical processes within model domain based on in-situ CTD, LADCP, AWS (in ice breaker Araon) observations. It represents our conclusion inferred from observational and experimental evidences in revised manuscript.**

To answer the first comment of the first reviewer, we described the physical oceanographic processes and our finding in Nansen ice shelf using schematic diagram (R-Figure 1)

Through observing negative velocity near sea surface and positive velocity at sub-ice shelf plume in our LADCP observations, we define the "Ice Front Circulation" (In this study, we define this circulation de novo). Shear by momentum of sub-ice shelf plume and salt flux by frazil ice

formation (~1 cm day$^{-1}$, it is determined by sensible heat (164.8W m$^{-2}$) based on air temperature (–7.76 °C) and wind velocity (16.23 m s$^{-1}$) obtained by AWS in ice breaker Araon) could trigger this circulation. The relationship between sensible heat and frazil ice production (frazil ice production = 0.1785 x $Q_s$ – 28.048) is referred from Thompson et al. (2020). This circulation pushes the sub-ice shelf plume, making that stratification line is moved to 400 m depth. Beneath the ice shelf, basal ice melting occur because sub-ice shelf plume has a warmer temperature (–2.06 °C) than local freezing temperature (–2.115 °C) (Note that the freezing temperature was set too high as –1.92 °C in previous simulation. New simulations show not refreezing but melting and it will be described in revised manuscript.). Newly generated meltwater by basal melting is located between ice shelf bottom and stratified sub-ice shelf plume. By density difference between new meltwater and sub-ice shelf plume, shear instability (e.g. Kelvin-Helmholtz instability) occurs at this depth range (from 280 m to 300 m depth).

To prove this hypothesis, we set the latent heat and salt fluxes (corresponding to 1 cm day$^{-1}$) at top boundary and velocity of sub-ice shelf plume (similar with LADCP) at inlet boundary. In temperature between sub-ice shelf plume and ice shelf, –2.06 °C was set. In salinity profile, high stratification was set at 300 m with initial depth (20 m) of new meltwater. In upper region (from 0 to 280 m depth) at the open ocean, 0 m s$^{-1}$ velocity and zero velocity shear at top boundary were set to prove the development of Ice Front Circulation and its trigger mechanism.

To resolve interfacial temperature and salinity by different depths, we used the equation of $\theta_b = \lambda_1 S_b + \lambda_2 + \lambda_3 P$, instead of Eq. (12) $\theta_b = -\lambda S_b$ in previous manuscript.

To investigate the effect of grid size on model performance, we tested the three kinds of grid systems (low case – 216 (16 m) x 216 (16 m) x 108 (8 m), moderate case – 288 (12 m) x 288 (12 m) x 144 (6 m), and high case – 432 (8 m) x 432 (8 m) x 216 (4 m), respectively. In this grid sensitivity study, we conclude that grid resolution in moderate case is enough for resolving this oceanic flow and IOBL.

[Figure]

**R-Figure 2. xz plane contours of time averaged variables of velocity (upper), potential temperature (middle) and salinity (lower) in new simulation (moderate case).**

[Figure]

**R-Figure 3. Vertical profiles of velocity, potential temperature and salinity in new simulation.**

Also, we summarized important parameters and constants for basal ice melting at ice shelf bottom and frazil ice formation at sea surface (R-Table 1).

R-Table 1. List of model parameters and constants

| $\lambda_1$ | Freezing temperature salinity coefficient | −0.0573 | °C kg g$^{-1}$ |
|---|---|---|---|
| $\lambda_2$ | Freezing temperature constant | 0.0832 | °C |
| $\lambda_3$ | Freezing temperature depth coefficient | −7.53 x 10$^4$ | °C m$^{-1}$ |
| $\Gamma_S$ | Salt turbulent exchange coefficient (sea surface, ice shelf bottom) | 2 x 10$^{-4}$ [a], 2.6 x 10$^{-4}$ [b] | - |

| | | | |
|---|---|---|---|
| $\Gamma_\theta$ | Heat turbulent exchange coefficient (sea surface, ice shelf bottom) | $5.8 \times 10^{-3}$ [a], $8 \times 10^{-3}$ [b] | - |
| $c_w$ | Specific heat capacity of pure water | 3974 | J kg$^{-1}$°C$^{-1}$ |
| $L_i$ | Latent heat of fusion | $3.35 \times 10^5$ | J kg$^{-1}$ |
| $\rho_w$ | Density of water | 1028 | kg m$^{-3}$ |
| $\rho_i$ | Density of ice | 917 | kg m$^{-3}$ |
| $z_0$ | Surface roughness (sea surface, ice shelf bottom) | 0.001, 0.07 [c] | m |
| - | Ice shelf thickness | 280 [d] | m |
| $\theta_f$ | Local freezing temperature (sea surface, ice shelf bottom) | $-1.9, -2.115$ | °C |
| $\theta_a$ | Ambient temperature (sea surface, ice shelf bottom) | $-1.9, -2.06$ | °C |
| $\theta_b$ | Interfacial temperature (sea surface, ice shelf bottom) | $-1.885, -2.092$ | °C |
| $Sa_a$ | Ambient salinity (sea surface & ice shelf bottom) | 34.69 | psu |
| $Sa_b$ | Interfacial salinity (sea surface, ice shelf bottom) | 34.864, 34.286 | psu |
| $k_\theta$ | molecular diffusivities of heat | $1.3 \times 10^{-7}$ | m$^2$ s$^{-1}$ |
| $k_S$ | molecular diffusivities of salt | $7.2 \times 10^{-10}$ | m$^2$ s$^{-1}$ |

a - Heorton et al. (2017)

b - based on friction velocity (0.145 m s$^{-1}$) and thermal driving in new simulation (refer to Vreugdenhil and Taylor (2019))

c - high drag coefficient ($C_d = 0.01$) of cold water cavity in Gwyther et al. (2015)

d - Stevens et al. (2017)
* * *
Anonymous Referee #2

- General comments

This paper has an interesting premise and some potentially significant results, but substantial analysis and revision is required before it can be published. There have been very few LES studies conducted on the ice shelf-ocean boundary layer (IOBL) and I have yet to see one with refreezing included, which makes this work of potentially great interest. However, I am concerned that the set up, validation and analysis of the LES needs considerably more work. I have outlined my major concerns below in the specific comments section, and I would like to see these comments thoroughly addressed.

- Specific comments

1. I have concerns about the set up of the LES. The authors describe that the inlet plume boundary condition beneath the ice shelf (and the initial conditions across the whole domain) are set from in situ observations. The in situ observations are measured at the front of the ice shelf, but the inlet plume boundary condition is much further beneath the ice shelf (1000m or so horizontally). According to the authors and the premise of the paper, the IOBL undergoes significant brine rejection, latent heat release and mixing due to refreezing at the ice edge. The resulting profiles of temperature, salinity and velocity in the open ocean are then compared against the same in situ

measurements used to force the simulation, to conclude that the refreezing effect is significant. It does not seem appropriate to force the model with the in situ field measurements (with the inlet forcing at a different region from where the field measurements were taken) and then validate the model against the same in situ measurements (but this time using model results in front of the ice shelf, similar to where the field measurements would have been taken). Perhaps I am missing something in the set up here?

The primary difference between the refreezing and no-refreezing LES depth profiles in front of the ice shelf is then found to be the increase in temperature in the top 400m (Figure 6). I agree that the refreezing case looks closer to the field measurements than the no-refreezing case. But I also question that the initial conditions, and perhaps more importantly the inlet conditions on the plume, have a temperature profile that has values smaller than the average measurement profiles (in particular at 400-550m depth on Figure 6). The authors need to further justify why they have chosen these initial profiles (dashed lines on Figure 6), and any differences that have been made from the field measurements. It is also not clear to me that the solution they have found is truly unique, in that the inlet temperature profile could potentially be tuned to find that the no-refreezing case also gives a good match with the field measurements. I would appreciate a more transparent explanation of the LES set up, including a thorough discussion on the scientific premise going into the set up.

- As the reviewer mentioned, we concluded that the validation and LES setup with forcing & boundary conditions had some issues. Therefore, we composed new simulation set-up with different boundary conditions based on our hypothesis (R-Figure 1), using only the properties of sub-ice shelf plume. In new simulation, initial and boundary conditions were composed by our hypothesis, not from the in-situ observation.
* * *
2. I would like to see further validation of the LES in terms of whether it is resolving the dynamics. The authors currently have only one grid resolution here, so further justification to show why they would expect convergence with a higher grid resolution would be appreciated. The LES resolution validation in the paper was solely focused on the energy spectrum. While it is reassuring to see the -5/3 spectrum, there are other variables that really should be considered, in particular the stress, heat and salt flux. Figure 10 already shows the momentum and heat fluxes with the resolved and SGS terms. The SGS terms appear quite small throughout the whole depth, which is a good sign for LES convergence, but I would like to see this compared against the total values for each depth, along with some discussion. Does the salt flux also look resolved throughout most of the depth (especially near the ice)?

- As the reviewer mentioned, we have tried to validate our grid resolution is proper in this oceanic flow through energy spectrum of $u$ velocity and resolved & SGS flux of heat and momentum. As shown in R-Figure 4, we will plot the $u$ and $w$ velocity which are related with shear and buoyancy effects. Also, we will plot the momentum, heat and salt fluxes. As the reviewer mentioned, SGS fluxes near the ice have relatively large values. It can be solved using stretched-vertical grid, but

this has a problem for employing in this study, because simulation domain in this study have an ice shelf geometry. We will discuss the impact of relatively large SGS fluxes near the ice surface in discussion section in revised manuscript.

[Figure]

**R-Figure 4. 1D energy spectra of $u$ and $w$ velocity in new simulation results.**
* * *
3. There are some really interesting dynamics in the LES output but I am struggling to understand them and put them into context of past work. For example, the eddy at the ice front is very intriguing but there is not much information on its dynamics. I did not know whether the authors were referring to a vertical overturning type eddy, or a horizontal eddy. If the latter, is it a baroclinic eddy? What is setting the size of this eddy (e.g. the domain size or a Rossby length scale)? Is the potential vorticity in the under-ice region important in constraining where the eddy forms?

- Instead of ice front eddy, we propose the ice front circulation in open ocean based on new simulation results. This circulation is vertical overturning type and has baroclinic eddy characteristics. We will evaluate a budget and balance through comparison between buoyancy effect and shear effect by sub-ice shelf plume in revised manuscript.

Other dynamics that come to mind are the brine rejection from refreezing. Is the brine rejection enough to generate convective plumes and a convective region? Is this the cause of the mixing in

the refreezing zone? I cannot see any brine rejection influence on the salinity field (Figure 3) but this might be hidden by the colormap scale. Is there a density inversion here? Also are there any double-diffusive effects expected in the water column?

- In new simulation, we can observe the density inversion near the sea surface (R-Figure 2, 3). But, double-diffusive effects was not observed.

What are the effects of the Coriolis parameter in these simulations? Is the horizontal movement of the plume affected by geostrophic balance (is there a strong velocity in the y-direction)? Would we expect an Ekman layer to form near the ice base e.g. Jenkins 2016 ("A Simple Model of the Ice Shelf–Ocean Boundary Layer and Current")?

- Within depth range (from 280 m to 400 m depth where $u$ velocity gradient is large), weak $v$ velocity was observed and structures of Ekman spiral layer was unclear.

[Figure]

R-Figure 5. Vertical profiles of velocities beneath the ice shelf.

In one of the few previous studies on the exit of meltwater plumes from under ice shelves, Naveira Garabato et al. 2017 ("Vigorous lateral export of the meltwater outflow from beneath an Antarctic ice shelf") concluded that centrifugal overturning instability played an important role in setting the mixing of the meltwater plume. Are the present LES similar to these at all?

- In new simulation results, we had come to a similar conclusion of study of Naveira Garabato et al. 2017. We focused on vertical distribution, whereas they focused on kinetic energy extraction from lateral shear. We will discuss this in revised manuscript.

The paper would be really strengthened by discussing the dynamics in the context of past work. In the introduction in general, I would recommend discussing some more field observations (e.g. Larsen C by Davis and Nicholls 2019, Ronne ice shelf by Jenkins et al. 2010, Fimbul ice shelf by Hattermann et al. 2012, Ross ice shelf by Arzeno et al. 2014) to really put the Nansen Ice Shelf observations into context. Similarly, I think it would be worthwhile citing some more process-based studies here also, especially ones involving a meltwater plume (e.g. McConnochie and Kerr 2018; Mondal et al. 2019).

- As the reviewer mentioned, it needs to be more strengthened by discussing the other cold water cavity such as Larsen C, Ronne ice shelf, Ross ice shelf. We will re-organize the literature survey in introduction section, adding those studies in literature survey.

4. The ice-ocean boundary condition for refreezing is the commonly used "three-equation" model, but how appropriate is this model for the refreezing case (in particular the frazil ice case)? The turbulent exchange velocity of heat is set to be 10^-4 m/s (please provide a citation for this value) but this could be influenced in some way by the refreezing dynamics, including the brine rejection or latent heat release. This three equation model is parameterising the heat and salt fluxes mixed towards the base of the ice, so I think more discussion should be included to justify the choice of turbulent exchange coefficients. This is a more minor comment, but could the authors define the friction velocity where it first appears (Eq. 14) and also explain how it is calculated (e.g. is a drag coefficient set?).

- When we performed the new simulation, we considered heat and salt exchange coefficients based on Vreugdenhil and Taylor (2019). Although it has a different friction velocity range, we can interpolate these values using physical relationship. We will discuss this in discussion section in revised manuscript. And, we will also explain the definition of friction velocity.

Li 26: please provide a citation for the line ". . .the ice shelf–ocean interaction in cavities is the dominant driving force for ice thinning"

- This sentence will be change to "the ice shelf-ocean interaction in cavities is one of the driving force for ice shelf instability" with reference.

Li 87: I would also appreciate more discussion of the type of LES used here, and why this type of LES SGS parameterisation would work well in this IOBL system. Has PALM has been used in previous studies of ice-ocean flow?

- In this study, 1.5 order Deardorff SGS model was employed for LES SGS parametrization. In this model, averaged features of IOBL can be resolved, because SGS stress is proportional to local gradient of mean quantity. It can have a huge difference in simulation results as we simulate the

flow with high temporal-variability of local gradient, having a various mixing length. We think that this is the first study for ice-ocean flow using PALM. We will discuss this in discussion section in revised manuscript.
* * *
Li 90: Please describe the turbulence closure scheme used here in more detail. Eqs. (1-9): Please ensure that all variables are defined when first introduced (e.g. u_g, Rd, Cp not defined here). Also try to include the values of any constants (e.g. what value of Coriolis parameter f is used?).

- Description of some parameters were missing. We will add the explanation of turbulence closure and parameters. Coriolis parameter is -1.41 x10$^{-4}$ which is corresponding to 75 S.
* * *
Eq. (6) and Li 113: are K_m and nu_T the same (they are defined similarly)?

- Those parameters are same. We will remove the nu_T in equations.
* * *
Li 113: citation for C_m= 0.1 and more explanation needed for the definition of l (where does 1.8z come from for instance?). Is Eq. (13) just the rearrangement of Eqs. (10-12)? If not, where does it come from?

- C_m is important factor for determining the eddy viscosity. We refer the Maronga et al. (2015). This mixing length is from Stable Boundary Lalyer study of Saiki et al. (2000). Eq. (13) is the solution of rearrangement equation of Eqs. (10-12). We will add this information to methodology section in revised manuscript.
* * *
Li 129: please include a citation for the turbulent exchange velocity of heat.

- We list-up these coefficients of heat and salt in R-Table 1.
* * *
Eqs (14-16): please define friction velocity u* when it first appears. Also is tau the boundary stress?

Li 141: Please justify the surface roughness value of 0.07m. Why was this chosen?

- As shown in R-Table 1, we set the surface roughness, referring the high drag coefficient, $C_d$=0.01 case in Gwether et al. (2015).
* * *
Li 143: How was the ice shelf modeled in the simulations? Was it immersed boundary, interface

condition, etc?

- We modeled ice shelf using mask method of Briscolini and Santangelo (1989) (fixed topography). We will explain this in methodology section in revised manuscript.
* * *
Li 150: How was the U_top value chosen? Is it based off observations? If so, please cite.

- In new simulation set-up, we set the 0 m s$^{-1}$ at the top boundary to exclude the wind effect.
* * *
Li 174: the mean velocity of 0.0729m/s – where did this value come from? Also what is gained by putting the time in terms of the overturning time here?

- In previous simulation, we used the mean velocity and total domain scale to obtain the large eddy turnover time (t$^*$). Since it was not IOBL property, we used the friction velocity and initial IOBL depth to obtain the large eddy turnover time in IOBL. In new simulation, t$^*$ is 3.83 hours which is determined by 0.145 m s$^{-1}$ and 20 m depth of new meltwater (R-Figure 6)

[Figure]

**R-Figure 6. Time series of friction velocity in new simulations with different grid resolutions.**
* * *
Li 174: I would like more discussion of the fluctuations – it is mentioned that they are because of large-scale eddies beneath the ice shelf, but it would be nice to see what these eddies look a like a little more in terms of velocity, etc.

- In results section of new simulation, we will discuss velocity and its shear beneath the ice shelf

to observe the shear effect in IOBL.
* * *
Li 177: "High momentum exchange by refreezing and its brine rejection. . . " please explain the physical processes here a little further. Why does refreezing mean high momentum exchange?

- In previous simulation, de-stratification by brine rejection induced the vertical fluxes of momentum and entrainment of momentum.
* * *
Li 178: The time averaged results of 1t* - why not use a longer time-averaging interval? Especially as the fluctuations appear to be on a slightly longer timescale (Figure 2).

- In new simulation, we used the last 3 t* for averaged results after 17 t* (R-Figure 6)
* * *
Figure 3 and onward: I have some confusion about zonal and meridional velocity here. For some reason I kept thinking that zonal was the y direction, but actually it was the x direction? Please define the velocity directions in terms of x and y, at least initially. This would help me out a lot!

- We will add the directions with boundary conditions in revised manuscript (R-Figure 7)

[Figure]

**R-Figure 7. Domain information and boundary conditions**

Figure 3: what are the undulations on the plume interface (top and bottom) on the far left hand side of the domain? Why do they form so strongly near the inlet condition? Are these the eddies referred to in Li 174?

- New simulation had also an undulations near the inlet conditions, showing the IOBL flow had the wave-like features (Kelvin-Helmholtz instability). In previous simulation, we did not consider the instability, but we will add the part for Kelvin-Helmholtz instability development which is linear stability analysis (Na et al. (2014)).

Li 181: What is the swirling strength criterion? Please include a one-line explanation.

- This method is for separating the swirling component of vorticity equation to identify the vortex structures. We will add a one-line explanation with reference in revised manuscript.

Li 182: "Due to neutral buoyancy, sub-ice shelf plume is about 100 m apart from the ice shelf and has a high velocity" This information is important and should come much sooner in the paper.

- In revised manuscript, we will add an explanation for this neutral buoyancy part to methodology section.

Li 192: "... where salinity stratification is not formed..." this is tricky to make out on Figure 3 colourmap. Perhaps think of using a different colormap here?

- We will try the different colormap, and choose proper colormap to examine those features obviously.

Li 194: "This demonstrates that the stratification is more dominant than flow shear near the ice front and play a major role in preventing flow advection from subice shelf plume." Is this really shown here? Please explain more about what is meant by the role of "stratification" here?

In new simulation, we cannot observe the stratification is dominant than flow shear near the ice front. Instead of this, we will discuss the analysis of comparison between shear and stratification to generate the Kelvin-Helmholtz instability in IOBL.

Figure 6: Where were the vertical profiles taken in the LES?

- We obtained the profiles at 1 km apart from ice front in both previous and new simulations.

Li 226: "However, the multi-layered stratified characteristics of the salinity profile at depth of ice shelf bottom and IOBL top are observed in the case with refreezing effect." What is "multi-layered stratified characteristics" referring to exactly?

- In previous simulation, we referred the multi-stratification layer at near IOBL top (400 m depth) in vertical profiles.

Li 229: "However, it should be noted that, since this is an idealized model, some differences can be expected between the simulated results and observations, in terms of ice shelf bathymetry and surface roughness, the temporal variability of the sub-ice water plume, the drag effect of frazil ice, etc." Please discuss these further. E.g. what effects would each of these processes potentially have on the LES?

- Ice shelf bathymetry (e.g. local slope of ice shelf) and surface roughness can affect the structure of mean velocity and turbulence intensity. Temporal variability can be expected the change in turbulence properties such as intensity, shear production. We will discuss this in discussion section in revised manuscript.

Li 245: "Additionally, meridional direction-stretched structures are observed at the interface between the inner and outer regions." Are these referring to the domain-sized undulations on Figure 7?

- In previous simulation, we referred the stretched eddy at interface between inner and outer regions.

Figure 8: Is this the vertical heat flux? Similarly in Li 251, "high negative heat flux" is this referring to the vertical or horizontal heat flux?

- Yes. It was the vertical heat flux. But, it is not related with melting (or refreezing) pattern. In revised manuscript, we will consider the integral of vertical flux (or entrainment quantity) to observe the relationship between melting pattern and fluxes.

Around Li 260: So is the supercooled water mixed up from the plume below, or horizontally (by eddy) from waters outside the ice shelf?

- In this part, we referred the super-cooled water was mixed up and advected to upper region.

Li 263: "Figure 10 shows the vertical profiles of momentum and heat fluxes within the IOBL. As shown in figure, the depth of the IOBL top (438 m) is determined to be the depth where the magnitude of the heat flux is 1% of the maximum heat flux induced by the sub-ice shelf plume." Is this for the inner or outer region?

- We referred IOBL top in the inner region (438 m). IOBL top of outer region is 447 m. In new simulation, there was another criterion for IOBL top using momentum flux, because it is shear-dominant flow.
* * *
Li 265: "In the vertical momentum flux in the inner region, negative flux induced by refreezing and stratification is observed, showing that the IOBL flow in the inner region is in a stable condition." I thought that there was little to no refreezing in the inner region?

- In previous simulation, there was little refreezing in the inner region, having a stable condition of IOBL flow, as the reviewer mentioned.
* * *
Li 267: "However, positive flux with large-scale advection (IOBL scale) induced by the ice front eddy is observed in the outer region, showing that the IOBL flow in the outer region is in an unstable condition" What about the top 20m where the flow appears to have negative momentum flux?

- In previous simulation, negative momentum flux at 20 m depth are induced by large refreezing rate and ice front eddy.
* * *
The flatness factor is of some interest, but I would be more interested to see the energy budget of the simulations (e.g. turbulent kinetic energy, buoyancy production term, etc). Have the authors thought about calculating the energy budget?

- We also have a vertical profiles of energy budget at each depth. If it is proper for the explanation our hypothesis, we will use the energy budget analysis, comparing this with fluxes or other turbulent properties.
* * *
First paragraph of the discussion. This paragraph is a nice description of the overall flow – it might be helpful to have this earlier, in the introduction or the simulation set up.

- In revised manuscript, we will re-organize the paragraph location to highlight our hypothesis and findings.
* * *
Conclusions: This should be more to the point, with a succinct summary of the main findings of

the paper.

- Using the schematic diagram, we will specify the conclusions for our study.
* * *
Figures: please consider using different colourmaps for each of the velocity, temperature and salinity figures. It might be easier to follow and make comparisons.

- We will re-plot the contours using another colourmap in revised manuscript.
* * *
- Technical corrections

Li 32: type "groundling" should be "grounding". Also ". . .dense and salty water melts the ice. . . "

- It will be corrected as the reviewer mentioned.
* * *
Li 34: "tidal pumping melts" – what process is this referring to?

- In this sentence, we referred to increment of melt rate by tidal shear.
* * *
Li 39: Please rephrase the sentence "Even iceberg calving . . ." it does not make sense as it stands.

- We will rephrase this sentence in revised manuscript.
* * *
Li 51: ". . . preventing the heat entrainment. . ." Please be clear what heat entrainment is being referred to here.

- "Stratification by a density gradient of meltwater near ice surface prevents the heat entrainment"
* * *
Li 64: Gayen et al. 2016 used DNS not LES.

- It will be corrected as the reviewer mentioned.
* * *
Li 105: bracket missing in the fourth term on the RHS.

- It will be corrected as the reviewer mentioned.
* * *
Li 185: ". . . with negative mean velocity. . ." mean velocity in which direction?

- Zonal direction. We will clarify velocity direction in revised manuscript.
* * *
If possible, try not to start all the paragraphs with "Figure x . . ." it is a little clunky.

- We will amend this in revised manuscript.
* * *
Reference

Thompson, L., Smith, M., Thomson, J., Stammerjohn, S., Ackley, S., & Loose, B. (2020). Frazil ice growth and production during katabatic wind events in the Ross Sea, Antarctica. The Cryosphere, 14(10), 3329-3347.

Heorton, H. D., Radia, N., & Feltham, D. L. (2017). A model of sea ice formation in leads and polynyas. Journal of Physical Oceanography, 47(7), 1701-1718.

Vreugdenhil, C. A., & Taylor, J. R. (2019). Stratification effects in the turbulent boundary layer beneath a melting ice shelf: Insights from resolved large-eddy simulations. Journal of Physical Oceanography, 49(7), 1905-1925.

Gwyther, D. E., Galton-Fenzi, B. K., Dinniman, M. S., Roberts, J. L., & Hunter, J. R. (2015). The effect of basal friction on melting and freezing in ice shelf–ocean models. Ocean Modelling, 95, 38-52.

Stevens, C., Lee, W. S., Fusco, G., Yun, S., Grant, B., Robinson, N., & Hwang, C. Y. (2017). The influence of the Drygalski Ice Tongue on the local ocean. Annals of Glaciology, 58(74), 51-59.

Yoon, S. T., Lee, W. S., Stevens, C., Jendersie, S., Nam, S., Yun, S., ... & Lee, J. (2020). Variability in high-salinity shelf water production in the Terra Nova Bay polynya, Antarctica.

Maronga, B., Gryschka, M., Heinze, R., Hoffmann, F., Kanani-Sühring, F., Keck, M., ... & Raasch, S. (2015). The Parallelized Large-Eddy Simulation Model (PALM) version 4.0 for atmospheric and oceanic flows: model formulation, recent developments, and future perspectives. Geoscientific Model Development Discussions 8 (2015), Nr. 2, S. 1539-1637.

Saiki, E. M., Moeng, C. H., & Sullivan, P. P. (2000). Large-eddy simulation of the stably stratified planetary boundary layer. Boundary-Layer Meteorology, 95(1), 1-30.

Briscolini, M., & Santangelo, P. (1989). Development of the mask method for incompressible unsteady flows. Journal of Computational Physics, 84(1), 57-75.

Na, J. S., Jin, E. K., & Lee, J. S. (2014). Investigation of Kelvin–Helmholtz instability in the stable boundary layer using large eddy simulation. Journal of Geophysical Research: Atmospheres, 119(13), 7876-7888.

---

## Author Response (AR1)

Response to comments

We wish to thank anonymous reviewer for their valuable comments, which will help us to improve our manuscript. We addressed each of the comments in turn below. Our responses are colored by green.

* New simulation summary

We really appreciate the reviewer's valuable and helpful comments on the improvement of model validity. We performed a suite of new simulations with proper forcing based on the reviewer's comments and physical oceanography of Nansen ice shelf. Before response to each of the comments, we would like to address major changes and summary of new simulation set-up for model validation with proper forcing based on in-situ observations.

[Figure]

R-Figure 1. Schedamic diagram of physical processes within model domain based on in-situ CTD, LADCP, AWS (in ice breaker Araon) observations. It represents our conclusion inferred from observational and experimental evidences in revised manuscript.

To answer the first comment of reviewer, we describe the physical oceanographic processes and our finding in Nansen ice shelf in late summer using schematic diagram (R-Figure 1)

Through observing negative velocity near sea surface and positive velocity at sub-ice shelf plume in our LADCP observations, we define the "Ice Front Circulation" (In this study, we define this circulation de novo). Shear by momentum of sub-ice shelf plume and salt flux by frazil ice

formation (~1.38 cm day$^{-1}$, it is determined by sensible heat (164.8W m$^{-2}$) based on air temperature (–7.76 °C) and wind velocity (16.23 m s$^{-1}$) obtained by AWS in ice breaker Araon) could trigger this circulation. The relationship between sensible heat and frazil ice production (frazil ice production = 0.1785 x $Q_s$ – 28.048) is referred from Thompson et al. (2020). This circulation pushes the sub-ice shelf plume, making that stratification line is moved to 350 m depth. Beneath the ice shelf, basal ice melting occurs because sub-ice shelf plume has a warmer temperature (–2.06 °C) than local freezing temperature (–2.115 °C) (Note that the freezing temperature was set too high as –1.92 °C in previous simulation. New simulations show not refreezing but melting and it will be described in revised manuscript.). Newly generated meltwater by basal melting is located between ice shelf bottom and stratified sub-ice shelf plume. By density difference between new meltwater (Positively buoyant ice shelf water, PISW in revised manuscript) and sub-ice shelf plume, high speed current in high turbulence case occurs at this depth range (from 280 m to 400 m depth).

To prove this hypothesis, we set the latent heat and salt fluxes (corresponding to 1.38 cm day$^{-1}$) at top boundary and velocity of sub-ice shelf plume (similar with LADCP) at inlet boundary. In temperature between sub-ice shelf plume and ice shelf, –2.06 °C was set. In salinity profile, high stratification was set at 280 m. In upper region (from 0 to 280 m depth) at the open ocean, 0 m s$^{-1}$ velocity and zero velocity shear at top boundary were set to prove the development of Ice Front Circulation and its trigger mechanism.

To resolve interfacial temperature and salinity by different depths, we used the equation of $\theta_b = \lambda_1 S_b + \lambda_2 + \lambda_3 P$, instead of Eq. (12) $\theta_b = -\lambda S_b$ in previous manuscript.

To investigate the effect of grid size on model performance, we tested the three kinds of grid systems (low case – 216 (16 m) x 216 (16 m) x 108 (8 m), moderate case – 288 (12 m) x 288 (12 m) x 144 (6 m), and high case – 432 (8 m) x 432 (8 m) x 216 (4 m), respectively. In this grid sensitivity study, we conclude that grid resolution in moderate case is enough for resolving this oceanic flow and IOBL.

[Figure]

**R-Figure 2. xz plane contours of time averaged variables of velocity (upper), potential temperature (middle) and salinity (lower) in new simulation (moderate case).**

[Figure]

**R-Figure 3. Vertical profiles of velocity, potential temperature and salinity in new simulation.**

Also, we summarized important parameters and constants for basal ice melting at ice shelf bottom and frazil ice formation at the sea surface (R-Table 1).

R-Table 1. List of model parameters and constants

| | | | |
|---|---|---|---|
| $\lambda_1$ | Freezing temperature salinity coefficient | −0.0573 | °C kg g$^{-1}$ |
| $\lambda_2$ | Freezing temperature constant | 0.0832 | °C |
| $\lambda_3$ | Freezing temperature depth coefficient | −7.53x10$^4$ | °C m$^{-1}$ |
| $\Gamma_S$ | Salt turbulent exchange coefficient (sea surface, ice shelf bottom) | 2x10$^{-4}$ [a], 2.6x10$^{-4}$ [b] | - |
| $\Gamma_\theta$ | Heat turbulent exchange coefficient (sea surface, ice shelf bottom) | 5.8x10$^{-3}$ [a], 8x10$^{-3}$ [b] | - |
| $c_w$ | Specific heat capacity of pure water | 3974 | J kg$^{-1}$°C$^{-1}$ |
| $L_i$ | Latent heat of fusion | 3.35x10$^5$ | J kg$^{-1}$ |
| $\rho_w$ | Density of water | 1028 | kg m$^{-3}$ |
| $\rho_i$ | Density of ice | 917 | kg m$^{-3}$ |
| $z_0$ | Surface roughness (sea surface, ice shelf bottom) | 0.001, 0.005 [c] | m |
| - | Ice shelf thickness | 280 [d] | m |
| $\theta_f$ | Local freezing temperature (sea surface, ice shelf bottom) | −1.9, −2.115 | °C |
| $\theta_a$ | Ambient temperature (sea surface, ice shelf bottom) | −1.9, −2.06 | °C |
| $\theta_b$ | Interfacial temperature (sea surface, ice shelf bottom) | −1.879, −2.092 | °C |
| $Sa_a$ | Ambient salinity (sea surface & ice shelf bottom) | 34.69 | Psu |
| $Sa_b$ | Interfacial salinity (sea surface, ice shelf bottom) | 34.9425, 34.286 | Psu |
| $k_\theta$ | molecular diffusivities of heat | 1.3x10$^{-7}$ | m$^2$ s$^{-1}$ |
| $k_S$ | molecular diffusivities of salt | 7.2x10$^{-10}$ | m$^2$ s$^{-1}$ |
| $u_*$ | Friction velocity (sea surface, ice shelf bottom) | 0.026, calculated | m s$^{-1}$ |
| | Wind speed, air temperature in AWS | 16.23, −7.76 | m s$^{-1}$, °C |
| $Q_s$ | Sensible heat flux at sea surface | 164.88[e] | W m$^{-2}$ |
| $F$ | Frazil ice formation | 1.38[e] | cm day$^{-1}$ |

a – Heorton et al. (2017)

b – based on friction velocity (0.168 m s-1) and thermal driving (refer to Vreugdenhil and Taylor (2019))

c – Smooth ice with melting case (Cd = 0.001) in Gwyther et al. (2016)

d – Stevens et al. (2017)

e – Thompson et al. (2020)
* * *
Anonymous Referee #1

- General comments

There are important gaps in our knowledge of sub-ice shelf ocean dynamics, particularly in the

freezing regime. Na et al. have set their sights high in this study, but the manuscript as submitted does not provide sufficient evidence of model validity. In fact, there are several theoretical reasons to believe that the dynamics are not adequately represented in the model. Their argument for validity chiefly rests on the match between simulated temperature and salinity profiles and observations. Since the model has observations as initial conditions and an inflow boundary condition, it is unclear how far from observations the simulations could evolve. Furthermore, their presentation of the results and discussion of the turbulent dynamics must be more thorough to substantiate many of their scientific conclusions. This study could be publishable after an expanded discussion of model limitations and scaled-back claims to model realism, more elaboration of simulation results, and an expanded discussion of existing literature on frazil ice dynamics.

- As the reviewer mentioned, this study is new approach about sub-ice shelf ocean dynamics with sub-ice shelf plume which was observed in 2016/17 in-situ shipboard using 3D large eddy simulation with thermal and salty fluxes based on Monin-Obukhov similarity. To reply to the reviewer's comment, we conducted new simulations of the IOBL and ocean dynamics under other conditions (profiles) to prove the validity of this model. In response to comments, we made some efforts to explain the validation issue and several limitations.

- ● Specific comments

The connection to existing literature is inadequate. The readers need to know the physical oceanographic context for the region to assess the strengths and shortcomings of the model and the model setup. What is driving ocean circulation in reality and in the model? We also need to know what observational estimates exist of melting and freezing rates of the Nansen Ice Shelf to determine whether the simulated freezing rates are realistic (Is Mode 3 melting present at Nansen Ice Shelf?). Regarding the result that refreezing rates are high at the ice shelf front, is there any observational basis for this pattern or are you proposing it de novo?

– As the reviewer mentioned, Mode 3 (Antarctic Surface Water, AASW) was observed in Nansen ice shelf (red circle in R-Figure 4). However, the Modified Surface Water (MSW) and sub-ice shelf plume was mainly observed in our 24 CTD observations in late summer season. Therefore, driving forces of ocean circulation in our simulation are sub-ice shelf plume (generated by Mode 1) and salt flux at top boundary in the summer season. We will add and re-organize the literature survey for physical oceanography near Nansen ice shelf in revised manuscript.

– To describe the sub-ice shelf plume, we set prescribed-profiles for velocity, temperature and salinity as the inflow, because sub-ice shelf plume is generated at outside of simulation domain. Also, non-cyclic boundary condition (Dirichlet & extrapolation boundary conditions) was used for inflow and outflow boundary owing to spatial heterogeneous due to the presence of the ice shelf. These set-up of our model has the advantage of being able to resolve the target phenomena (e.g. IOBL flow, sub-ice shelf plume dynamics and ice front circulation) under this environment. We will explain this in methodology section in revised manuscript.

– Limitations and shortcomings are absence of the slope of ice shelf and temporal variability of sub-ice shelf plume because we only observe time-averaged features without the slope of ice shelf.

This contents will be discussed in discussion section in revised manuscript.

[Figure]

**R-Figure 4. T-S diagram (Yoon et al. (2020))**
* * *
Please provide more context for the observations that are used to initialize the model. There should be a brief presentation of the water masses that are present in the water column and their flow orientation (only zonal velocities are presented). The location of the observations should be shown on the study area figure, and text should indicate their distance from the ice shelf front and the time span over which these observations were collected. It is unclear whether these observations were presented in Yoon et al. (2020) or whether they are published here for the first time. Furthermore, the apparent bimodal temperature distribution at ～500 m depth also needs to be explained so that it's clear why you try to match the low temperature cases.

– As the reviewer mentioned, the explanation of observation was insufficient. With schematic diagram of R-Figure 1, we will explain the observation information in detail in methodology section. In new simulation, we used the average value of temperature (–2.06 °C) of apparent bimodal temperature distribution to examine the averaged feature of sub-ice shelf plume. Since this temperature determines the thermal driving at ice surface, we could consider various temperatures of sub-ice shelf plume in future study.
* * *
I have philosophical concerns about the manner in which the authors validate the model. Authors argue that the model is valid because the LES results match the observations, but the model is initialized to observations and has inflow that roughly matches the observations. Therefore, the argument seems to be that the model does not drift too far from observations, which may be too

weak an argument for model validity. It is also unclear to me where in the LES domain you are evaluating the agreement with observations. There should be a more thorough discussion of model limitations and a discussion of their possible impact on simulation results. There should be an explicit argument addressing whether your LES model can capture freezing dynamics in the absence of frazil ice dynamics. Frazil ice dynamics significantly influence IOBL evolution as documented in previous literature, which also should be cited (including the work of Galton-Fenzi). The lack of ice shelf slope should also be discussed. Furthermore, the parameterization of heat, salt and momentum fluxes at the ice base were developed for the ice melting case by McPhee et al. 1987. The applicability of this parameterization as well as the gamma_T, gamma_S exchange coefficients to the freezing case needs to be discussed.

– To prove the validity of this model, we made some efforts in set-up of initial and boundary conditions. In new simulation, model was validated under complex environments (new meltwater, sub-ice shelf plume and frazil ice formation at the sea surface), in terms of driving ocean circulation, its scale and quantities & trend of variables (velocity, PT and Sa). Differ to the previous simulation, we considered ice melting dynamics beneath ice shelf in new simulation. In latent heat and salt fluxes at sea surface, we considered only total heat content and salt quantity by frazil ice formation (1 cm day$^{-1}$) without the frazil ice dynamics. Also, we imposed gamma_T, gamma_S based on high-resolution LES study of Vreugdenhil and Taylor (2019) and sea ice study of Heorton et al. (2017). These parameters and absence of frazil ice dynamics will be discussed in discussion section in revised manuscript.
* * *
How do you know that the strong refreezing anomaly at the ice front is not a numerical artifact related to the front geometry?

– In new simulation, similar trend of melt rate was observed near this region because upwelling of new meltwater and entrainment of outer ocean, as shown in R-Figure 5. We conclude that there is no numerical artifact because we have checked that momentum flux layer of Monin-Obukhov similarity is established well at node points near the edge of ice front.

[Figure]

R-Figure 5. Melt rate (m yr-1) distribution of new simulation

What physically determines the location of the transition from the inner to the outer region in terms of refreezing rate? How do you know that this transition isn't just characterized by the development of turbulence along-flow from a less-turbulent inflow? Is there any observational evidence for these trends in refreezing rates, either at this ice shelf or any ice shelf?

– Because we imposed theoretical inflow of sub-ice shelf plume (based on in-situ LADCP observation profile), the region of turbulent development is relatively small (~ 300 m). In result analysis of revised manuscript, we will exclude the region of turbulence development only to observe the fully developed features.
* * *
Line 13: "In particular, it is evident that, when the refreezing effect is considered, the IOBL flow can be more realistically resolved, especially upward advection from the sub-ice shelf plume and the ice front eddy." You don't show that the entrainment and eddying in the freezing case are more realistic, if by realistic you mean closer to observations.

– In new simulation, we observed the development of ice front circulation under environments with meltwater production and salt flux based on in-situ observation. Therefore, we conclude that this simulated oceanic flow is realistic, representing the physical oceanographic processes in Nansen ice shelf.
* * *
Lines 30-45: The modes of ice shelf melting and the classification of warm and cold water cavities should be presented in the context of your study. This paragraph feels too general and unfocussed.

- As the reviewer mentioned, we will include the modes of ice melting, the classification of warm and cold water cavities and Nansen ice shelf characteristics in introduction section in revised manuscript.
* * *
Line 41: This sentence should state that the basal melt rate of ice shelves is determined by the rate of BOTH heat and salt exchange, as salt exchange plays a key role in the dissolving regime characteristic of most ice shelf settings.

- We will amend this sentence, including the importance of dissolving regime and salt exchange.
* * *
Line 76: "we were able to account for refreezing patterns, detailed flow structures including turbulent characteristics, fluxes and the relationship between refreezing and entrainment of supercooled water from sub-ice shelf plume within the IOBL." This makes it sound as if all of those properties were observed, when in fact you don't present observations that can be linked with those characteristics.

- As the reviewer mentioned, this sentence was not clear. We will amend this sentence, emphasizing that those characteristics were obtained from validated simulation results.
* * *
Line 134: I might have missed it but I can't find the value of z_0.

- In R-Table 1, we list-up z_0 values and its reference.
* * *
Line 136: It is unclear how the no-refreezing case is configured. Are both heat and salt fluxes set to zero at the top boundary?

- In revised manuscript, we will exclude the no-refreezing case (case without heat and salt fluxes).
* * *
Line 145: Argue for the appropriateness of setting a CTD profile that was observed in the open ocean as inflow conditions under the ice shelf.

- Because sub-ice shelf plume is generated from ice melting near grounding line, we cannot resolve this in our simulation domain. Therefore, we have to use the prescribed profiles of temperature and salinity based on observations. For velocity profile in new simulation, we set theoretical profile of turbulent flow with different turbulence. In additional, we set the inflow conditions using observation under the assumption that temperature, salinity of sub-ice shelf plume in the open ocean is similar with those beneath ice shelf. We will explain this assumption and appropriateness in methodology section in revised manuscript.
* * *
Line 149: Please specify how the radiation boundary condition is implemented.

- Radiation boundary condition at outlet boundary is extrapolation boundary condition (gradient of variables are zero) that outlet boundary does not affect the oceanic flow. We will explain this in methodology section in revised manuscript.
* * *
Line 150: Further explanation is needed to address how Dirichlet conditions at top boundary are appropriate even under the ice shelf and what they represent in the open ocean. Is this consistent with observed winds?

- Through the AWS observation in Araon, observed wind direction was opposite to upper current direction (negative zonal velocity). It represents that upper current was not developed by winds. Therefore, we set the Dirichlet condition ($v = 0$) at top boundary to exclude the wind effect, because wind effect is out of scope.
* * *
Line 153: With what metric is quasi-steady state evaluated?

- We plotted time series of the friction velocity near ice shelf bottom to examine the time convergence. As shown in R-figure 6, it is observed that friction velocity is converged after 14 t$^*$. Through this feature, we evaluate the quasi-steady state in simulation result.

[Figure]

**R-Figure 6. Time series of friction velocity in new simulations.**

Results: Explain in the refreezing case to what degree the increase in boundary layer temperature is due to the release of latent heat and differences in entrainment.

- In new simulation, we did not consider the release of latent heat at ice shelf bottom. Instead of this, we will discuss the specific degree of changes by ice melting at ice shelf bottom and frazil ice formation at sea surface.

Line 178: "This difference is induced by high momentum exchange by refreezing and its brine rejection." This statement is unclear. Is the high momentum flux related to the destruction of stratification by brine release?

- In new simulation, we did not consider the release of latent heat at ice shelf bottom. This

statement will be removed in revised manuscript.
* * *
The horizontal velocity orientation throughout the text is unclear. Are zonal velocities aligned with the x-axis of the simulation domain? In some places in the text the zonal velocity but not the meridional velocity is presented.

- As shown in R-Figure 7, we will present flow structures of velocities with Ekman layer in revised manuscript.

[Figure]

**R-Figure 7. u (zonal), v (meridional) velocity profiles of new simulations.**
* * *
Line 185: "negative mean velocity": the velocity vector orientation is unclear.

- We will provide information for directions in revised manuscript.
* * *
Line 215: "stream-wise zonal velocity": the velocity vector orientation is unclear.

- We will provide information for directions in revised manuscript.
* * *
Line 186: The description of the forces responsible for the ice front eddy are unclear. Can you also describe the eddy structure more clearly? Is it a singular overturning cell spanning the length of the ice front?

- In new simulation, the overturning cell near ice front was not observed. So, we will remove these contents for the ice front eddy in revised manuscript.

Line 189: Are you saying here that convection due to brine rejection inhibits entrainment? It is unclear why this would be the case.

- In new simulation, this part will be removed.

Line 194: "upward advection from sub-ice shelf plume" is unclear. Are you talking about entrainment due to turbulence or advection by the mean flow?

- In previous results, we talked about entrainment due to turbulence (heat advection by turbulent mixing).

Line 195: "stratification is more dominant than flow shear" You haven't made a strong case for this in Results. Perhaps you could move this statement to the discussion and expand on it there.

- In new simulation, ratio between buoyancy force and shear production will be examined via flux Richardson number flow shear in revised manuscript.

Line 200: "upward flow advection" again, is this mean flow or turbulence?

- In this part in previous simulation, we used "upward flow advection" as mean flow (upward moving of sub-ice shelf plume).

Line 200: "Since there is no downward force due to brine rejection, the upper region of the sub-ice shelf plume expanded to the upward direction immediately after it passed the ice front." It's unclear what you determine the driving mechanism to be. Is it that changes in stratification determine the sub-ice shelf plume extent or changes in the degree of mixing between water masses or something else?

- In new simulation, dynamics and physics were different with previous simulation. So, we will remove this sentence.

Results: The relationship between refreezing and entrainment of supercooled water at the ice shelf front is unclear in the manuscript. How can we tell that the higher rates of refreezing at the ice shelf front are due to entrainment as opposed to reversed flow of cooler water from the open ocean below the ice front?

- In new simulation, it was observed that reversed flow of warm water from open ocean caused ice

melting (R-Figure 5).
* * *
Figure 5. Show the inertial subrange in wavenumber space. You mention that the low wavenumber values don't fit the -5/3 slope but aren't these wavenumbers outside the inertial subrange? Why are these spectra so noisy? Could this indicate insufficient spatial or temporal averaging? You don't specify these details in your description of the methodology. It's also unclear why there are several curves for each case and depth. Furthermore, why are the spectra evaluated in 1-d?

- As the reviewer mentioned, energy spectra in low wavenumbers is outside the inertial subrange. When we obtain this spectra plot, we consider velocity fluctuation of spanwise direction (homogeneous by cyclic boundary condition) at specific depth and specific zonal (streamwise) location, because energy spectra calculation have to be performed under the assumption for homogeneous turbulence. in previous simulation, insufficient spatial averaging caused noisy feature. In new simulation, we will plot the energy spectra as shown in R-Figure 8.

[Figure]

**R-Figure 8. 1D Energy spectra in new simulation. These plots are obtained at 297m (IOBL center), 567m depth (Plume center).**
* * *
Figure 7, 8. There doesn't appear to be a relationship between heat flux at -281 m and refreezing rate. Can you explain why this is the case?

- As the reviewer mentioned, heat flux at -281 m was not related with refreezing pattern (melting pattern in new simulation). Because melting pattern is related with temperature, integral of local heat flux near ice shelf will be related with melting pattern. We will re-plot this figure, including relationship between the melting pattern and vertical structure of heat flux.
* * *
Lines 215-231: You address differences in velocity magnitude but not orientation.

- To orientation of oceanic flow, we will add vertical profiles of velocities (x, y directions) in revised manuscript. (R-Figure 7)
* * *
Line 230: This caveat should be explored more deeply in the discussion.

- In revised manuscript, this part will be moved to discussion section.
* * *
Line 234: "it is shown that the LES model adequately resolves the oceanic flow beneath the ice shelf with the proper refreezing effect." The analysis presented in Section 3.1 does not demonstrate this. You don't show evidence for sufficient resolution (e.g., a comparison of resolved vs. subgrid energy) or the proper refreezing effect (e.g., a combination of the right theory and match to observations).

- In new simulation, we will discuss the model validity with proper melting effect and latent heat and salt by frazil ice formation with our hypothesis.
* * *
Line 240: Needs further discussion in text to explain why vertical velocity is not zero given boundary conditions.

- We will discuss this based on Monin-Obukhov similarity at ice surface in revised manuscript.
* * *
Line 257: "strong velocity gradient" in what direction?

- It was vertical gradient.
* * *
Line 264: the definition of IOBL should come at the first mention of IOBL in the Results or Methods.

- We will add the IOBL definition in the results section.
* * *
Figure 11: Is IOBL depth the same for both regions? Inset: Is the arrow for the inset meant to correspond to a certain depth? Is sigma the same for both runs? Are there multiple PDFs for each region overlain, because I was expecting to see a single curve for each region as opposed to the scattered points?

In previous simulation, IOBL depth between inner and outer regions was slightly different (inner: 438 m, outer: 447 m).

Line 292: "This water plume refreezes. . ." this statement is only true for some ice shelves.

- Reviewer's comments is correct. We will amend this sentence.

Line 304: "we used the LES model to expand the one-dimensional observation profile in oceanic region to the three-dimensional flow-field in oceanic region and the sub-ice shelf region." This statement is imprecise and possibly misleading. It would be more accurate to say that you used the observational profile as initial and boundary conditions and then investigated spatial and temporal variability arising from brine rejection and mixing.

- It will be corrected as the reviewer mentioned.

Line 306: "We assumed that the LES results for the sub-ice shelf region are validated if the LES results for the oceanic region are validated." See validation comment above.

- Based on new simulation and its validation, we will suggest this statement in revised manuscript.

Line 307: "Via an evaluation of the refreezing effect" It's not clear what this evaluation consists of and how it supports the validity of the model.

- Based on new simulation and its validation, this part will be changed to "via an evaluation of the melting effect" in revised manuscript.

Line 314: You haven't provided much explanation of the causes of heterogeneous freezing rates and what controls the scale of turbulence features.

- We will remove contents for the freezing rate beneath the ice shelf.

Line 315: what is the scale of the ice front eddy and what controls it?

- In new simulation, reversed flow and ice front eddy were not observed.

Line 316: what determines the IOBL depth and does it match the observations?

- We determined the IOBL depth as the depth at 1% of heat flux at ice shelf bottom. In future study, we will consider this with new observations using AUV, glider.

Line 319: "this study can be improved by comparing LES results with observations and its feedback" What do you mean by "its feedback"?

- "Its feedback" meant the modification of constants or model parameters by the comparison between LES result and in-situ observation. We will rephrase this sentence in revised manuscript.

Line 321: "If a database for flow physics in various parameters is completed" I can see what you're getting at here, but the wording here is awkward and it's unclear what you mean by "a database for flow physics.

- In this sentence, database meant the model parameters (e.g. turbulent exchange coefficients) according to various flow environments. We will rephrase this sentence to clarify original meaning.

Line 328: "convergence trend in temporal variance" of what quantity?

- When we mentioned the temporal variance, we examined time series of the friction velocity.

- Technical comments

Make it clear in the introduction why you use the term "refreezing" as opposed to "freezing" There are a few places in the text where you say that water melts, but I think you mean to say that the water mass has a contribution of meltwater from the ice shelf.

- Because there is no refreezing beneath ice shelf in new simulation, we will not use the "refreezing" in introduction section. Text for water mass will be modified as the reviewer mentioned.

Specify which version of PALM you are using.

- We will add the version of PALM (version 6, r4536) at the first mention of PALM in methodology section in revised manuscript.

Line 17: "high shear impact"?

- We will amend this to high velocity shear in revised manuscript.

Line 46, 184: "the shear impact" is confusing if what you mean is more similar to "the direct impact" and does not relate to velocity shear.

- We will amend this to the velocity shear in revised manuscript.
* * *
Line 107: Sub-grid parameterizations need a citation.

- This part was referred to PALM description of Maronga et al. (2015). We will add the citation.
* * *
Line 181: include a citation for the original definition of the swirling strength criterion

- We will remove the swirling strength criterion part.
* * *
Line 182: "Due to" to "At its"

- It will be corrected as the reviewer mentioned.
* * *
Line 182: "apart from the ice" to "below the ice"

- It will be corrected as the reviewer mentioned.
* * *
Line 206: "with few dissipations" to "with little dissipation"

- It will be corrected as the reviewer mentioned.
* * *
Line 235: "explores" to "explore"

- It will be corrected as the reviewer mentioned.
* * *
Line 274: "to IOBL the flow"

- It will be corrected as the reviewer mentioned.
* * *
Line 277: reference for the flatness factor

- We will add the reference for the flatness factor.
* * *
Line 281: strange to say that the vertical velocity fluctuations "have" 3 sigma.

- We will remove this part in revised manuscript.
* * *
Line 302: "the numerical approach including the LES" is confusing. Do you just mean LES on its own?

- Yes. We just meant LES. We will amend this in revised manuscript.
* * *
In all figures, specify whether results are derived from simulation or observations or both.

- We will specify this in all figures and its captions.
* * *
Figure 1: a. Unclear what blue shading designates. Add "sea" to Ross, Amundsen, Weddell, Bellingshausen labels. b. Show ice shelf boundaries and label sea ice areas. c. label ice shelf length dimension.

- We will add the "sea" to Antarctica map. Also, we will add ice shelf length dimension.
* * *
Figure 6. b. Show local freezing point, especially given that you claim supercooled water entrainment.

- We will type the local freezing temperature in explanation for vertical profile of temperature.
* * *
Figure 9. I can't tell the sign of vertical velocity without a reference zero line.

- We will add reference zero line at velocity figure.

Reference

Thompson, L., Smith, M., Thomson, J., Stammerjohn, S., Ackley, S., & Loose, B. (2020). Frazil ice growth and production during katabatic wind events in the Ross Sea, Antarctica. The Cryosphere, 14(10), 3329-3347.

Heorton, H. D., Radia, N., & Feltham, D. L. (2017). A model of sea ice formation in leads and polynyas. Journal of Physical Oceanography, 47(7), 1701-1718.

Vreugdenhil, C. A., & Taylor, J. R. (2019). Stratification effects in the turbulent boundary layer beneath a melting ice shelf: Insights from resolved large-eddy simulations. Journal of Physical Oceanography, 49(7), 1905-1925.

Gwyther, D. E., Galton-Fenzi, B. K., Dinniman, M. S., Roberts, J. L., & Hunter, J. R. (2015). The effect of basal friction on melting and freezing in ice shelf–ocean models. Ocean Modelling, 95, 38-52.

Stevens, C., Lee, W. S., Fusco, G., Yun, S., Grant, B., Robinson, N., & Hwang, C. Y. (2017). The influence of the Drygalski Ice Tongue on the local ocean. Annals of Glaciology, 58(74), 51-59.

Yoon, S. T., Lee, W. S., Stevens, C., Jendersie, S., Nam, S., Yun, S., ... & Lee, J. (2020). Variability in high-salinity shelf water production in the Terra Nova Bay polynya, Antarctica.

Maronga, B., Gryschka, M., Heinze, R., Hoffmann, F., Kanani-Sühring, F., Keck, M., ... & Raasch, S. (2015). The Parallelized Large-Eddy Simulation Model (PALM) version 4.0 for atmospheric and oceanic flows: model formulation, recent developments, and future perspectives. Geoscientific Model Development Discussions 8 (2015), Nr. 2, S. 1539-1637.

Anonymous Referee #2

- General comments

This paper has an interesting premise and some potentially significant results, but substantial analysis and revision is required before it can be published. There have been very few LES studies conducted on the ice shelf-ocean boundary layer (IOBL) and I have yet to see one with refreezing included, which makes this work of potentially great interest. However, I am concerned that the set up, validation and analysis of the LES needs considerably more work. I have outlined my major concerns below in the specific comments section, and I would like to see these comments thoroughly addressed.

- Specific comments

1. I have concerns about the set up of the LES. The authors describe that the inlet plume boundary condition beneath the ice shelf (and the initial conditions across the whole domain) are set from in situ observations. The in situ observations are measured at the front of the ice shelf, but the inlet plume boundary condition is much further beneath the ice shelf (1000m or so horizontally). According to the authors and the premise of the paper, the IOBL undergoes significant brine rejection, latent heat release and mixing due to refreezing at the ice edge. The resulting profiles of temperature, salinity and velocity in the open ocean are then compared against the same in situ measurements used to force the simulation, to conclude that the refreezing effect is significant. It does not seem appropriate to force the model with the in situ field measurements (with the inlet forcing at a different region from where the field measurements were taken) and then validate the model against the same in situ measurements (but this time using model results in front of the ice shelf, similar to where the field measurements would have been taken). Perhaps I am missing something in the set up here?

The primary difference between the refreezing and no-refreezing LES depth profiles in front of the ice shelf is then found to be the increase in temperature in the top 400m (Figure 6). I agree that the refreezing case looks closer to the field measurements than the no-refreezing case. But I also question that the initial conditions, and perhaps more importantly the inlet conditions on the plume, have a temperature profile that has values smaller than the average measurement profiles (in particular at 400-550m depth on Figure 6). The authors need to further justify why they have chosen these initial profiles (dashed lines on Figure 6), and any differences that have been made from the field measurements. It is also not clear to me that the solution they have found is truly unique, in that the inlet temperature profile could potentially be tuned to find that the no-refreezing case also gives a good match with the field measurements. I would appreciate a more transparent explanation of the LES set up, including a thorough discussion on the scientific premise going into the set up.

- As the reviewer mentioned, we concluded that the validation and LES setup with forcing & boundary conditions had some issues. Therefore, we composed new simulation set-up with different boundary conditions based on our hypothesis (R-Figure 1), using only the properties of sub-ice shelf plume. In new simulation, initial and boundary conditions were composed by our hypothesis, not directly from the in-situ observation.
* * *
2. I would like to see further validation of the LES in terms of whether it is resolving the dynamics. The authors currently have only one grid resolution here, so further justification to show why they would expect convergence with a higher grid resolution would be appreciated. The LES resolution validation in the paper was solely focused on the energy spectrum. While it is reassuring to see the -5/3 spectrum, there are other variables that really should be considered, in particular the stress, heat and salt flux. Figure 10 already shows the momentum and heat fluxes with the resolved and SGS terms. The SGS terms appear quite small throughout the whole depth, which is a good sign for LES convergence, but I would like to see this compared against the total values for each depth, along with some discussion. Does the salt flux also look resolved throughout most of the depth (especially near the ice)?

- As the reviewer mentioned, we have tried to validate our grid resolution is proper in this oceanic flow through energy spectrum of $u$ velocity and resolved & SGS flux of heat and momentum. In new simulation, we will discuss our grid size via energy spectra, grid sensitivity study, magnitude comparison in resolved and SGS fluxes. As the reviewer mentioned, SGS fluxes near the ice have relatively large values. It can be solved using stretched-vertical grid, but this has a problem for employing in this study, because simulation domain in this study have an ice shelf geometry. We will discuss the impact of relatively large SGS fluxes near the ice surface in discussion section in revised manuscript. Also, we can plot the resolved salt flux (R-Figure 4), and stabilized features are observed.

[Figure]

R-Figure 4. Vertical profile of salinity flux in new simulations

3. There are some really interesting dynamics in the LES output but I am struggling to understand them and put them into context of past work. For example, the eddy at the ice front is very intriguing but there is not much information on its dynamics. I did not know whether the authors were referring to a vertical overturning type eddy, or a horizontal eddy. If the latter, is it a baroclinic eddy? What is setting the size of this eddy (e.g. the domain size or a Rossby length scale)? Is the potential vorticity in the under-ice region important in constraining where the eddy forms?

- Instead of ice front eddy, we propose the development of ice front circulation in open ocean based on new simulation results. The circulation is vertical overturning type by salinity flux at the sea surface.

Other dynamics that come to mind are the brine rejection from refreezing. Is the brine rejection enough to generate convective plumes and a convective region? Is this the cause of the mixing in the refreezing zone? I cannot see any brine rejection influence on the salinity field (Figure 3) but this might be hidden by the colormap scale. Is there a density inversion here? Also are there any double-diffusive effects expected in the water column?

- In xz contour of $3^*$ time-averaged salinity in new simulation, we cannot observe the density inversion near the sea surface because colormap scale and time averaging (R-Figure 2). Double-diffusive effects (only salt finger) can be observed at initial state (R-Figure 5), but it is not observed in time-averaged salinity contour.

[Figure]

**R-Figure 5. xz contour of salinity at initial state (t = 2 hour)**

What are the effects of the Coriolis parameter in these simulations? Is the horizontal movement of the plume affected by geostrophic balance (is there a strong velocity in the y-direction)? Would we expect an Ekman layer to form near the ice base e.g. Jenkins 2016 ("A Simple Model of the Ice Shelf–Ocean Boundary Layer and Current")?

- Within depth range (from 280 m to 400 m depth where $u$ velocity gradient is large), $v$ meridional velocity with Ekman layer was observed in new simulations.

[Figure]

**R-Figure 6. Vertical profiles of velocities beneath the ice shelf.**

In one of the few previous studies on the exit of meltwater plumes from under ice shelves, Naveira Garabato et al. 2017 ("Vigorous lateral export of the meltwater outflow from beneath an Antarctic ice shelf") concluded that centrifugal overturning instability played an important role in setting the mixing of the meltwater plume. Are the present LES similar to these at all?

- In new simulation results, we had come to a similar conclusion of study of Naveira Garabato et al. 2017. Positively buoyant ice shelf water (PISW) in our study has similar feature. We will discuss this in revised manuscript.

The paper would be really strengthened by discussing the dynamics in the context of past work. In the introduction in general, I would recommend discussing some more field observations (e.g. Larsen C by Davis and Nicholls 2019, Ronne ice shelf by Jenkins et al. 2010, Fimbul ice shelf by Hattermann et al. 2012, Ross ice shelf by Arzeno et al. 2014) to really put the Nansen Ice Shelf observations into context. Similarly, I think it would be worthwhile citing some more process-based studies here also, especially ones involving a meltwater plume (e.g. McConnochie and Kerr 2018; Mondal et al. 2019).

- As the reviewer mentioned, it needs to be more strengthened by discussing the other cold water cavity such as Larsen C, Ronne ice shelf, Ross ice shelf. We will re-organize the literature survey in introduction section, adding those studies in literature survey.

4. The ice-ocean boundary condition for refreezing is the commonly used "three-equation" model, but how appropriate is this model for the refreezing case (in particular the frazil ice case)? The turbulent exchange velocity of heat is set to be 10^-4 m/s (please provide a citation for this value) but this could be influenced in some way by the refreezing dynamics, including the brine rejection or latent heat release. This three equation model is parameterising the heat and salt fluxes mixed towards the base of the ice, so I think more discussion should be included to justify the choice of turbulent exchange coefficients. This is a more minor comment, but could the authors define the friction velocity where it first appears (Eq. 14) and also explain how it is calculated (e.g. is a drag coefficient set?).

- When we performed the new simulation, we considered heat and salt exchange coefficients based on Vreugdenhil and Taylor (2019). Although it has a different friction velocity range, we can interpolate these values using physical relationship. We will discuss this in discussion section in revised manuscript. And, we will also explain the definition of friction velocity.

Li 26: please provide a citation for the line ". . .the ice shelf–ocean interaction in cavities is the dominant driving force for ice thinning"

- This sentence will be changed with reference.

Li 87: I would also appreciate more discussion of the type of LES used here, and why this type of LES SGS parameterisation would work well in this IOBL system. Has PALM has been used in previous studies of ice-ocean flow?

- In this study, 1.5 order Deardorff SGS model was employed for LES SGS parametrization. Using this model, we can solve fluid dynamics and thermohaline physics within IOBL, but small scale physics within viscous sublayer cannot be solved. This is because we have to use turbulent exchange coefficients from high-resolution study such as DNS, another LES. We think that this is the first study for ice-ocean flow using PALM. We will discuss this in discussion section in revised manuscript.

Li 90: Please describe the turbulence closure scheme used here in more detail. Eqs. (1-9): Please ensure that all variables are defined when first introduced (e.g. u_g, Rd, Cp not defined here). Also try to include the values of any constants (e.g. what value of Coriolis parameter f is used?).

- Description of some parameters were missing. We will add the explanation of turbulence closure and parameters. Coriolis parameter is $-1.41 \times 10^{-4}$ which is corresponding to 75 S.

Eq. (6) and Li 113: are K_m and nu_T the same (they are defined similarly)?

- Those parameters are same. We will remove the nu_T in equations.

Li 113: citation for C_m= 0.1 and more explanation needed for the definition of l (where does 1.8z come from for instance?). Is Eq. (13) just the rearrangement of Eqs. (10-12)? If not, where does it come from?

- C_m, mixing length are important factor for determining the eddy viscosity, and we refer the Maronga et al. (2015) and Stable Boundary Lalyer study of Saiki et al. (2000). Eq. (13) is the solution of rearrangement equation of Eqs. (10-12). We will add this information to methodology section in revised manuscript.

Li 129: please include a citation for the turbulent exchange velocity of heat.

- We list-up these coefficients of heat and salt in R-Table 1.

Eqs (14-16): please define friction velocity u* when it first appears. Also is tau the boundary stress?

- As reviewer mentioned, we add the definition for u$^*$, tau (non-dimensional variable for the relaxation of surface fluxes).
* * *
Li 141: Please justify the surface roughness value of 0.07m. Why was this chosen?

- As shown in R-Table 1, we set 0.005m of surface roughness, referring the low drag coefficient, $C_d$=0.001 case in Gwether et al. (2016).
* * *
Li 143: How was the ice shelf modeled in the simulations? Was it immersed boundary, interface condition, etc?

- We modeled ice shelf using mask method of Briscolini and Santangelo (1989) (fixed topography). We will explain this in methodology section in revised manuscript.
* * *
Li 150: How was the U_top value chosen? Is it based off observations? If so, please cite.

- In new simulation set-up, we set the 0 m s$^{-1}$ at the top boundary to exclude the wind effect.
* * *
Li 174: the mean velocity of 0.0729m/s – where did this value come from? Also what is gained by putting the time in terms of the overturning time here?

- In previous simulation, we used the mean velocity and total domain scale to obtain the large eddy turnover time (t$^*$). Since it was not IOBL property, we used the friction velocity and IOBL depth to obtain the large eddy turnover time in IOBL.
* * *
Li 174: I would like more discussion of the fluctuations – it is mentioned that they are because of large-scale eddies beneath the ice shelf, but it would be nice to see what these eddies look a like a little more in terms of velocity, etc.

- In results section of new simulation, we will discuss velocity and its shear beneath the ice shelf to observe the shear effect in IOBL.
* * *
Li 177: "High momentum exchange by refreezing and its brine rejection. . . " please explain the physical processes here a little further. Why does refreezing mean high momentum exchange?

- In previous simulation, de-stratification by brine rejection was induced the vertical fluxes of

momentum and entrainment of momentum.
* * *
Li 178: The time averaged results of 1t* - why not use a longer time-averaging interval? Especially as the fluctuations appear to be on a slightly longer timescale (Figure 2).

- In new simulation, we used the last 3 t* for averaged results after 14 t*
* * *
   Figure 3 and onward: I have some confusion about zonal and meridional velocity here. For some reason I kept thinking that zonal was the y direction, but actually it was the x direction? Please define the velocity directions in terms of x and y, at least initially. This would help me out a lot!

- We will add the directions with boundary conditions in revised manuscript (R-Figure 7)

[Figure]

**R-Figure 7. Domain information and boundary conditions**
* * *
Figure 3: what are the undulations on the plume interface (top and bottom) on the far left hand side of the domain? Why do they form so strongly near the inlet condition? Are these the eddies referred to in Li 174?

- In previous simulation, there was a flow instability and rapid transition of inflow. But, there is no these effects in new simulations.
* * *
Li 181: What is the swirling strength criterion? Please include a one-line explanation.

- We remove this analysis in revised manuscript.

Li 182: "Due to neutral buoyancy, sub-ice shelf plume is about 100 m apart from the ice shelf and has a high velocity" This information is important and should come much sooner in the paper.

- In revised manuscript, we will add an explanation for this neutral buoyancy part to methodology section.

Li 192: ". . . where salinity stratification is not formed. . ." this is tricky to make out on Figure 3 colourmap. Perhaps think of using a different colormap here?

- We will use the different colormap to examine those features obviously.

Li 194: "This demonstrates that the stratification is more dominant than flow shear near the ice front and play a major role in preventing flow advection from subice shelf plume." Is this really shown here? Please explain more about what is meant by the role of "stratification" here?

In new simulation, we will explain more the role of the stratification and stratified force by positively buoyant meltwater.

Figure 6: Where were the vertical profiles taken in the LES?

- We obtained the profiles at 1 km apart from ice front in both previous and new simulations.

Li 226: "However, the multi-layered stratified characteristics of the salinity profile at depth of ice shelf bottom and IOBL top are observed in the case with refreezing effect." What is "multi-layered stratified characteristics" referring to exactly?

- In previous simulation, we referred the multi-stratification layer at near IOBL top (400 m depth) in vertical profiles.

Li 229: "However, it should be noted that, since this is an idealized model, some differences can be expected between the simulated results and observations, in terms of ice shelf bathymetry and surface roughness, the temporal variability of the sub-ice water plume, the drag effect of frazil ice, etc." Please discuss these further. E.g. what effects would each of these processes potentially have on the LES?

- Ice shelf bathymetry (e.g. local slope of ice shelf) and surface roughness can affect the structure of mean velocity and turbulence intensity. Temporal variability can be expected the change in

turbulence properties such as intensity, shear production. We will discuss this in discussion section in revised manuscript.
* * *
Li 245: "Additionally, meridional direction-stretched structures are observed at the interface between the inner and outer regions." Are these referring to the domain-sized undulations on Figure 7?

- In previous simulation, we referred the stretched eddy at interface between inner and outer regions.
* * *
Figure 8: Is this the vertical heat flux? Similarly in Li 251, "high negative heat flux" is this referring to the vertical or horizontal heat flux?

- Yes. It was the vertical heat flux.
* * *
Around Li 260: So is the supercooled water mixed up from the plume below, or horizontally (by eddy) from waters outside the ice shelf?

- In this part, we referred the super-cooled water was mixed up and advected to upper region.
* * *
Li 263: "Figure 10 shows the vertical profiles of momentum and heat fluxes within the IOBL. As shown in figure, the depth of the IOBL top (438 m) is determined to be the depth where the magnitude of the heat flux is 1% of the maximum heat flux induced by the sub-ice shelf plume." Is this for the inner or outer region?

- We referred IOBL top in the inner region (438 m). IOBL top of outer region is 447 m. In new simulation, we will use same criterion.
* * *
Li 265: "In the vertical momentum flux in the inner region, negative flux induced by refreezing and stratification is observed, showing that the IOBL flow in the inner region is in a stable condition." I thought that there was little to no refreezing in the inner region?

- In previous simulation, there was little refreezing in the inner region, having a stable condition of IOBL flow, as the reviewer mentioned.
* * *
Li 267: "However, positive flux with large-scale advection (IOBL scale) induced by the ice front eddy is observed in the outer region, showing that the IOBL flow in the outer region is in an

unstable condition" What about the top 20m where the flow appears to have negative momentum flux?

- In previous simulation, negative momentum flux at 20 m depth was induced by large refreezing rate and ice front eddy.
* * *
The flatness factor is of some interest, but I would be more interested to see the energy budget of the simulations (e.g. turbulent kinetic energy, buoyancy production term, etc). Have the authors thought about calculating the energy budget?

- In new simulations, we will not consider the flatness factor analysis. To observe the production and energy budget, we will add the flux Richardson number, although it has not a distinct trend in turbulence intensity.
* * *
First paragraph of the discussion. This paragraph is a nice description of the overall flow – it might be helpful to have this earlier, in the introduction or the simulation set up.

- In revised manuscript, we will re-organize the paragraph in introduction to highlight our hypothesis and findings.
* * *
Conclusions: This should be more to the point, with a succinct summary of the main findings of the paper.

- Using the schematic diagram, we will specify the conclusions for our study.
* * *
Figures: please consider using different colourmaps for each of the velocity, temperature and salinity figures. It might be easier to follow and make comparisons.

- We will re-plot the contours using another colourmap in revised manuscript.
* * *
- Technical corrections

Li 32: type "groundling" should be "grounding". Also ". . .dense and salty water melts the ice. . . "

- It will be corrected as the reviewer mentioned.
* * *
Li 34: "tidal pumping melts" – what process is this referring to?

- In this sentence, we referred to increment of melt rate by tidal shear.

Li 39: Please rephrase the sentence "Even iceberg calving . . ." it does not make sense as it stands.

- We will rephrase this sentence in revised manuscript.

Li 51: ". . . preventing the heat entrainment. . ." Please be clear what heat entrainment is being referred to here.

- "Stratification by a density gradient of meltwater near ice surface prevents the heat entrainment"

Li 64: Gayen et al. 2016 used DNS not LES.

- It will be corrected as the reviewer mentioned.

Li 105: bracket missing in the fourth term on the RHS.

- It will be corrected as the reviewer mentioned.

Li 185: ". . . with negative mean velocity. . ." mean velocity in which direction?

- Zonal direction. We will clarify velocity direction in revised manuscript.

If possible, try not to start all the paragraphs with "Figure x . . ." it is a little clunky.

- We will amend this in revised manuscript.

Reference

Thompson, L., Smith, M., Thomson, J., Stammerjohn, S., Ackley, S., & Loose, B. (2020). Frazil ice growth and production during katabatic wind events in the Ross Sea, Antarctica. The Cryosphere, 14(10), 3329-3347.

Heorton, H. D., Radia, N., & Feltham, D. L. (2017). A model of sea ice formation in leads and polynyas. Journal of Physical Oceanography, 47(7), 1701-1718.

Vreugdenhil, C. A., & Taylor, J. R. (2019). Stratification effects in the turbulent boundary layer beneath a melting ice shelf: Insights from resolved large-eddy simulations. Journal of Physical Oceanography, 49(7), 1905-1925.

Gwyther, D. E., Galton-Fenzi, B. K., Dinniman, M. S., Roberts, J. L., & Hunter, J. R. (2015). The effect of basal friction on melting and freezing in ice shelf–ocean models. Ocean Modelling, 95, 38-52.

Stevens, C., Lee, W. S., Fusco, G., Yun, S., Grant, B., Robinson, N., & Hwang, C. Y. (2017). The influence of the Drygalski Ice Tongue on the local ocean. Annals of Glaciology, 58(74), 51-59.

Yoon, S. T., Lee, W. S., Stevens, C., Jendersie, S., Nam, S., Yun, S., ... & Lee, J. (2020). Variability in high-salinity shelf water production in the Terra Nova Bay polynya, Antarctica.

Maronga, B., Gryschka, M., Heinze, R., Hoffmann, F., Kanani-Sühring, F., Keck, M., ... & Raasch, S. (2015). The Parallelized Large-Eddy Simulation Model (PALM) version 4.0 for atmospheric and oceanic flows: model formulation, recent developments, and future perspectives. Geoscientific Model Development Discussions 8 (2015), Nr. 2, S. 1539-1637.

Saiki, E. M., Moeng, C. H., & Sullivan, P. P. (2000). Large-eddy simulation of the stably stratified planetary boundary layer. Boundary-Layer Meteorology, 95(1), 1-30.

Briscolini, M., & Santangelo, P. (1989). Development of the mask method for incompressible unsteady flows. Journal of Computational Physics, 84(1), 57-75.

---

## Referee Report (RR1)

I appreciate the effort the authors have made to revise the manuscript, including the revision of simulations under a modified set-up. The authors provided useful clarification on many of my initial questions about the simulation set-up; however key questions still remain or are raised by the new set-up that should be clarified before publication. The new dynamics examined by the manuscript are generally not explained clearly enough, particularly the driving forces behind the ice-shelf front circulation cell. Na et al. mention 3-d turbulent structures several times in the manuscript but do not describe them. I'd like to see descriptions of the eddies in the IOBL (mentioned on line 177) and those in the PISW. This could help readers understand the simulated dynamics. I appreciate that the authors have offered more information about the oceanographic observations, but the manuscript could still benefit from more discussion of the relationship of LES results to both the oceanographic observations presented and melt rate observations.

Major revisions:

Methodological questions/concerns that should be addressed in the manuscript text:

- Lines 135-139 do not provide enough detail on how the theoretical velocity profiles are utilized in the model and how this relates to the solution of friction velocity. These are really important details for understanding the momentum fluxes in the ice-ocean boundary layer, the validity of PISW results, and what the steady-state friction velocity means in Figure 2.
- I'm troubled by the momentum and heat flux profiles shown in Figure 9. These profiles appear to show that momentum and heat fluxes go to zero at the ice shelf base, implying that there is negligible melting and drag. I found the text addressing these fluxes (paragraph starting on line 279) hard to understand. Can you also relate these fluxes to the spatial evolution of PISW and IOBL as they are advected (i.e., are they gaining or losing heat or momentum)?
- The calculation of the freezing rate in the open ocean is not included in the methods. Is it permitted only at the surface or throughout the water column? There should be associated caveats in the Methods and Discussion about potential frazil ice effects not considered in your simulations, with citations to existing literature on frazil ice effects.
- You imply on line 329 that you aren't using a dynamic SGS model but on line 110 you have included a dynamic SGS equation.
- The thermodynamics of the ice-shelf front are not addressed. Do you allow lateral melting?

The introduction of an ice-front circulation cell warrants further explanation of these dynamics than is currently included in the manuscript. On line 187, the authors write "the development of this circulation is mainly induced by the downward force of salt flux by sea ice formation and the shear stress of sub-ice shelf plume." How does sea-ice formation relate to downwelling? I'd expect convective mixing. What role does sub-ice shelf plume momentum play? Since winds are excluded, how might the results change if winds were included? Is the hypothesized circulation cell compatible with observed sea ice advection patterns? How is this similar to or different from

the role that ice-shelf meltwater plays in this study: Malyarenko, A., Robinson, N. J., Williams, M. J. M. & Langhorne, P. J. 2019. A wedge mechanism for summer surface water inflow into the Ross Ice Shelf cavity. *Journal of Geophysical Research: Oceans*. 10.1029/2018JC014594

The relationship between LES results and oceanographic observations also warrants further explanation. You say (line 216) that the signature of PISW is in the CTD profiles but it is unclear what this signature is in relation to Figure 4.

The agreement between the simulated ice-shelf melt rate distribution and the observed melt rate distribution also needs to be discussed.

The explanation for heterogeneous PISW upwelling is unclear to me. I think it would help to see a planar view of ice-shelf cavity circulation. I wonder if the boundary conditions imposed may be influencing the circulation to a greater extent in the high turbulent shear case.

189: "The circulation pushes the sub-ice shelf plume with downward forcing, making that stratification line near ice shelf is moved to about 350 m depth." Clarify the relative importance of downwelling and mixing for deepening the halocline.

204: This paragraph would be a good place to include a comparison of PISW depth and meltwater fluxes between the 4 cases.

226: "Because the amount of the PISW in the strong turbulence case is larger than that in the weak turbulence case, its turbulence energy spectra within IOBL (297 m) is the lowest." This needs more explanation.

243: "it is shown that the LES model adequately resolves the oceanic flow beneath the ice shelf with the proper thermohaline dynamics by the melting effect beneath the ice shelf and the freezing effect at the sea surface" This is a very general statement. What thermohaline dynamics do you have confidence in?

246: "there are shear forces caused by the momentum of the sub-ice shelf plume and the buoyancy force..." This is not adequately explained. What is the relationship between the stratification, momentum fluxes, and buoyancy? It's worth reminding the reader that the ice shelf based is not sloped so buoyancy does not drive mean flow.

Figure 6: The x-axis appears to span the whole domain, but there should only be freezing at the sea surface in the open ocean part of the domain.

Figure 7: Specify whether this figure includes or excludes the region of underdeveloped turbulence.

The features we're seeing in Figure S3 should be explained in the text. I wasn't able to figure out the difference between right and left panels in this figure.

317: It sounds as if you're saying that PISW is transferring momentum to IOBL in the ice-shelf cavity. Where is this momentum coming from?

322: Explain how the circulation is similar to centrifugal overturning.

330: How will the findings be used to interpret observations?

Minor revisions:

23: "which in turn slows sea level rise" This can be misleading because the rate of sea level rise may increase when ice shelves are removed but not necessarily when the ice sheet reaches a new equilibrium without the ice shelf.

The authors provided more detail about the CTD and ADCP data collection, which was appreciated. One remaining detail that would be useful to the reader is the distance between the ice front and the observations.

Specify $t^*$ in hours.

Thank you for clarifying the velocity orientation in this revision. I think it's also worth pointing out to the readers that the zonal velocity is perpendicular to the ice-shelf front geometry (especially in the caption for Figure 3), even though this can be seen in the new figure.

128: I think the notation should be $S(z_1)$ rather than $Sa(z_1)$ since you use $S_b$.

261: The way this is written, there is an apparent inconsistency in the strong turbulence case that it has the smallest mean shear gradient, highest TKE, yet you say that TKE production is proportional to mean shear gradient and turbulent shear stress.

300 "we used the LES model with proper boundary conditions" "proper" isn't appropriate here, as it is quite subjective.

Section 4.2. It would be helpful to mention the temperature difference between PISW as it exits the cavity and the sea surface freezing point.

319: "showing that this result is in agreement with the previous study of Jenkins (2016)" This is too general. It's simply that both this study and Jenkins have Ekman layers below the ice shelf, right?

327: "constant turbulent coefficients in SGS model" Which coefficients? I thought you were using a dynamic SGS model.

335: It's unclear what effects you'll be examining with the "vertical distribution of pressure"

336: "With better understanding of various parameters on basal melting" Which parameters?

The schematic diagram (Figure 10) is a great addition to the manuscript. I do find it somewhat confusing to include katabatic winds in the schematic when they do not play a role in your explanation of the dynamics, particularly as they appear to be opposed to the ice front circulation pattern.

**There are several places where the meaning of the text is unclear:**

16: "In the strong turbulence case, there are distinct features in basal melting and flow characteristics." This is too vague.

30 "The driving forces for basal melting in cold water cavity are shear force by tidal mixing and the thermohaline process by sea ice formation" Both of these forcings need more introduction and explanation here.

35 "Therefore, because driving forces from the ocean and opposite forces by the meltwater merge within the boundary layer" I don't know what is meant here. What forces?

43: "Similar features for weak stratification" Which features are you referring to?

48: "controlling the shear impact of its momentum" Please clarify

57: "In order to find out the effects of various forcing clearly" Please specify which forcings

185: "velocities in two cases" Which two cases?

211: "upper streamwise direction" Would be more clear to put in terms of zonal/meridional or parallel/perpendicular to ice front

302: "Additionally, we set to ambient values" set what to ambient values?

356: "it means that stratified forcing by PISW has a nonlinear feature for flow shear by strong turbulence"

332: "this study can be improved by comparing LES results with observations and their feedback" "and their feedback" is unclear.

**There are also several places where the grammar needs revision:**

9. "but there is a poor understanding of the *fluid dynamic, thermohaline physics* of the IOBL flow"

11: "velocity's theoretical profile" >> "theoretical profile for velocity"

28: "The sub-ice shelf oceanic environment can be divided into broad classifications" >> "The sub-ice shelf oceanic environment can be divided into two classes"

38: "physics in ice shelves" >> "physics below ice shelves"

69: "Nansen Ice Shelf (NIS; cold-water cavity)" >> "Nansen Ice Shelf (NIS), a cold-water cavity,"

72: "while remaining thermohaline forcing by the melting and freezing"

177: "are highly fluctuated in" >> "greatly fluctuated during"

178: "As turbulence within IOBL is stronger, the magnitude of fluctuation is larger."

189: "Noticeable difference between the two cases" >> "A noticeable difference between the two cases"

194: "with different momentum along to streamwise direction in two cases."

307: "for resolving the IOBL and oceanic flow in reality" >> "for simulating the IOBL and oceanic flow more realistically"

---

## Referee Report (RR2)

This paper reports on large-eddy simulations of an ice shelf-edge region, inspired by observations made in front of Nansen Ice Shelf, Antarctica. The simulations are run with an idealised geometry and boundary/initial conditions for temperature, salinity and velocity that are inspired by observations. The velocity boundary condition beneath the ice shelf is varied to simulate four different regimes with varying levels of turbulence. The velocity profile between the ice shelf and the continental shelf below is modelled using a power-law velocity profile, where the power-law relationship is varied to simulate varying degrees of turbulence. A three-equation model is used to model the basal melting of the ice shelf, where changes in ice shelf geometry, volume flux input and lateral ice shelf melt are excluded. In the open-ocean portion of the simulation domain, a rigid sea-ice lid is assumed, where the freezing rates are calculated using a three-equation model. Constant heat and salt exchange coefficients are used for the sea-ice region and varying coefficients, calculated using Monin-Obukhov similarity theory, are used in the ice shelf region (this needs to be clarified, as I may have got this wrong?). The high turbulence simulations show an increased basal melt rate and a stronger Ekman layer, resulting in a modified circulation pattern in the open-ocean region.

I found these simulations to be interesting and I think this paper is of interest to the community. Some changes and clarifications need to be made however before I would be comfortable with publication. Specifically, I think further reference to the existing literature needs to be made. I think the ice front region is a particularly interesting region, not least because it is relatively easy to observe compared with the grounding line. Your simulations are relatively idealised, but I think you should still be able to link your work to previous work in the ice shelf front region, even if the conclusion is that elements of your simulations make comparison difficult. Most importantly, the reader needs to be left with an understanding of what the implications of your simulations are for studies of the ice shelf region. In Garabato et al. (2017), they make the point that the intrusion depth of the ice shelf plume has important ramifications for the effect of ice shelf melt on the Southern Ocean (specifically in simulations). I think motivating your study with this scientific question (or some other broad question) would help clarify the point of your work to the reader.

I have a series of other clarifications and edits to the text that I would like to see which are listed below:

1. It needs to be clear that you are studying an ice front throughout the abstract and the introduction. The geometry of your situation is key to the physics of interest, so must avoid trying to say too much about generic IOBL plumes. i.e. I would revise the sentence in the abstract:
   "In this study, we utilize a large-eddy simulation to investigate the role of the turbulence within the IOBL flow with sub-ice shelf plume"
   To something involving explicitly focused on ocean dynamics at the edge of an ice shelf.
2. This line in the abstract:
   "This demonstrates that the larger baroclinic eddies enforces heterogeneous distribution of positively buoyant meltwater upwelling"
   Is confusing to me. How does a larger melt rate demonstrate that there is a heterogeneous distribution of meltwater upwelling? Surely the melt rate could be homogeneous but larger? This point about heterogenous/homogenous response needs

to be explained more fully in the text (or excluded, as I am not totally sure what it adds to the conclusions of the paper).

3. Lines 32-35 "Shear force generated by tidal mixing and the thermohaline process during sea ice formation are the basal melting driving forces in cold water cavity (e.g., high salinity shelf water), whereas the intrusion of circumpolar deep water (CDW), which is the water well above the local freezing temperature, is the main driving force for basal melting in the warm-water cavity (Davis and Nicholls, 2019; Jacobs et al., 1992; Yoon et al., 2020)."
Needs to be split up into two sentences maybe. One saying shear forces drive turbulent mixing of T and S through the IOBL, and another saying that shear is generated by tidal mixing or by circulation, where the source of temperature and salinity mixed up to the ice base is HSSW or CDW

4. Line 67 : This would be a great opportunity to introduce the novelty of your geometry e.g. However, applications of the LES to IOBL at sub-ice shelf environment are quite limited. The geometry and scales of ice-ocean interaction may be qualitatively different for an ice shelf, particularly when considering the ice front. Ice shelves typically have a thickness of 100s of metres compared with the $O(1\ m)$ scales of sea ice.

5. Line 68 : "In this study, we performed LES experiments for the IOBL and oceanic flow including freezing effect at sea surface and the basal melting process with neutrally buoyant sub-ice shelf plume near ice front."
Split into two sentences, first saying that you are studying the ice shelf plume at the ice front, then saying the effects you are including within your study.

6. Introduction: Paragraph on what would we expect of this flow? Cite observations of this near ice shelf region and the ideas that people use (i.e. Garabato instability work, ice front blocking work from Wahlin et al, 2020?)

7. Line 86: "To simulate the oceanic flow with refreezing"
maybe expand on this, melting is included as well as refreezing so that's worth noting

8. Sa is confusing notationally, it might be better just as S

9. Line 116: you've introduced the governing equations and the turbulent closure which is great. I think this paragraph could do with a sentence that summarises the key positives and drawbacks of your chosen sub-grid scale model. For instance, I don't imagine it works well for regions where the flow becomes laminar?

10. Elaboration is needed on the melt/freeze condition that you apply. My understanding is that you apply a three equation model with constant coefficients in the open-ocean/freezing region, and a three equation model with exchange coefficients calculated using Monin-Obukhov theory for the ice shelf region. Is this correct? If so you need to state this explicitly

11.  You include citations for your three-equation model values in the table, but I think mentioning your sources in the text would be beneficial for the reader

12. Line 135: "where $u*$ is the friction velocity which is calculated by the velocity at first node and roughness length", are you saying that you infer the friction velocity using a drag coefficient, using the relation $U^2 = C_d u*^2$ ? If so, you should state this explicitly including your value for $C_d$ and your source for that number. The true drag coefficient is defined using the vertical shear at the boundary; however, I don't

think you would resolve those scales in your simulations, so I presume you are using a drag coefficient.

13. Line 157 : "Initial profiles were set in the variation range of vertical profiles of our 24 CTD and 23 LADCP observations conducted near the ice front of the NIS." I don't understand this sentence. Are you saying that the initial profiles are taken as a mean of the vertical profiles or a smoothed version? Can you detail the exact method for choosing your idealised profiles?

14. Line 158-159: "The outlet boundary condition was determined to match the radiation boundary condition (extrapolation)", could you elaborate further on this boundary condition? I do not understand which radiation you are referring to and what the form of the boundary condition is? This is important for determining the utility of your inferred circulation

15. Line 160-162: Your Dirichlet boundary condition for velocity implies that you have a rigid lid. Your satellite imagery shows a region of ice-free ocean at the edge of the ice shelf. You have assumed essentially that it is ice-covered and the ice does not move (similar to land-fast sea ice). I do not think this is necessarily a problem, but it should be pointed out when you introduce your boundary condition. Are you simulating a winter-version of this ice front region?

16. Line 221: "Below 400 m depth, a well-stratification features appear in salinity distribution." I do not know what this sentence refers to, please clarify.

17. Line 246 : "it is necessary to confirm that the turbulence characteristics of the LES result are similar to the turbulence characteristics of inertial subrange in which energy cascading occurred with few dissipations", I understand most of this sentence but I do not understand the last three words. Are you just saying it is necessary to confirm that the resolved turbulence in your model follows a inertial scaling?

18. You should include details of how you calculated the 1D energy spectra, perhaps including the equation for your calculation. Specifically, I'm wondering whether it's calculated along a single line in the y-direction for instance? I'm slightly confused as to why you have so many fewer points at the high wavenumber end of your plot than at the low wavenumber end. Usually, power spectra show the opposite trend, as you have more points to evaluate small wavelengths with, you get a higher density of points at the high wavenumber end. If you tried calculating your spectra using a numerical method such as the Welch method, you might also get a smoother result, as currently it's difficult to determine whether the presented spectra do indeed show a -5/3 slope or not. Also, I don't think it's a problem if they do not show a -5/3 spectrum, as you are simulating an anisotropic flow, so you might not expect a classic inertial subrange. Your SGS model may assume homogeneous isotropic turbulence, but your resolved turbulence does not need to.

19. Line 317 : "Negative heat flux at 320–400 m depths denotes that some of the entrained heat by the intrusion of the outer ocean is transferred to the downward direction." I don't understand this argument. In your temperature and salinity plots, the profiles seem to be increasing with depth in the sub-ice shelf region, so why would the heat flux be negative? I think this needs further explanation

20. Your first discussion section seems more like introduction than discussion to me. Your results aren't discussed, rather the broad field and approach is discussed. I

suggest including the key points from this section in a paragraph or two in the introduction then removing the section.

21. If you wanted to make the point about the two circulations more convincing to the reader, you could include a snapshot of vorticity as well as zonal velocity (which doesn't show us the vertical velocities that complete your overturning.
22. You mention the work of Garabato et al. 2017, and I think you could explore the connections here further. If you claim that the mechanism in your simulations is similar, then you need to provide evidence of this fact. I suggest calculating some of the metrics used in Garabato et al. 2017, specifically the Richardson angle (Thomas et al. 2013).
23. Turbulence intensity is a confusing metric, could you instead show dissipation rate? This would make your work more directly comparable to the observations in Garabato et al. 2017.
24. Should we expect any ice front blocking effect in your simulations as in the work of Wahlin et al. 2020? If not, why not? This paper would be worth discussing with reference to your simulations.
25. Ensure your list of references is consistent (specifically the placement of the year)

Garabato, A. C. N., Forryan, A., Dutrieux, P., Brannigan, L., Biddle, L. C., Heywood, K. J., Jeknins, A., Firing, Y. L., Kimura, S. 2017, Vigorous lateral export of the meltwater outflow from beneath an Antarctic ice shelf. Nature, 542(7640), 219–222

Wåhlin, Anna K., Nadine Steiger, Elin Darelius, Karen M. Assmann, Mirjam S. Glessmer, Ho Kyung Ha, Laura Herraiz-Borreguero et al. "Ice front blocking of ocean heat transport to an Antarctic ice shelf." *Nature* 578, no. 7796 (2020): 568-571.

Thomas, L. N., Taylor, J. R., Ferrari, R. & Joyce, T. M. Symmetric instability in the Gulf Stream. *Deep-Sea Res. Part II* **91**, 96–110 (2013)

---

## Editor Decision (ED1)

**Referee Report #1**

I appreciate the effort the authors have made to revise the manuscript, including the revision of simulations under a modified set-up. The authors provided useful clarification on many of my initial questions about the simulation set-up; however key questions still remain or are raised by the new set-up that should be clarified before publication. The new dynamics examined by the manuscript are generally not explained clearly enough, particularly the driving forces behind the ice-shelf front circulation cell. Na et al. mention 3-d turbulent structures several times in the manuscript but do not describe them. I'd like to see descriptions of the eddies in the IOBL (mentioned on line 177) and those in the PISW. This could help readers understand the simulated dynamics. I appreciate that the authors have offered more information about the oceanographic observations, but the manuscript could still benefit from more discussion of the relationship of LES results to both the oceanographic observations presented and melt rate observations.

Major revisions:
Methodological questions/concerns that should be addressed in the manuscript text:
● Lines 135-139 do not provide enough detail on how the theoretical velocity profiles are utilized in the model and how this relates to the solution of friction velocity. These are really important details for understanding the momentum fluxes in the ice-ocean boundary layer, the validity of PISW results, and what the steady-state friction velocity means in Figure 2.
● I'm troubled by the momentum and heat flux profiles shown in Figure 9. These profiles appear to show that momentum and heat fluxes go to zero at the ice shelf base, implying that there is negligible melting and drag. I found the text addressing these fluxes (paragraph starting on line 279) hard to understand. Can you also relate these fluxes to the spatial evolution of PISW and IOBL as they are advected (i.e., are they gaining or losing heat or momentum)?
● The calculation of the freezing rate in the open ocean is not included in the methods. Is it permitted only at the surface or throughout the water column? There should be associated caveats in the Methods and Discussion about potential frazil ice effects not considered in your simulations, with citations to existing literature on frazil ice effects.
● You imply on line 329 that you aren't using a dynamic SGS model but on line 110 you have included a dynamic SGS equation.
● The thermodynamics of the ice-shelf front are not addressed. Do you allow lateral melting?

The introduction of an ice-front circulation cell warrants further explanation of these dynamics than is currently included in the manuscript. On line 187, the authors write "the development of this circulation is mainly induced by the downward force of salt flux by sea ice formation and the shear stress of sub-ice shelf plume." How does sea-ice formation relate to downwelling? I'd expect convective mixing. What role does sub-ice shelf plume momentum play? Since winds are excluded, how might the results change if winds were included? Is the hypothesized circulation cell compatible with observed sea ice advection patterns? How is this similar to or different from the role that ice-shelf meltwater plays in this study: Malyarenko, A., Robinson, N. J., Williams, M. J. M. & Langhorne, P. J. 2019. A wedge mechanism for summer surface water inflow into the Ross Ice Shelf cavity. Journal of Geophysical Research: Oceans . 10.1029/2018JC014594

The relationship between LES results and oceanographic observations also warrants further explanation. You say (line 216) that the signature of PISW is in the CTD profiles but it is unclear what this signature is in relation to Figure 4.

The agreement between the simulated ice-shelf melt rate distribution and the observed melt rate distribution also needs to be discussed.

The explanation for heterogeneous PISW upwelling is unclear to me. I think it would help to see a planar view of ice-shelf cavity circulation. I wonder if the boundary conditions imposed may be influencing the circulation to a greater extent in the high turbulent shear case.

189: "The circulation pushes the sub-ice shelf plume with downward forcing, making that stratification line near ice shelf is moved to about 350 m depth." Clarify the relative importance of downwelling and mixing for deepening the halocline.

204: This paragraph would be a good place to include a comparison of PISW depth and meltwater fluxes between the 4 cases.

226: "Because the amount of the PISW in the strong turbulence case is larger than that in the weak turbulence case, its turbulence energy spectra within IOBL (297 m) is the lowest." This needs more explanation.

243: "it is shown that the LES model adequately resolves the oceanic flow beneath the ice shelf with the proper thermohaline dynamics by the melting effect beneath the ice shelf and the freezing effect at the sea surface" This is a very general statement. What thermohaline dynamics do you have confidence in?

246: "there are shear forces caused by the momentum of the sub-ice shelf plume and the buoyancy force..." This is not adequately explained. What is the relationship between the stratification, momentum fluxes, and buoyancy? It's worth reminding the reader that the ice shelf based is not sloped so buoyancy does not drive mean flow.

Figure 6: The x-axis appears to span the whole domain, but there should only be freezing at the sea surface in the open ocean part of the domain.

Figure 7: Specify whether this figure includes or excludes the region of underdeveloped turbulence.

The features we're seeing in Figure S3 should be explained in the text. I wasn't able to figure out the difference between right and left panels in this figure.

317: It sounds as if you're saying that PISW is transferring momentum to IOBL in the ice-shelf cavity. Where is this momentum coming from?

322: Explain how the circulation is similar to centrifugal overturning.

330: How will the findings be used to interpret observations?

Minor revisions:
23: "which in turn slows sea level rise" This can be misleading because the rate of sea level rise may increase when ice shelves are removed but not necessarily when the ice sheet reaches a new equilibrium without the ice shelf.

The authors provided more detail about the CTD and ADCP data collection, which was appreciated. One remaining detail that would be useful to the reader is the distance between the ice front and the observations.

Specify t* in hours.

Thank you for clarifying the velocity orientation in this revision. I think it's also worth pointing out to the readers that the zonal velocity is perpendicular to the ice-shelf front geometry (especially in the caption for Figure 3), even though this can be seen in the new figure.
128: I think the notation should be $S(z_1)$ rather than $S_a(z_1)$ since you use $S_b$.

261: The way this is written, there is an apparent inconsistency in the strong turbulence case that it has the smallest mean shear gradient, highest TKE, yet you say that TKE production is proportional to mean shear gradient and turbulent shear stress.

300 "we used the LES model with proper boundary conditions" "proper" isn't appropriate here, as it is quite subjective.

Section 4.2. It would be helpful to mention the temperature difference between PISW as it exits the cavity and the sea surface freezing point.

319: "showing that this result is in agreement with the previous study of Jenkins (2016)" This is too general. It's simply that both this study and Jenkins have Ekman layers below the ice shelf, right?

327: "constant turbulent coefficients in SGS model" Which coefficients? I thought you were using a dynamic SGS model.

335: It's unclear what effects you'll be examining with the "vertical distribution of pressure"

336: "With better understanding of various parameters on basal melting" Which parameters? The schematic diagram (Figure 10) is a great addition to the manuscript. I do find it somewhat confusing to include katabatic winds in the schematic when they do not play a role in your explanation of the dynamics, particularly as they appear to be opposed to the ice front circulation pattern.

There are several places where the meaning of the text is unclear:

16: "In the strong turbulence case, there are distinct features in basal melting and flow characteristics." This is too vague.

30 "The driving forces for basal melting in cold water cavity are shear force by tidal mixing and the thermohaline process by sea ice formation" Both of these forcings need more introduction and explanation here.

35 "Therefore, because driving forces from the ocean and opposite forces by the meltwater merge within the boundary layer" I don't know what is meant here. What forces?

43: "Similar features for weak stratification" Which features are you referring to?

48: "controlling the shear impact of its momentum" Please clarify

57: "In order to find out the effects of various forcing clearly" Please specify which forcings

185: "velocities in two cases" Which two cases?

211: "upper streamwise direction" Would be more clear to put in terms of zonal/meridional or parallel/perpendicular to ice front

302: "Additionally, we set to ambient values" set what to ambient values?

356: "it means that stratified forcing by PISW has a nonlinear feature for flow shear by strong turbulence"

332: "this study can be improved by comparing LES results with observations and their feedback" "and their feedback" is unclear.

There are also several places where the grammar needs revision:

9. "but there is a poor understanding of the fluid dynamic, thermohaline physics of the IOBL flow"

11: "velocity's theoretical profile" >> "theoretical profile for velocity"

28: "The sub-ice shelf oceanic environment can be divided into broad classifications" >> "The sub-ice shelf oceanic environment can be divided into two classes"

38: "physics in ice shelves" >> "physics below ice shelves"

69: "Nansen Ice Shelf (NIS; cold-water cavity)" >> "Nansen Ice Shelf (NIS), a cold-water cavity,"

72: "while remaining thermohaline forcing by the melting and freezing"

177: "are highly fluctuated in" >> "greatly fluctuated during"

178: "As turbulence within IOBL is stronger, the magnitude of fluctuation is larger."

189: "Noticeable difference between the two cases" >> "A noticeable difference between the two cases"

194: "with different momentum along to streamwise direction in two cases."

307: "for resolving the IOBL and oceanic flow in reality" >> "for simulating the IOBL and oceanic flow more realistically"

**Referee Report #2**

General comments:

This study uses large-eddy simulations (LES) to examine the effect of turbulence in the ice-shelf-ocean boundary layer (IOBL) of Nansen Ice Shelf. The simulations are based on recent observations of ocean conditions near the Nansen Ice Shelf. The dynamics are forced by a neutrally buoyant plume moving beneath the ice shelf and penetrating into the open ocean. Key results are the basal melt rate of the ice shelf and the freezing occurring at the ocean surface (modelling sea ice formation), both of which increase when the plume is more turbulent. The authors also attempt to discuss some interesting heterogenous structures in terms of Ekman layer dynamics, but I did not follow this explanation as it stands.

The study is of some interest and has been improved from the previous iteration. The new simulations have an idealised velocity input condition to model a plume with four different levels of turbulence. The temperature and salinity input conditions are broadly based on observations taken further away from the ice shelf. The scientific premise (what happens when turbulence is increased?) is much clearer. There are few LES of sub-ice shelf flow, in particular with the aim to match observations. I think that the study has potential for publication, but I have some comments that I would like to see addressed first.

Specific comments:

1. The authors vary only the shape of the input velocity profiles across the four runs. As the set-up is so constrained (in terms of setting specific input profiles of velocity, temperature and salinity) I wonder what portion of the results are caused by the input conditions versus the dynamics in the system. What do the authors expect would happen if there was no basal melt? Would there still be plenty of mixing in the sub-ice shelf flow, and would we still see the increase in surface freezing at the ice edge? Similarly, if the surface freezing was turned off, how would this change the outflowing plume dynamics?

2. I disagree with (or perhaps I did not understand) the Ekman layer explanation for the heterogeneity seen in the strong turbulence cases. I agree that stronger velocities beneath the ice shelf could lead to a stronger Ekman layer, but why wouldn't this be a homogenous response underneath the whole ice shelf? What is constraining the length scale of the heterogeneity? (E.g. could it alternatively be a baroclinic eddy with a Rossby radius of deformation?) I would appreciate more discussion on these points and an in-depth explanation of the physical mechanisms.

3. I think more can be done to compare against other cases, in particular Naveira Garabato et al. (2017). (Side note: this study seemed to be missing from the citation list in the manuscript.) The only comparison with the Naveira Garabato et al. study was L321 "These physics with the Ekman layer and the upwelling behavior of PISW are similar to centrifugal overturning instability and lateral shear proposed by Naveira Garabato et al. (2017)." I was confused by this. Are the authors saying that the mechanisms are similar? And if so, how similar and what are the differences? Again, I would appreciate a more in-depth explanation of the exciting phenomena that is noted in this study. I think this would really strengthen our understanding of the phenomena we might expect to see sub-ice shelf and on the ice edge.

L11: In the abstract "…we impose velocity's theoretical profile varying the power-law index." Consider rephrasing as I did not know what "velocity's theoretical profile" meant when I first

read through. Highlight that the velocity profile is varied and therefore turbulence is also changed.

L35: "Therefore, because driving forces from the ocean and opposite forces by the meltwater merge within the boundary layer (meters to tens of meters) right beneath ice shelf, which is known as the ice shelf–ocean boundary layer (IOBL), we have to investigate the IOBL flow and its structure to reveal the basal melting physics in ice shelves (Holland et al., 2020)." This was a long sentence and a bit unclear. Please consider rephrasing.

L135: The power-law equation $U=U_t (z/z_t)^{(1/n)}$. Firstly, I assume the x signified multiply and not x-direction (was written in the manuscript as $U=U_t \times (z/z_t)^{(1/n)}$)? Secondly, is $z_t$ the surface roughness? Thirdly, when I quickly plotted this power law I got something that looked different to the Figure 2 inset. So how does this equation directly relate to the profiles in Figure 2 inset? Is there some normalisation coming in to play? Finally, are there other studies (apart from Irwin 1979) that have used these profiles? I am wondering if the justification for the plume shape can be strengthened by citing some more studies that use these profiles.

L152: The wind effect is excluded in the simulations by setting the surface velocity to zero, even though observations showed a non-zero value of wind. In the reply to reviewers there was some justification for the zero velocity condition, would it be possible to have the brief explanation included in the manuscript also?

Section 2: I might have missed this, but what was the surface freezing condition on temperature and salinity? Was it a Dirichlet or flux condition?

L175: How was the large-eddy turnover time calculated here?

Figure 2: There is a lot of variability in this figure, which makes it difficult to determine whether it is in equilibrated state. Are there other results that show the simulations are equilibrated?

Results: Are there any indications of inertial waves present in the simulations, due to having Coriolis parameter (f)? I am wondering if this variability would appear on the friction velocity (Figure 2).

Figure 5: Caption does not seem to be consistent with legend. What is the difference between (a) and (b) figures? If it is n=3 and n=7, then why are these values also varied in the legend?

L239: Could the spatial-averaged freezing rate values also be included here? Perhaps an earlier reference to Table 2 would help (first time you reference Table 2 is in Section 4.2 Discussion)?

L262: "Because turbulence kinetic energy production is proportional to mean shear gradient and turbulent shear stress, turbulent shear forcing is highest in the strong turbulence case." Could a citation please be included for the first half of this sentence?

Figure 9: There are some negative heat flux values below the positive values in the IOBL. What is causing this?

Section 4.2: There are a few different ideas discussed in this section. Please consider breaking into separate paragraphs.

L324: Please include a definition for the flux Richardson value. How was it calculated in the

simulations?

Technical corrections:

L34: "opposite forces" perhaps "opposing forces"?

L70: Here and in several other places there is a "the" missing (e.g. "THE main parameter…").

L71: "For consistency in experiments, we considered different turbulence state while remaining thermohaline forcing by the melting and freezing." Was this meant to be more along the lines of: "For consistency in experiments, we considered different turbulence states while keeping the same thermohaline forcing by the melting and freezing."

L97: Units missing on Coriolis parameter.

L124: "…was used in whole cases." Should this be "…was used in all cases."?

L183: Domain center (y=1536m) is different to that stated in Figure 3 caption (y=1728m).

L223: Wavenumber is missing units.

Figure 8, Figure 9a: Are units on (rho u v) correct? Should they be kg m^-1 s^-2?

L257: "Important to not is that a noticeable trend of heterogeneous patterns of melting rate in the meridional direction is not observed." I think this sentence is saying that the melt rate is homogenous? But it is a bit unclear, please consider rephrasing.

L259: "Freezing rate" should this be "melting rate" as talking about sub-ice flow?

---

## Author Response (AR2)

Response to comments

We wish to thank anonymous reviewer for their valuable comments, which will help us to improve our manuscript. We addressed each of the comments in turn below. Our responses are colored by green.

**Anonymous Referee #1**

- General comments

I appreciate the effort the authors have made to revise the manuscript, including the revision of simulations under a modified set-up. The authors provided useful clarification on many of my initial questions about the simulation set-up; however key questions still remain or are raised by the new set-up that should be clarified before publication. The new dynamics examined by the manuscript are generally not explained clearly enough, particularly the driving forces behind the ice-shelf front circulation cell. Na et al. mention 3-d turbulent structures, but I'd like to see descriptions of the eddies in the IOBL (mentioned on line 177) and those in the PISW. This could help readers understand the simulated dynamics. I appreciate that the authors have offered more information about the oceanographic observations, but the manuscript could still benefit from more discussion of the relationship of LES results to both the oceanographic observations presented and melt rate observations.

- As the reviewer mentioned, there were unclear explanations (e.g. IOBL depth, eddies in IOBL, PISW and comparison with observed melt rate) for our claimed physics. We have amended and added the explanations to clarify the structures and mechanism of these physics.

Added part for the eddies in the PISW and the IOBL:

Figure 2 - In this study, we determined the IOBL region to the depth where the heat flux was 5% of maximum heat flux near the ice shelf base as IOBL physics is analogous to the atmospheric boundary layer (Derbyshire, 1990). Detailed analysis of the IOBL top is discussed at later analysis for the vertical heat flux profile.

Figure 9 – As shown in Figure S4 which depicts the vertical buoyancy flux profiles, PISWs in both weak and strong turbulence cases have a stabilizing effect because of its positively buoyant characteristic. Moreover, PISW in the weak turbulence case has more buoyancy than that in the strong turbulence case because of the momentum difference between the PISWs in the two cases. Subsequently, we determine the PISW top (297 m). Combining the regions where the meltwater and its stabilizing effect dominated with the region where heat entrainment by turbulence was vigorous (5% of maximum heat flux), the IOBL top is determined.

Major revisions:

Methodological questions/concerns that should be addressed in the manuscript text:

● Lines 135-139 do not provide enough detail on how the theoretical velocity profiles are utilized in the model and how this relates to the solution of friction velocity. These are really important details for understanding the momentum fluxes in the ice-ocean boundary layer, the validity of PISW results, and what the steady-state friction velocity means in Figure 2.

- We have amended these phrases to clarify how we used these profiles.

Modified part:

Based on the power-law assumption of turbulent boundary layer flow ($U=U_t (z/z_0)^{(1/n)}$), different velocity profiles were composed via different power-law indices, n = 3 (weak turbulence), 4, 5 and 7 (strong turbulence) to resolve the turbulence intensity within IOBL (Irwin, 1979; Kikumoto et al., 2017). These velocity profiles for the four different cases were used at the initialization of the flow field and inlet boundary condition. The freestream (geostrophic) velocity $U_t$ was set as 0.06 m s-1, based on in situ observation near the ice front. The simulation dimensions were 3456 m × 3456 m × 864 m in the x, y, and z directions, respectively. For the simulations, a grid of 288 × 288 × 144 cells was used with a 12m horizontal grid and a 6 m vertical grid with a surface roughness ($z_0$) of 0.005m (Gwyther et al., 2016).
* * *
● I'm troubled by the momentum and heat flux profiles shown in Figure 9. These profiles appear to show that momentum and heat fluxes go to zero at the ice shelf base, implying that there is negligible melting and drag. I found the text addressing these fluxes (paragraph starting on line 279) hard to understand. Can you also relate these fluxes to the spatial evolution of PISW and IOBL as they are advected (i.e., are they gaining or losing heat or momentum)?

- Fluxes at first grid are quite small because the first grid from the ice shelf base was interfacial (right near surface) grid. To examine depths of PISW and IOBL, we have added the dot-line in heat flux profile of Figure 9. PISW (above the 297 m) was always gaining heat and momentum from lower region of IOBL. We have amended this paragraph to clarify the determination of the IOBL top, PISW scale and heat entrainment with Figure S4 (buoyancy flux).

Modified paragraph:

Figure 9 shows the vertical profiles of the vertical fluxes of momentum and heat beneath the ice shelf. As shown in Figure S4 which depicts the vertical buoyancy flux profiles, PISWs in both weak and strong turbulence cases have a stabilizing effect because of its positively buoyant characteristic. Moreover, PISW in the weak turbulence case has more buoyancy than that in the strong turbulence case because of the momentum difference between the PISWs in the two cases. Subsequently, we determine the PISW top (297 m). Combining the regions where the meltwater and its stabilizing effect dominated with the region where heat entrainment by turbulence was vigorous (5% of maximum heat flux), the IOBL top is determined. In the strong turbulence case, the vertical momentum flux is negative and its maximum is located within IOBL (IOBL top - 319

m depth). This implies that the momentum entrainment from the sub-ice shelf plume to IOBL is effective, having large heat entrainment. However, the depth of maximum negative flux in the weak turbulence case is located at 347 m, slightly away from the IOBL. This difference causes the difference in heat flux magnitude at the PISW top. Negative heat flux at 320–400 m depths denotes that some of the entrained heat by the intrusion of the outer ocean is transferred to the downward direction. For steady basal melting, positive heat flux has to remain within the IOBL through flow advection penetrating the stratified IOBL. The maximum positive heat flux for the weak and strong turbulence cases is 138, 213 W m$^{-2}$, respectively, with a 54% difference. This difference is comparable with a difference (66%) in the melting rate near the ice front, confirming that basal melting is proportional to the amount of heat flux and entrainment by flow advection penetrating the stratified IOBL.
* * *
● The calculation of the freezing rate in the open ocean is not included in the methods. Is it permitted only at the surface or throughout the water column? There should be associated caveats in the Methods and Discussion about potential frazil ice effects not considered in your simulations, with citations to existing literature on frazil ice effects.

- Freezing rate was also calculated by same equations of melting rate with different ambient values of temperature and salinity. It is flux boundary at sea surface (permitted only at the surface). We have added the part about potential frazil ice effects in the Discussion section.

Added phrase:

These fluxes for the melting and the freezing rates were applied at the first grid from the ice shelf base or sea surface.

Furthermore, the effect of frazil ice dynamics (e.g. crystal growth rate, nucleation and gravitational removal) in sea surface or marine ice should be investigated because the change in plume characteristics and amount of temperature and salinity is highly related with frazil ice dynamics (Galton-Fenzi et al., 2012; Rees Jones and Wells, 2018).
* * *
● You imply on line 329 that you aren't using a dynamic SGS model but on line 110 you have included a dynamic SGS equation.

- In this study, we used the Deardorff SGS model with constant model coefficient ($C_m$=0.1) and mixing length. We have amended this part to clarify this.

Modified phrase:

where $l$ is the turbulent mixing length (depends on wall distance, grid spacing and stratification), $\Delta$ is the length scale of the filter and $\rho_\theta$ is potential density.

● The thermodynamics of the ice-shelf front are not addressed. Do you allow lateral melting?

- In this study, we did not consider lateral melting of ice front with assumption that lateral melting is not significant. Thermal driving at lateral melting is low because outer ocean temperature is comparable with freezing temperature at depth from 0 m to 140 m, and existence of PISW at depth from 140 m to 280 m. We have added this to the methodology section.

Added phrase:

In this study, the lateral melting at the ice shelf front is not included based on assumption that the lateral melting is negligible because of extremely low thermal driving at the lateral side of the ice shelf front.
* * *
The introduction of an ice-front circulation cell warrants further explanation of these dynamics than is currently included in the manuscript. On line 187, the authors write "the development of this circulation is mainly induced by the downward force of salt flux by sea ice formation and the shear stress of sub-ice shelf plume." How does sea-ice formation relate to downwelling? I'd expect convective mixing. What role does sub-ice shelf plume momentum play? Since winds are excluded, how might the results change if winds were included? Is the hypothesized circulation cell compatible with observed sea ice advection patterns? How is this similar to or different from the role that ice-shelf meltwater plays in this study: Malyarenko, A., Robinson, N. J., Williams, M. J. M. & Langhorne, P. J. 2019. A wedge mechanism for summer surface water inflow into the Ross Ice Shelf cavity. Journal of Geophysical Research: Oceans. 10.1029/2018JC014594

- As the reviewer mentioned, downward force is induced by convective mixing at local salt maximum (x = 1766 m (n=3), 1688 m (n=7)). To clarify this, we added the concept of inner ice front circulation near ice front. Between inner and outer circulations, there is local salt maximum. Those circulations share the downward force of the salt flux. The role of sub-ice shelf plume momentum is momentum transport from plume to upper layer via shear stress, and this shear force is contributing to the formation of outer ice front circulation. Inner circulation is induced by upwelling of PISW plume and salt flux and outer circulation is induced by salt flux and shear force by sub-ice shelf plume. These characteristics can be observed in circulation-stretching direction. We cannot find sea ice advection patterns which is corresponding to our simulation domain scale.

- As the reviewer mentioned, the inclusion of wind stress can affect the scale and magnitude of circulations. The effect of wind stress will be considered in future study because our objective in this study is to examine the detailed physics in the sub-ice shelf environment.

- In terms of meltwater accumulation at the sea surface near the ice front (it is clearly observed in temperature contour of Figure 3), our findings can be a supportive evidence for the mechanism of wedge formation and a thin layer of local meltwater outflow in basal melting input. However, we think that ice shelf front ablation (lateral melting) is not significant in ocean conditions of our study. We have added this to discussion section.

Modified phrase:

Figure 3 - In the ocean region, velocities in cases with weak and strong turbulence had similar patterns for two ocean circulations in the upper ocean region (0–280 m depth). In this study, we refer to these circulations as the "ice front circulations". Since we did not impose the wind effect at the top boundary, we can conclude that the development of outer circulation is mainly induced by the downward force (convective mixing) of the salt flux by sea ice formation as well as the shear stress by the momentum difference between the upper region and sub-ice shelf plume. Moreover, the development of the inner circulation is mainly due to the upwelling of the buoyant water and the downward force of the salt flux. Thus, the inner circulation is stretched in the vertical direction, whereas the outer circulation is stretched in the horizontal direction. The downward force that the two circulations share pushes the sub-ice shelf plume, moving the stratification line (280 m depth) near the ice shelf to about 350 m depth.

Discussion - Moreover, PISW upwelling and its accumulation near the sea surface could evidentially support the wedge formation mechanism proposed by Malyarenko et al. (2018).
* * *
The relationship between LES results and oceanographic observations also warrants further explanation. You say (line 216) that the signature of PISW is in the CTD profiles but it is unclear what this signature is in relation to Figure 4.

- We have added the explanation for the relationship between LES results and oceanographic observation.

Added phrases:

One of them is why the sub-ice shelf plume is located below 400 m depth even though the ice shelf base is located at 280 m depth. The LES results show that the downward force of the salt flux and the development of the inner ice front circulation (Figure 3) push the stratification line near the ice shelf. Second is the existence of relatively low-temperature water (–1.96 °C) at 100 m depth. This feature can be explained by the PISW upwelling process. Through the comparison of quantity and its characteristics, we conclude that the LES results are similar to the in situ observations of oceanic environments, in terms of the physical process of ocean circulation and the magnitude of the main variables.
* * *
The agreement between the simulated ice-shelf melt rate distribution and the observed melt rate distribution also needs to be discussed.

- We have added the part for comparison between observed melt rate (Wray, 2019) and melt rate obtained by our LES result.

Added phrases:

The melt rate for the strong turbulence case is quite low (1.74 fold) compared to the observed melt

rate (0.42 m yr$^{-1}$) proposed by the study of Wray (2019), which is about basal channel near the NIS front. However, the melt rate in our LES results is comparable to the observed melt rate, considering the change of heat transfer coefficient by thermal driving and the difference of thermal driving (0.032°C in our LES simulations and 0.14°C in the study of Wray (2019)). The heterogeneous patterns of melting rate in the meridional direction (parallel direction to the ice front) are not noticeable.
* * *
The explanation for heterogeneous PISW upwelling is unclear to me. I think it would help to see a planar view of ice-shelf cavity circulation. I wonder if the boundary conditions imposed may be influencing the circulation to a greater extent in the high turbulent shear case.

- As the reviewer recommended, we have examined the horizonal contour to find the cause of PISW upwelling and heterogeneous freezing pattern. As shown in new Figure S2, there are PISW layer induced by PISW upwelling and large scale disturbance of outer ocean. In weak turbulence case, strong PISW layer with strengthen inner ice front circulation block the intrusion of outer ocean intrusion. However, weak PISW layer in strong turbulence case cannot block the intrusion of outer ocean with baroclinic eddy, causing a heterogeneous pattern of PISW and freezing pattern. We have amended the part for heterogeneous pattern with new Figure S2.

Modified phrase:

This feature is highly related to its perturbation scale of the outer ocean (Rossby radius of deformation) and PISW upwelling (Figure S2). In the weak turbulence case, PISW upwelling occurs along the ice front edge, comprising a strong, narrow PISW layer near the ice front with strengthened inner ice front circulation and this blocks an intrusion of the outer ocean with baroclinic eddy. However, it is observed the heterogeneous patterns of PISW and freezing rate in strong turbulence case, because the PISW layer near the ice front is wide and weak, permitting the intrusion of the outer ocean with large baroclinic eddy.
* * *
189: "The circulation pushes the sub-ice shelf plume with downward forcing, making that stratification line near ice shelf is moved to about 350 m depth." Clarify the relative importance of downwelling and mixing for deepening the halocline.

- We have amended this phrase to clarify the downward force is dominant at this depth.

Modified phrase:

The downward force that the two circulations share pushes the sub-ice shelf plume, moving the stratification line (280 m depth) near the ice shelf to about 350 m depth.
* * *
204: This paragraph would be a good place to include a comparison of PISW depth and meltwater

fluxes between the 4 cases.

- Instead of this paragraph (because PISW cannot be observed clearly in Figure 4), we have added the part for PISW depth and meltwater fluxes with quantification of buoyancy flux (Figure S4) to Figure 9.

Modified phrase:

As shown in Figure S4 which depicts the vertical buoyancy flux profiles, PISWs in both weak and strong turbulence cases have a stabilizing effect because of its positively buoyant characteristic. Moreover, PISW in the weak turbulence case has more buoyancy than that in the strong turbulence case because of the momentum difference between the PISWs in the two cases. Subsequently, we determine the PISW top (297 m). Combining the regions where the meltwater and its stabilizing effect dominated with the region where heat entrainment by turbulence was vigorous (5% of maximum heat flux), the IOBL top is determined.
* * *
226: "Because the amount of the PISW in the strong turbulence case is larger than that in the weak turbulence case, its turbulence energy spectra within IOBL (297 m) is the lowest." This needs more explanation.

- With the definition of PISW depth and IOBL top, we re-plot the turbulence energy spectra within IOBL (291 m) at different zonal distance to show the spatial transition of IOBL flow following the inertial subrange slope ($k \sim$ -5/3).

Modified paragraph:

The one-dimensional turbulence energy spectra at 291 m depth (within IOBL) in the cases with weak and strong turbulence are plotted in Figure 5. Moreover, we examine different zonal locations (x = 400, 800, 1200, 1800, 2300 and 2800 m) to observe a spatial transition of the IOBL flow. For a wavenumber greater than 0.002033 m$^{-1}$ which represents the approximately 500 m scale of the energy-containing eddy, the energy spectra of the LES results follow the –5/3 slope of the Kolmogorov scale in the inertial subrange. For the cases with weak and strong turbulence, a similar trend of spatial transition of energy spectra at the IOBL region is observed. At zonal distance of 400 and 800 m, turbulence is under a fully-developed state with a similar magnitude of energy spectra. Near the ice front (1200 m) and right after passing the ice front (1800 m), the turbulence energy spectra are suppressed by the inner ice front circulation and downward forcing. At the region of the outer ice front circulation, the turbulence energy spectra exhibits the highest energy level.
* * *
243: "it is shown that the LES model adequately resolves the oceanic flow beneath the ice shelf with the proper thermohaline dynamics by the melting effect beneath the ice shelf and the freezing effect at the sea surface" This is a very general statement. What thermohaline dynamics do you have confidence in?

- As the reviewer mentioned, this phrase was unclear for thermohaline dynamics we emphasized. We have amended this phrase to clarify what thermohaline dynamics we emphasize.

Modified phrase:

The afore-mentioned analysis shows that the LES model adequately resolves the oceanic flow beneath the ice shelf with the thermohaline dynamics, such as IOBL dynamics, PISW upwelling and convective mixing by salt flux at the sea surface.
* * *
246: "there are shear forces caused by the momentum of the sub-ice shelf plume and the buoyancy force..." This is not adequately explained. What is the relationship between the stratification, momentum fluxes, and buoyancy? It's worth reminding the reader that the ice shelf based is not sloped so buoyancy does not drive mean flow.

- As the reviewer mentioned, we have amended this phrase to clarify the flat base of the ice shelf.

Modified phrase:

Since we assumed a flat base of the ice shelf in this study, the buoyant force of PISW does not accelerate the PISW at the ice shelf base. Driving forces within the IOBL flow are shear forces caused by the momentum of the sub-ice shelf plume and the stratification force (stabilizing force) caused by the PISW.
* * *
Figure 6: The x-axis appears to span the whole domain, but there should only be freezing at the sea surface in the open ocean part of the domain.

- The x-axis in Figure 6 is from 1280 m (ice front) to 3456 m. We have added the location of the ice front to Figure 6.
* * *
Figure 7: Specify whether this figure includes or excludes the region of underdeveloped turbulence.

- This figure includes the region of underdeveloped turbulence. We have added the region of underdeveloped turbulence to Figure 7.
* * *
The features we're seeing in Figure S3 should be explained in the text. I wasn't able to figure out the difference between right and left panels in this figure.

- To explain the heterogeneous pattern of PISW, freezing rate, we have removed previous Figure S3 and have added the new Figure S2 about the different scales of baloclinic eddy and Rossby radius of deformation. Moreover, we have added the explanation part for this.

Modified Figure S2 and explanation part:

[Figure]

**Figure S2.** xy distribution of meridional velocity at 3 m depth to estimate baloclinic eddy and Rossby radius of deformation by Coriolis force. Calculation for Rossby radius of deformation is based on depth-averaged buoyancy frequency and depth (scale) between the sea surface and IOBL top.

This feature is highly related to its perturbation scale of the outer ocean (Rossby radius of deformation) and PISW upwelling (Figure S2). In the weak turbulence case, PISW upwelling occurs along the ice front edge, comprising a strong, narrow PISW layer near the ice front with strengthened inner ice front circulation and this blocks an intrusion of the outer ocean with baroclinic eddy. However, it is observed the heterogeneous patterns of PISW and freezing rate in strong turbulence case, because the PISW layer near the ice front is wide and weak, permitting the intrusion of the outer ocean with large baroclinic eddy.
* * *
317: It sounds as if you're saying that PISW is transferring momentum to IOBL in the ice-shelf cavity. Where is this momentum coming from?

- That phrase was wrong. We have amended this phrase.

Modified phrase:

High turbulence intensity causes strong momentum transfer, resulting in increased melting and high-speed currents within the IOBL.
* * *
322: Explain how the circulation is similar to centrifugal overturning.

- In our study, we observed the PISW upwelling (gravitational instability) with its baroclinic eddy (centrifugal instability), showing that baroclinic eddy and its vorticity is corresponding to PISW momentum and Ekman layer. These mechanisms are similar with mechanism proposed by Naviera

Garabato (2017). We have amended this phrase, clarifying the specific mechanisms proposed by Naviera Garabato (2017).

Modified phrase:

Similar to the observation of Naveira Garabato et al. (2017), we also observe the PISW upwelling (gravitational instability) and the development of its baroclinic eddies (centrifugal instability by density gradient) with a strong Ekman layer.
* * *
330: How will the findings be used to interpret observations?

- In revised manuscript, we have added the part for the relationship between some features of the observations and how we explain these features. We have amended this phrase to clarify how our findings can be used to interpret features of observations.

Modified phrase:

One of them is why the sub-ice shelf plume is located below 400 m depth even though the ice shelf base is located at 280 m depth. The LES results show that the downward force of the salt flux and the development of the inner ice front circulation (Figure 3) push the stratification line near the ice shelf. Second is the existence of relatively low-temperature water (–1.96 °C) at 100 m depth. This feature can be explained by the PISW upwelling process. Through the comparison of quantity and its characteristics, we conclude that the LES results are similar to the in situ observations of oceanic environments, in terms of the physical process of ocean circulation and the magnitude of the main variables.

The main findings and claimed mechanism of this study can be used to fill the gap in the sub-ice shelf cavity observation.
* * *
Minor revisions:

23: "which in turn slows sea level rise" This can be misleading because the rate of sea level rise may increase when ice shelves are removed but not necessarily when the ice sheet reaches a new equilibrium without the ice shelf.

- We have amended this phrase to avoid the misleading.

Modified phrase:

One of the important roles of ice shelves in controlling the mass balance of the AIS is to hinder the flow of inland ice into the ocean to prevent sea level rise (Holland et al., 2020).
* * *
The authors provided more detail about the CTD and ADCP data collection, which was appreciated.

One remaining detail that would be useful to the reader is the distance between the ice front and the observations.

- We have added information for the distance between ice front and the observation.

Added part:

The observation location is approximately 1 km away from the ice front.
* * *
Specify t* in hours.

- $t^*$ in four cases is listed up in Table 2. We have the reference of Table 2 in this paragraph.

Added part:

Table 2 presents these friction velocities and the large-eddy turnover times for the four cases.
* * *
Thank you for clarifying the velocity orientation in this revision. I think it's also worth pointing out to the readers that the zonal velocity is perpendicular to the ice-shelf front geometry (especially in the caption for Figure 3), even though this can be seen in the new figure.

- As the reviewer mentioned, we have added the part for pointing out that the zonal velocity is perpendicular direction to the ice front. Moreover, we have added this in the caption for Figure 3.

Added phrase:

After passing the ice shelf, this high-speed current flows in the perpendicular direction to the ice front.

(Caption for Figure 3) In these contours, the zonal direction is perpendicular to the ice shelf front.
* * *
128: I think the notation should be S(z_1) rather than Sa(z_1) since you use S_b.

- As the reviewer mentioned, we have amended the notation.
* * *
261: The way this is written, there is an apparent inconsistency in the strong turbulence case that it has the smallest mean shear gradient, highest TKE, yet you say that TKE production is proportional to mean shear gradient and turbulent shear stress.

- Turbulence kinetic energy production ($-u'w'\frac{\partial u}{\partial z}$) is highest in strong turbulence case. Therefore, we can know that turbulent shear stress ($-u'w'$) is highest, showing that turbulent shear stress has a larger portion of turbulent kinetic energy production than mean shear gradient, because mean

shear gradient ($\frac{\partial u}{\partial z}$) is smallest in strong turbulence case. We have amended this phrase to clarify this.

Modified phrases:

The strong turbulence case displays the smallest mean shear gradient but the largest turbulence intensity, whereas the weak turbulence case has an opposing features. Since the turbulence kinetic energy production is proportional to the mean shear gradient and turbulent shear stress (Pope, 2000), turbulent shear stress is the highest in the strong turbulence case, showing that the turbulent shear stress has a large portion of turbulent kinetic energy production.
* * *
300 "we used the LES model with proper boundary conditions" "proper" isn't appropriate here, as it is quite subjective.

- We have amended this word to *in-situ* based boundary conditions
* * *
Section 4.2. It would be helpful to mention the temperature difference between PISW as it exits the cavity and the sea surface freezing point.

- We have added the temperature difference between interfacial temperature (not freezing temperature) and PISW at the ice shelf base because melt rate is also calculated based on interfacial temperature.

However, the melt rate in our LES results is comparable to the observed melt rate, considering the change of heat transfer coefficient by thermal driving and the difference of thermal driving (0.032°C in our LES simulations and 0.14°C in the study of Wray (2019)).
* * *
319: "showing that this result is in agreement with the previous study of Jenkins (2016)" This is too general. It's simply that both this study and Jenkins have Ekman layers below the ice shelf, right?

- We have amended this phrase, referring the specific case and physics in Jenkins (2016).

Modified phrase:

Furthermore, a turbulent Ekman layer developed in all cases, showing that these classical Ekman layers are similar to the non-inclined ice shelf case of Jenkins (2016) in terms of the steady Ekman layer independent of thermal driving broadening.
* * *
327: "constant turbulent coefficients in SGS model" Which coefficients? I thought you were using a dynamic SGS model.

- We have amended this phrase to avoid misleading.
* * *
335: It's unclear what effects you'll be examining with the "vertical distribution of pressure"

- We have removed these words.
* * *
336: "With better understanding of various parameters on basal melting" Which parameters?

- Parameters mean afore-mentioned important factors. We have amended this phrase.

Modified phrase:

A better understanding of the effect of important factors on basal melting and its meltwater dynamics will help in the improvement of the parameterizations (e.g. vertical mixing within the IOBL and sea-ice formation and behavior) in the regional ocean model.
* * *
The schematic diagram (Figure 10) is a great addition to the manuscript. I do find it somewhat confusing to include katabatic winds in the schematic when they do not play a role in your explanation of the dynamics, particularly as they appear to be opposed to the ice front circulation pattern.

- As the reviewer mentioned, katabatic winds can be somewhat confusing. We have removed this.
* * *
There are several places where the meaning of the text is unclear:

16: "In the strong turbulence case, there are distinct features in basal melting and flow characteristics." This is too vague.

- As the reviewer mentioned, this phrase is too vague. We have amended this phrase to clarity what we observe.

Modified phrase:

In the strong turbulence case, distinct features are present in the high momentum of meltwater with a strong Ekman layer and the large scale of the baroclinic eddies.
* * *
30 "The driving forces for basal melting in cold water cavity are shear force by tidal mixing and the thermohaline process by sea ice formation" Both of these forcings need more introduction and explanation here.

- We have added more explanation with reference in this phrase.

Modified phrase:

Shear force generated by tidal mixing and the thermohaline process during sea ice formation are the basal melting driving forces in cold water cavity (e.g., high salinity shelf water), whereas the intrusion of circumpolar deep water (CDW), which is the water well above the local freezing temperature, is the main driving force for basal melting in the warm-water cavity (Davis and Nicholls, 2019; Jacobs et al., 1992; Yoon et al., 2020).
* * *
35 "Therefore, because driving forces from the ocean and opposite forces by the meltwater merge within the boundary layer" I don't know what is meant here. What forces?

- We have amended this phrase to clarify the forces for IOBL.

Modified phrase:

The ice shelf–ocean boundary layer (IOBL), which is the boundary layer (meters to tens of meters) right beneath the ice shelf, is difficult to investigate because the shear forces from the ocean and stabilizing force of the meltwater combine within the IOBL (Begeman et al., 2018; Naveira Garabato et al., 2017).
* * *
43: "Similar features for weak stratification" Which features are you referring to?

- We refer to moderate melt rate with low thermal driving.

Modified phrase:

Similar melt rates for weak stratification were also observed beneath the Fimbul and Ross ice shelves (Arzeno et al. 2014; Hattermann et al., 2012).
* * *
48: "controlling the shear impact of its momentum" Please clarify

- As the reviewer mentioned, this phrase was vague. We change this to "Occurring the shear by buoyant moving of meltwater".

Modified phrase:

Positively buoyant sub-ice shelf plumes created near the grounding line can affect the stratification and heat entrainments within the IOBL; they yield the shear because of the buoyant moving of meltwater near the grounding line and ice shelf front (Hewitt, 2020; Holland and Jenkins, 1999).
* * *
57: "In order to find out the effects of various forcing clearly" Please specify which forcings

- We have amended this phrase to specify which forcing.

Modified phrase:

To clearly determine the effects of various forcing (e.g. shear, stabilizing force of the meltwater and buoyant moving of meltwater), independent experiments or observations for the forcing are needed.
* * *
185: "velocities in two cases" Which two cases?

- "Cases with weak and strong turbulence" We have amended this phrase.
* * *
211: "upper streamwise direction" Would be more clear to put in terms of zonal/meridional or parallel/perpendicular to ice front

- We have amended this word to perpendicular direction to ice shelf front.
* * *
302: "Additionally, we set to ambient values" set what to ambient values?

- We have amended this without "to".
* * *
356: "it means that stratified forcing by PISW has a nonlinear feature for flow shear by strong turbulence"

- We have amended this phrase without "nonlinear".

Modified phrase:

However, the stratification intensity in the four different turbulence cases did not exhibit a distinct trend, denoting that the stratified forcing by PISW varies according to the flow shear caused by turbulence.
* * *
332: "this study can be improved by comparing LES results with observations and their feedback" "and their feedback" is unclear.

- We have added the example in this phrase

Modified phrase:

If direct observation for the IOBL flow structures and turbulence characteristics in the sub-ice shelf environment is available, this study can be improved by comparing LES results with observations and their feedback (e.g. correction of ambient values and transfer coefficients).

There are also several places where the grammar needs revision:

9. "but there is a poor understanding of the fluid dynamic, thermohaline physics of the IOBL flow"

- We have amended this phrase.

Modified phrase:

Ice melting beneath Antarctic ice shelf is caused by heat transfer through the ice shelf–ocean boundary layer (IOBL); however, our understanding of the fluid dynamic and thermohaline physics of the IOBL flow is poor.
* * *
11: "velocity's theoretical profile" >> "theoretical profile for velocity"

- As the reviewer mentioned, we have amended this phrase.
* * *
28: "The sub-ice shelf oceanic environment can be divided into broad classifications" >> "The sub-ice shelf oceanic environment can be divided into two classes"

- As the reviewer mentioned, we have amended this phrase.
* * *
38: "physics in ice shelves" >> "physics below ice shelves"

- As the reviewer mentioned, we have amended this phrase.
* * *
69: "Nansen Ice Shelf (NIS; cold-water cavity)" >> "Nansen Ice Shelf (NIS), a cold-water cavity,"

- As the reviewer mentioned, we have amended this phrase.
* * *
72: "while remaining thermohaline forcing by the melting and freezing"

- We changed this phrase to clarify our forcing.

Modified phrase:

For consistency in experiments, we considered different turbulence state while keeping the same thermohaline forcing by the melting and freezing.
* * *
177: "are highly fluctuated in" >> "greatly fluctuated during"

- As the reviewer mentioned, we have amended this part.

178: "As turbulence within IOBL is stronger, the magnitude of fluctuation is larger."

- We have amended this phrase, referring large-scale eddy.

Modified phrase:

As the turbulence near the IOBL is stronger, the large-scale eddy corresponding to the Rossby radius of deformation is larger (Figure S2).
* * *
189: "Noticeable difference between the two cases" >> "A noticeable difference between the two cases"

- As the reviewer mentioned, we have amended this phrase.
* * *
194: "with different momentum along to streamwise direction in two cases."

- We have amended this phrase.

Modified phrase:

These momentum differences in the two cases mainly affects the magnitude and scale of the ice front circulations.
* * *
307: "for resolving the IOBL and oceanic flow in reality" >> "for simulating the IOBL and oceanic flow more realistically"

- As the reviewer mentioned, we have amended this part.
* * *
**Anonymous Referee #2**

General comments:

This study uses large-eddy simulations (LES) to examine the effect of turbulence in the ice-shelf-ocean boundary layer (IOBL) of Nansen Ice Shelf. The simulations are based on recent observations of ocean conditions near the Nansen Ice Shelf. The dynamics are forced by a neutrally buoyant plume moving beneath the ice shelf and penetrating into the open ocean. Key results are the basal melt rate of the ice shelf and the freezing occurring at the ocean surface (modelling sea ice formation), both of which increase when the plume is more turbulent. The authors also attempt

to discuss some interesting heterogenous structures in terms of Ekman layer dynamics, but I did not follow this explanation as it stands.

The study is of some interest and has been improved from the previous iteration. The new simulations have an idealised velocity input condition to model a plume with four different levels of turbulence. The temperature and salinity input conditions are broadly based on observations taken further away from the ice shelf. The scientific premise (what happens when turbulence is increased?) is much clearer. There are few LES of sub-ice shelf flow, in particular with the aim to match observations. I think that the study has potential for publication, but I have some comments that I would like to see addressed first.

Specific comments:

1. The authors vary only the shape of the input velocity profiles across the four runs. As the set-up is so constrained (in terms of setting specific input profiles of velocity, temperature and salinity) I wonder what portion of the results are caused by the input conditions versus the dynamics in the system. What do the authors expect would happen if there was no basal melt? Would there still be plenty of mixing in the sub-ice shelf flow, and would we still see the increase in surface freezing at the ice edge? Similarly, if the surface freezing was turned off, how would this change the outflowing plume dynamics?

- As input conditions are varied, shear force and turbulent intensity within IOBL can be changed, having change in a vertical momentum flux and vertical heat transport. If there is no basal melt, similar trend is also observed, in terms of momentum and heat flux beneath the ice shelf. However, if there is no PISW, PISW upwelling is not occurred and of the shape and the magnitude of ice front circulations are changed. Similarly, if there is no surface freezing, ice front circulation can be shrink or disappear because the absent of downward forcing and convective mixing.

We have amended the figures and explanations to clarify these dynamics by thermohaline process of melting or freezing.

Modified paragraph for Figure 3:

In the ocean region, velocities in cases with weak and strong turbulence had similar patterns for two ocean circulations in the upper ocean region (0–280 m depth). In this study, we refer to these circulations as the "ice front circulations". Since we did not impose the wind effect at the top boundary, we can conclude that the development of outer circulation is mainly induced by the downward force (convective mixing) of the salt flux by sea ice formation as well as the shear stress by the momentum difference between the upper region and sub-ice shelf plume. Moreover, the development of the inner circulation is mainly due to the upwelling of the buoyant water and the downward force of the salt flux. Thus, the inner circulation is stretched in the vertical direction, whereas the outer circulation is stretched in the horizontal direction. The downward force that the two circulations share pushes the sub-ice shelf plume, moving the stratification line (280 m depth) near the ice shelf to about 350 m depth. A noticeable difference between the two cases is observed near the ice front and beneath the ice shelf. At depths from 280 to 320 m (IOBL region), high zonal

velocity beneath the ice shelf is observed in the strong turbulence case (n = 7). After passing the ice shelf, this high-speed current flows in the perpendicular direction to the ice front.

Modified Figure 10:

[Figure]

2. I disagree with (or perhaps I did not understand) the Ekman layer explanation for the heterogeneity seen in the strong turbulence cases. I agree that stronger velocities beneath the ice shelf could lead to a stronger Ekman layer, but why wouldn't this be a homogenous response underneath the whole ice shelf? What is constraining the length scale of the heterogeneity? (E.g. could it alternatively be a baroclinic eddy with a Rossby radius of deformation?) I would appreciate more discussion on these points and an in-depth explanation of the physical mechanisms.

- As the reviewer mentioned, there was no explanation for length scale of the heterogeneous pattern of PISW and freezing. As the reviewer recommended, we examined meridional velocity (parallel to ice front) at sea surface and the baroclinic eddy with a Rossby radius of deformation based on buoyancy frequency, depth (310 m) between sea surface and IOBL and Coriolis parameter. As new Figure S2, wave-like features of baloclinic eddy are observed in meridional velocity and its scale is larger in strong turbulence case. Moreover, the layer of upwelling PISW near the ice front is wide and weak (more advective) in strong turbulence case. Therefore, we conclude that strong and narrow PISW layer blocks the intrusion of outer ocean by the baroclinic eddy in weak turbulence case, whereas weak, wide PISW layer cannot block the intrusion of outer ocean, having a heterogeneous pattern which is corresponding to Rossby radius of deformation.

We have added new Figure S2 and related explanations.

Modified Figure S2:

[Figure]

**Figure S2. xy distribution of meridional velocity at 3 m depth to estimate baloclinic eddy and Rossby radius of deformation by Coriolis force. Calculation for Rossby radius of deformation is based on depth-averaged buoyancy frequency and depth (scale) between the sea surface and IOBL top.**

This feature is highly related to its perturbation scale of the outer ocean (Rossby radius of deformation) and PISW upwelling (Figure S2). In the weak turbulence case, PISW upwelling occurs along the ice front edge, comprising a strong, narrow PISW layer near the ice front with strengthened inner ice front circulation and this blocks an intrusion of the outer ocean with baroclinic eddy. However, it is observed the heterogeneous patterns of PISW and freezing rate in strong turbulence case, because the PISW layer near the ice front is wide and weak, permitting the intrusion of the outer ocean with large baroclinic eddy.
* * *
3. I think more can be done to compare against other cases, in particular Naveira Garabato et al. (2017). (Side note: this study seemed to be missing from the citation list in the manuscript.) The only comparison with the Naveira Garabato et al. study was L321 "These physics with the Ekman layer and the upwelling behavior of PISW are similar to centrifugal overturning instability and lateral shear proposed by Naveira Garabato et al. (2017)." I was confused by this. Are the authors saying that the mechanisms are similar? And if so, how similar and what are the differences? Again, I would appreciate a more in-depth explanation of the exciting phenomena that is noted in this study. I think this would really strengthen our understanding of the phenomena we might expect to see sub-ice shelf and on the ice edge.

- As the reviewer mentioned, previous comparison for the Naveira Garabato et al. study was too vague. We have amended this part to clarify that our mechanism is similar with that of Naveira Garabato et al. study.

Modified phrase:

Similar to the observation of Naveira Garabato et al. (2017), we also observe the PISW upwelling (gravitational instability) and the development of its baroclinic eddies (centrifugal instability by density gradient) with a strong Ekman layer.
* * *
L11: In the abstract "…we impose velocity's theoretical profile varying the power-law index." Consider rephrasing as I did not know what "velocity's theoretical profile" meant when I first read through. Highlight that the velocity profile is varied and therefore turbulence is also changed.

- As the reviewer mentioned, we have amended this phrase.

Modified phrase:

To resolve the different turbulence states, we impose the theoretical profile of velocity at the turbulent boundary layer by varying the power-law index.
* * *
L35: "Therefore, because driving forces from the ocean and opposite forces by the meltwater merge within the boundary layer (meters to tens of meters) right beneath ice shelf, which is known as the ice shelf–ocean boundary layer (IOBL), we have to investigate the IOBL flow and its structure to reveal the basal melting physics in ice shelves (Holland et al., 2020)." This was a long sentence and a bit unclear. Please consider rephrasing.

- As the reviewer mentioned, we have divided and amended this phrase.

Modified phrases:

The ice shelf–ocean boundary layer (IOBL), which is the boundary layer (meters to tens of meters) right beneath the ice shelf, is difficult to investigate because the shear forces from the ocean and stabilizing force of the meltwater combine within the IOBL (Begeman et al., 2018; Naveira Garabato et al., 2017). Therefore, the IOBL flow and its structure need to be investigated with a turbulence resolving model to reveal the basal melting physics below ice shelves (Jenkins, 2016; Holland et al., 2020).
* * *
L135: The power-law equation $U=U_t (z/z_t)^{(1/n)}$. Firstly, I assume the x signified multiply and not x-direction (was written in the manuscript as $U=U_t$ x $(z/z_t)^{(1/n)}$)? Secondly, is $z_t$ the surface roughness? Thirdly, when I quickly plotted this power law I got something that looked different to the Figure 2 inset. So how does this equation directly relate to the profiles in Figure 2 inset? Is there some normalisation coming in to play? Finally, are there other studies (apart from Irwin 1979) that have used these profiles? I am wondering if the justification for the plume shape can be strengthened by citing some more studies that use these profiles.

- Firstly, x means multiply as the reviewer mentioned. Secondly, z_t (z_0 in revised manuscript) is surface roughness. Thirdly, we had a confusion for plotting power law profiles. We have changed the inset profile in Figure 2. Finally, we have added the reference which is about power law fitting study for stable boundary layer.

Modified phrases:

Based on the power-law assumption of turbulent boundary layer flow ($U=U_t\,(z/z_0)^{(1/n)}$), different velocity profiles were composed via different power-law indices, n = 3 (weak turbulence), 4, 5 and 7 (strong turbulence) to resolve the turbulence intensity within IOBL (Irwin, 1979; Kikumoto et al., 2017). These velocity profiles for the four different cases were used at the initialization of the flow field and inlet boundary condition. The freestream (geostrophic) velocity $U_t$ was set as 0.06 m s-1, based on in situ observation near the ice front. The simulation dimensions were 3456 m $\times$ 3456 m $\times$ 864 m in the x, y, and z directions, respectively. For the simulations, a grid of $288 \times 288 \times 144$ cells was used with a 12m horizontal grid and a 6 m vertical grid with a surface roughness ($z_0$) of 0.005m (Gwyther et al., 2016).

L152: The wind effect is excluded in the simulations by setting the surface velocity to zero, even though observations showed a non-zero value of wind. In the reply to reviewers there was some justification for the zero velocity condition, would it be possible to have the brief explanation included in the manuscript also?

- As the reviewer mentioned, we have added the brief explanation for zero velocity at the sea surface.

Modified phrase:

The cyclic boundary condition was applied to lateral boundaries, while a Dirichlet boundary condition (Utop = 0 m s-1) was imposed on the top layer. Zero velocity at the sea surface implies the exclusion of the wind effect to examine the ocean dynamics caused by the sub-ice shelf plume solely.

Section 2: I might have missed this, but what was the surface freezing condition on temperature and salinity? Was it a Dirichlet or flux condition?

- Melting and freezing conditions in this study are flux condition based on Monin-Obukhov similarity.

L175: How was the large-eddy turnover time calculated here?

- Large-eddy turnover time is calculated by IOBL characteristic length (IOBL top) divided by friction velocity. We have added the brief explanation to this part.

Modified phrase:

The total simulation time (96 h) is normalized by large-eddy turnover time (t*) which is calculated by the scale of overturning large eddy within the IOBL divided by the friction velocity.
* * *
Figure 2: There is a lot of variability in this figure, which makes it difficult to determine whether it is in equilibrated state. Are there other results that show the simulations are equilibrated?

- As the reviewer mentioned, a lot of variability could be induced by large-scale disturbance (baroclinic eddy). We have added the phrase for explanation of this.

Modified phrase:

As the turbulence near the IOBL is stronger, the large-scale eddy corresponding to the Rossby radius of deformation is larger (Figure S2).
* * *
Results: Are there any indications of inertial waves present in the simulations, due to having Coriolis parameter (f)? I am wondering if this variability would appear on the friction velocity (Figure 2).

- Because the disturbance by baroclinic eddy (larger scale in strong turbulence) affects to IOBL dynamics, this variability of the disturbance appears on the friction velocity. We have added this in Figure 2 explanation.

Modified phrase:

As the turbulence near the IOBL is stronger, the large-scale eddy corresponding to the Rossby radius of deformation is larger (Figure S2).
* * *
Figure 5: Caption does not seem to be consistent with legend. What is the difference between (a) and (b) figures? If it is n=3 and n=7, then why are these values also varied in the legend?

- With new definition of PISW depth and IOBL depth, we replot the energy spectra at different zonal distance to observe development and change of turbulence within the PISW.

Modified paragraph for Figure 5:

The one-dimensional turbulence energy spectra at 291 m depth (within IOBL) in the cases with weak and strong turbulence are plotted in Figure 5. Moreover, we examine different zonal locations (x = 400, 800, 1200, 1800, 2300 and 2800 m) to observe a spatial transition of the IOBL flow. For a wavenumber greater than $0.002033$ $m^{-1}$ which represents the approximately 500 m scale of the

energy-containing eddy, the energy spectra of the LES results follow the –5/3 slope of the Kolmogorov scale in the inertial subrange. For the cases with weak and strong turbulence, a similar trend of spatial transition of energy spectra at the IOBL region is observed. At zonal distance of 400 and 800 m, turbulence is under a fully-developed state with a similar magnitude of energy spectra. Near the ice front (1200 m) and right after passing the ice front (1800 m), the turbulence energy spectra are suppressed by the inner ice front circulation and downward forcing. At the region of the outer ice front circulation, the turbulence energy spectra exhibits the highest energy level.
* * *
L239: Could the spatial-averaged freezing rate values also be included here? Perhaps an earlier reference to Table 2 would help (first time you reference Table 2 is in Section 4.2 Discussion)?

- As the reviewer mentioned, we have added the earlier reference to Table 2 (Figure 2 part)

Modified phrase:

Table 2 presents these friction velocities and the large-eddy turnover times for the four cases.
* * *
L262: "Because turbulence kinetic energy production is proportional to mean shear gradient and turbulent shear stress, turbulent shear forcing is highest in the strong turbulence case." Could a citation please be included for the first half of this sentence?

- We have added the reference of Pope (2000) for this.
* * *
Figure 9: There are some negative heat flux values below the positive values in the IOBL. What is causing this?

- At depth from 320 m to 400 m, negative heat flux (downward heat transport) is observed. It is from the heat entrainment by interaction of outer ocean. Entrained heat at depth from 320 m to 400 m is transferred to upward and downward direction, by in-there turbulence. We have added this explanation to Figure 9 part.

Modified phrase:

Negative heat flux at 320–400 m depths denotes that some of the entrained heat by the intrusion of the outer ocean is transferred to the downward direction
* * *
Section 4.2: There are a few different ideas discussed in this section. Please consider breaking into separate paragraphs.

- As the reviewer mentioned, Section 4.2 in revised manuscript is divided to two paragraphs.

L324: Please include a definition for the flux Richardson value. How was it calculated in the simulations?

- As the reviewer mentioned, we have added the definition for the flux Richardson number. We calculated this value using $3t^*$ averaged 3d-flow field data.

Technical corrections:

L34: "opposite forces" perhaps "opposing forces"?

- To clarify the force we consider, we have amended this phrase.

Modified phrase:

The ice shelf–ocean boundary layer (IOBL), which is the boundary layer (meters to tens of meters) right beneath the ice shelf, is difficult to investigate because the shear forces from the ocean and stabilizing force of the meltwater combine within the IOBL (Begeman et al., 2018; Naveira Garabato et al., 2017).

L70: Here and in several other places there is a "the" missing (e.g. "THE main parameter…").

- In revised manuscript, we have edited this manuscript with copyediting services of skilled editor.

L71: "For consistency in experiments, we considered different turbulence state while remaining thermohaline forcing by the melting and freezing." Was this meant to be more along the lines of: "For consistency in experiments, we considered different turbulence states while keeping the same thermohaline forcing by the melting and freezing."

- As the reviewer mentioned, we have amended this phrase.

L97: Units missing on Coriolis parameter.

- As the reviewer mentioned, we have added units of Coriolis parameter.

L124: "…was used in whole cases." Should this be "…was used in all cases."?

- As the reviewer mentioned, we have amended this part.

L183: Domain center (y=1536m) is different to that stated in Figure 3 caption (y=1728m).

- As the reviewer mentioned, we have amended this part.
* * *
L223: Wavenumber is missing units.

- As the reviewer mentioned, we have added units of wavenumber.
* * *
Figure 8, Figure 9a: Are units on (rho u v) correct? Should they be kg m^-1 s^-2?

- As the reviewer mentioned, we have amended the units of momentum fluxes in Figure 8, 9a.
* * *
L257: "Important to not is that a noticeable trend of heterogeneous patterns of melting rate in the meridional direction is not observed." I think this sentence is saying that the melt rate is homogenous? But it is a bit unclear, please consider rephrasing.

- As the reviewer mentioned, we have amended this phrase.

Modified phrase:

The heterogeneous patterns of melting rate in the meridional direction (parallel direction to the ice front) are not noticeable.
* * *
L259: "Freezing rate" should this be "melting rate" as talking about sub-ice flow?

- We have amended this part with new analysis for heterogeneous freezing pattern.

---

## Author Response (AR3)

Response to comments

We wish to thank anonymous reviewer for their valuable comments, which will help us to improve our manuscript. We addressed each of the comments in turn below. Our responses are colored by green.
* * *
**Anonymous Referee #1**

- General comments

I appreciate that the authors have made significant additions to the text since the last round. I have indicated major revisions because there are still many aspects of this work that remain unclear. Some of this needed clarification arose due to additions made since last round, and some questions remain from last round so I have tried to be more specific in my comments. My comments this round pertain mostly to the science rather than language. I acknowledge that the authors are likely non-native English speakers. Unfortunately, non-standard grammar and awkwardness of some of the sentences will deter some readers and in cases will contribute to confusion about the science (as they did for me), and consequently reduce the impact of the article. I recommend getting a careful read-through for grammar and awkwardness before the next round.

- In this revision round, we have double-checked the grammar and awkwardness with additional English correction service.
* * *
Given the edits to the manuscript, some reorganization is needed. For instance, much more than validation happens in Section 3.1. The paragraphs should be split in several places to divide distinct topics, as I've noted in my comments. Some places also lack adequate transitions between topics.

- Previous section 3.1 have been split in two sections (3.1 Quasi-steady, ocean environment near the ice front, 3.2 Validation of simulation results). Moreover, we have modified several places to clarify the transitions between topics.
* * *
It's clear to me now what the definition of PISW is. However, I do believe that the definition of IOBL will be misleading. I would have expected that the IOBL would include all of the PISW thickness and that the 5% heat flux definition would have been used for IOBL bottom and not its top. It's not clear to me in what sense that would be considered a boundary layer (if not related somehow to shear at the boundary) or what the bottom of the IOBL is. I might have missed it, but I don't think it's stated explicitly what you mean by sub-shelf plume. My best guess was the full-cavity depth outflow. If that's right, I think that would also be confusing for readers, as plume usually denotes flow driven by buoyancy whereas this seems to be a geostrophic flow. If you decide to keep this notation, it should be clear what these terms mean.

- As the reviewer mentioned, 5% heat flux definition have been used for IOBL bottom (not IOBL top). In this study, we defined the IOBL as the boundary layer where PISW (not sub shelf plume) and its thermodynamic impacts are dominant. In the revised manuscript, we have explained this explicitly.

There are a few instances where the choice of terminology could be improved. I'm not thrilled with the choice to call the different initial and boundary conditions different "turbulent states" as that seems to imply that there is a transition in the mode of turbulence or the kinds of instabilities that occur. There are also instances where the authors use a "downward force" framing, which doesn't reveal what specifically is influencing the relative buoyancy of water masses. See also a later comment about more specific terminology for "ice front circulations."

- As the reviewer mentioned, those terms were not clear. To clarify what we explained, we have amended those terms in revised manuscript. (1. turbulent states -> turbulence intensity, 2. downward force -> downwelling, and 3. ice front circulation -> inner and outer overturning cells)

Some aspects of the ice-shelf front circulation remain unclear. Are there convective downwellings throughout the outer cell or only at the convergence of the two cells? You show the horizontal velocity but there is no way for the readers to gauge the strength of these circulatory cells. A streamfunction plot would be the most clear but a vertical velocity plot might work if it's easily interpretable.

- Although weak downwelling motions by salt flux are throughout sea surface, distinct feature of downwelling is observed at the convergence of the two cells. To show this, we have added the vertical velocity contours in weak (n=3) and strong turbulence (n=7) cases to supplement material.

*New supplement material (Figure S5)

[Figure]

Furthermore, it's unclear in the text why you think that the outer circulation should be a cell. Why would there be upward velocity at the outermost edge of the domain given that there is a radiation boundary condition? What would set the size of this cell in the real world? (I appreciate that the authors added more discussion of length scales with respect to the ice-front circulation.)

- In this study, we have observed the negative velocity (toward ice front) at sea surface, downwelling motion at the convergence of the two cells, and positive velocity at 300 m depth. Through these values, we speculated that there is outer overturning cell. Because this overturning cell is caused by highly shear flow (sub-ice shelf plume), it is similar to reattachment flow (backward facing step) near the geometry. According to Rygg et al. (2011), reattachment length in highly turbulent (Reynolds number $> 2.0 \times 10^6$) flow of ocean is approximately 7-10 (secondary cell (inner) ~ 1.3–1.5, primary cell (outer) ~ 5.3–8.4) of geometry height. Because ice shelf thickness was 280 m and Reynolds number in this study was approximately $9.0 \times 10^6$, we can speculate 364–420 m of inner overturning cell and 1,484–2,352 m of outer overturning cell. In the revised manuscript, we have discussed the length scales of overturning cells.
* * *
I'd like to see you explain more thoroughly how these simulations relate to the real world a bit better given that understanding observations is given as the main motivation. Are your simulation conditions specific to a season? I assume the observations are from summer, so is there sea-ice freezing here year-round? How does your choice of SST of -1.9 degC relate to observations and season? If there were these two circulation cells, wouldn't you see a signature in sea ice advection including convergence and ridging or do you not expect these features to be long-lived or to migrate? Besides oceanographic observations, what might validation look like? I appreciate that the authors added a citation to basal melting observations but they say it's from a channel. Are there no satellite estimates of basal melt on broader scales? Was Nansen not included in the Adusumilli et al. 2020 dataset https://doi.org/10.6075/J04Q7SHT?

- As the reviewer mentioned, we have added the explanation for Nansen ice shelf and ocean environment in 2.1 section (observation part) of the revised manuscript. Unfortunately, we cannot find distinct features of sea ice advection to clarify the existence of overturning cells. We have compared the simulated basal melts with satellite estimate of basal melt (Adusumilli et al., 2020, Dow et al., 2021).

*Modified part in the revised manuscript:

The melt rates obtained in this study are significantly low compared to those reported by Wray (2019) (0.45–0.95 m yr$^{-1}$) and estimated via the Cryosat-2 satellite observation during 2010–2018 ($1 \pm 0.6$ m yr$^{-1}$) at the NIS ice front region (Adusumilli et al., 2020). However, the melt rate in our LES results is comparable to the observed melt rate, considering the difference in the thermal driving (0.032 °C in our LES simulations and 0.14 °C in the study of Wray (2019)). In this study, only the effect of sub-ice shelf plume (formed by HSSW) was considered, but the observations included the effects of relatively warm Antarctic surface water and sub-ice shelf plume, resulting in a difference in the thermal driving and melt rates. If the melt rate in this study is assumed to be

0.12 m yr$^{-1}$ (averaged value of 0.092 and 0.153), we can estimate that 12–25 % of the total basal melting near the NIS front is due to sub-ice shelf plumes.
* * *
I appreciate that the authors added some text pertaining to wind stress forcing, but I do believe that there should be more discussion of how this might affect the strength of the inner overturning cell since that is one of the main results of the paper.

- Because wind stress by katabatic wind reduced the shear stress between the sea surface and sub-ice shelf plume, strength, and horizontal scale of the outer (primary) overturning cell may be decreased if there is wind stress. The weakened outer cell weakens the inner (secondary) cell. However, the strength and scale of the inner overturning cell may be similar to this study because wind stress at the sea surface imposes at the upper region of the inner overturning cell. To sum up, if the wind stress effect is included, we speculate the similar scale of the inner overturning cell and decreased scale of the outer overturning cell. In the revised manuscript, we have added the discussion for wind stress and its impact on two overturning cells.
* * *
I appreciate the caveat the authors added for frazil ice dynamics, but I strongly feel that there should be more discussion (at least a few sentences) of how inclusion of frazil dynamics might change your key findings. Unless it is the case that you don't get pockets of supercooled water, in which case that should be stated.

- As the reviewer mentioned, frazil dynamics and its process highly affect plume characteristics and ocean circulation. We have discussed more about how inclusion of frazil dynamic in PISW might change inner overturning cell.
* * *
The main remaining gap in the Methods is the determination of heat and salt transfer coefficients. This is crucial to the interpretation of freezing and melting rates. It should also be made more clear in Section 2.2 why you choose the initial and inflow velocity profile you do and how it varies with depth in and outside of the cavity.

- We have added the detailed explanation for heat and salt transfer coefficients, and velocity profile of initial & inflow boundary condition.
* * *
I don't like to harp on this, but I'm still confused by the flux profiles shown in Figure 9. The sub-grid fluxes should increase at the boundary because the turbulent length scales decrease near the boundary. Instead it looks like both resolved and sub-grid fluxes go to 0 at the boundary. If I just can't see the values, perhaps an inset for the PISW would be helpful. Figure S4 also appears to show 0 buoyancy fluxes at the boundary.

- As the reviewer mentioned, SGS fluxes should increase and resolved fluxes should decrease at the boundary (go to 0 at boundary) because turbulent length scales decrease. In the previous plot, we had a mistake for the vertical dimension. SGS momentum flux has to be shifted to 1 upper point of vertical grid.
* * *
11: It is unclear how imposing a velocity profile is related to different turbulent states

- We have amended this phrase to clarify the relationship between velocity profile and turbulence intensity.
* * *
12: The flow of the abstract needs improvement. "To resolve" and "to simulate" make it unclear how these objectives are related.

- To clarify what we meant, we have amended abstract.
* * *
14: State more clearly which properties are used to validate findings.

- We have amended these phrases to clarify what we focused on validation.
* * *
16: It is unclear what the distinct features are.

- We have amended this phrase to clarify what are distinct features.
* * *
18: It is unclear how this sentence is related to the previous sentence.

- In the revised manuscript, we have removed this.
* * *
19: Writing needs improvement here.

- We have amended this phrase to clarify what we meant.
* * *
31: I think you mean shear generated by tides.

- As the reviewer mentioned, we have amended this phrase.
* * *
32: "thermohaline process" is always unclear to me. I'd prefer that you specify melting or

thermohaline stratification, mixing, or something else.

- We have amended this phrase (thermohaline process -> brine rejection).
* * *
36: I wouldn't say that these processes make it difficult to investigate, just a complex problem.

- As the reviewer mentioned, we have amended this phrase.
* * *
58-60: Improve the logical flow of these sentences

- As the reviewer mentioned, we have amended these phrases to improve the logic.
* * *
61: More realistic than what?

- This phrase was not clear. We have removed the part for realistic boundary condition in the previous phrase.
* * *
67: Citations needed.

- We have amended this phrase with additional reference.
* * *
74: I think you mean that the melting/freezing boundary condition is the same but the way you've written it makes it sound as if you impose the same melt/freeze rate in all runs.

- As the reviewer mentioned, this phrase was not clear. We have removed this phrase and added the additional phrase for simulation set up and objective.
* * *
74: Keep the language consistent. I prefer that you stick with "turbulence intensity" rather than switching to "turbulence state" for clarity.

- In the revised manuscript, we have amended all "turbulent state" to "turbulence intensity".
* * *
86: Only mentioning "refreezing" here implies there isn't melting

- Instead of refreezing, we have added frazil ice formation and basal melting.
* * *
110: It would be best to specify in the text what H_k, S_k, K_h, K_m are. I might have missed it, but there should also be an explanation for the primes.

- We have added the explanation of these terms and primes.
* * *
114: Specify z here.

- We have added the definition of z (distance from the wall).
* * *
115: Relate the filter length to the model resolution.

- We have added the definition of filter length.
* * *
Figure 1. It think it would be helpful if the velocity profile looked more like the initial or mean profile. Can you label IOBL and sub-ice plume separately?

- As the reviewer's comments, we have amended Figure 1

*Modified Figure 1

128: This is confusing since wind stresses are 0 in your simulations, right? Also, the friction velocity should be determined based on the difference in velocity of air and water at the surface. I assume you've done this but it should be stated.

- As the reviewer's comments, we have amended this phrase.

*Modified phrase:

We used $0.026$ m s$^{-1}$ of friction velocity in the calculation of thermal and salinity change by frazil

ice formation, although the effect of wind stress in momentum change was excluded to focus the relationship between sub-ice shelf plume and the development of ocean circulation.
* * *
136: It's unclear how the transfer coefficients are determined.

- As the reviewer's comments, we have added the explanation for transfer coefficients.

*Added phrases:

Based on high resolution LES study for heat and salt transfer coefficients which were described as a function of friction velocity and thermal driving, $1/\Gamma_\theta$ and $1/\Gamma_S$ at the ice shelf base were $8 \times 10^{-3}$ and $2.6 \times 10^{-4}$ for the basal melting, respectively (Vreugdenhil and Taylor, 2019). For the frazil ice formation at the sea surface, the same coefficients of the previous study of sea ice formation in polynyas were used because thermal driving in this study was comparable to its thermal driving (Heorton et al., 2017).
* * *
139: It's hard to believe that melting is negligible unless thermal driving is extremely low since IOBL is rapidly rising along the front.

- In this phrase, we have removed the part (melting is negligible). We have discussed more melting or freezing effects at lateral side of the ice front in discussion section.
* * *
140: I don't see the sea surface boundary condition described in this section. The scalar flux boundary condition is in Figure 1 but it should be stated here as well. Is the velocity boundary condition no stress/free-stream? I think it would be helpful to readers to explicitly state that wind affects scalar fluxes but not momentum fluxes.

- As the reviewer's comments, we have amended this phrase.
* * *
145: cite Figure 2

- As the reviewer's comments, we have mentioned Figure 2.
* * *
145: Mention that the profile is vertically symmetric and z is the distance from the boundary, justify that choice in relation to observations and mention the depth interval over which it's applied. It's unclear here what the initial velocity is above the ice shelf base.

- As the reviewer's comments, we have amended this phrase and added the explanation of initial velocity above the ice shelf base and no wind stress at the sea surface.

145: Discuss this choice in relation to ice-shelf cavity overturning circulation. Is there reason to believe that the baroclinic velocity is small under Nansen? If the IOBL does rise along the ice front, then the velocity structure due to the IOBL will not be reflected in the profile you implement.

145: Generating momentum through an inflow boundary condition may affect the flow differently than imposing geostrophic flow through pressure gradients. I'd like to see you at least acknowledge possible limitations.

- Because we do not know velocity structure and profile within IOBL and beneath ice-shelf cavity, we have imposed the theoretical velocity profile with parameterized melting effect. The flow started from the inflow boundary condition transited and modified under imposed forcings. Owing to issues mentioned by reviewer, we excluded the developing region in analysis of analysis. We have mentioned this limitation.

197: When after 14h is the averaging occurring?

- Last 3 t* period in all cases. We have amended this phrase.

198: This paragraph could be broken into several. I recommend describing the circulation and PISW generally, then discussing differences between the cases.

- As the reviewer mentioned, we have separated this paragraph.

198: In this paragraph there should be more references to panels within Figure 3.

- In the revised manuscript, we have added the references for Figures.

199: Use "end-member" instead of "extreme" unless you believe those n values are extreme in the sense that they are unlikely to be realized in the real world.

- As the reviewer's comments, we have amended this term.

202: Would "two overturning cells" be an appropriate description? "Ice-front circulations" and "ocean circulations" are quite general and it's strange to have circulation in the plural.

- We think that overturning cells are appropriate. As the reviewer's comments, we have amended these terms in the revised manuscript.

203: You might want to reconsider where it makes most sense to present melting and freezing fields. You mention sea ice formation without presenting Figure 6.

- As the reviewer's comments, we have amended the structure of manuscript.

205: Might be helpful to the readers to specify that the downward force due to salt flux is the negative buoyancy flux.

- As the reviewer's comments, we have amended this phrase.

207: Stratification line is unclear. Can you provide the pycnocline contour or something similar in Figure 3?

- In this contour, stratification line and its down were not observed. We have removed this phrase and discuss this at vertical profiles.

210: To me, this indicates that your simulation results are sensitive to the inflow boundary condition, even far from that inflow condition. This should be clearly stated and why this is the case should be discussed a bit more.

- In discussion, we have added the limitation of prescribed inflow condition.

214: The sentence beginning "Positive buoyancy" is a rough transition from the previous sentence. I was expecting you to elaborate on the magnitude and scale of those circulatory cells next. PISW would be better introduced in a new paragraph

- As the reviewer mentioned, we have modified the structure of manuscript.

214: Rather than the arrow for PISW shown in Figure 3, can you contour this water mass on all panels? I imagine it's mostly the blue contour in salinity but the reader would need density in order to understand which regions are positively buoyant. I would also appreciate a contour for the IOBL.

- To contour this water mass (PISW), we have added the potential density (isopycnal lines)

217: Cite Figure 7

- As the reviewer's comments, we have mentioned related Figure.

218: Outer ocean is a confusing term. Do you just mean open ocean?

- We have modified this term (outer ocean -> open ocean)

219: rephrase to avoid "-2 degC of potential temperature." It's unclear how this detail is important. It would be more meaningful in relation to the surface freezing point.

- We have amended this phrase to emphasize the lower temperature than surface freezing temperature and vigorous frazil ice formation.

221: Sentence beginning "Below 400m" and the following sentence together are unclear.

- To clarify the stratification feature and change of isopycnal line, we have amended these phrases.

223: This paragraph could also be broken into several.

- As the reviewer mentioned, previous paragraph was broken into three parts (observation interpretation, circulation and velocity, and different turbulence intensities) in the revised manuscript.

225: It's unclear at this point how varying the velocity profiles helps you evaluate the cause of the ice-front circulation patterns.

- In the revised manuscript, we have removed this phrase and added paragraph for the interpretation of observations.

227: I think you need to be more clear about what exactly the CTD, ADCP data show. Here the reader might think that the data suggest the circulation cells you simulate, but I don't think they establish those cells.

- In the revised manuscript, we have added paragraph for the interpretation of observations.

230: It's more clear in the text to use directions (toward or away from ice-shelf front) rather than positive/negative. And what is this difference (stronger in observations or simulations)?

- As the reviewer's comments, we have added the directions and the phrase for this difference.

231: "The difference is from..." How do you know this? How can you exclude wind forcing as the reason? By downward force, do you mean that the freezing rates are stronger/weaker or that the water mass properties are different and thus the relative buoyancy is different?

- As the reviewer mentioned, wind stress also a candidate for strength of overturning cell. In discussion section, we have added the phrase for the cause of underestimated strength of overturning cells.

Figure 4: Explain why simulated velocity profiles look so similar. Also, it's unclear

- We have added the explanation of similar velocity profiles in all cases.

233: Explain why the differences in ML temperature and salinity can be attributed to differences in ice-shelf melting from observations rather than differences in PISW dynamics.

- Based on contour and plot of frazil ice formation, total temperature in upper mixed layer of the open ocean in strong turbulence case is lower than that in weak turbulence case. We have amended this phrase to clarify this.

240: Here I think you're arguing that your LES results explain a feature in the observations. This should be stated explicitly. However, if this is the cause then this process has a weaker effect in the simulation than in reality because the pycnocline is not depressed enough. Do you need these large-scale circulation cells to explain the observations or would regular small-scale convective mixing from sea ice formation suffice?

Figure 4 makes it clear that the simulated thermocline is too weak relative to observations. This should be discussed.

- The pycnocline of LES results is similar to observation as shown in vertical profiles of salinity. It means that downwelling of salinity flux is enough in the LES simulation. However, weak simulated thermocline near the 350 m was observed. This is caused by weak strength of large-scale overturning cells. In the revised manuscript, we have amended the additional discussion for this.

241: If PISW upwelling can explain the temperature and salinity excursions in the mixed layer, then why don't we see it in Figure 4? I imagine this could be because the simulated profiles are time-averaged. Do you ever see such features in the instantaneous profiles? If not, why might this be?

- R-Figure 1 is the instantaneous contour of potential temperature at t = 91.26 (23.28 t$^*$). We can observe the PISW advection (excursion of temperature and salinity) to open ocean region.

R-Figure 1.

[Figure]
* * *
241: "the comparison of quantity and its characteristics" this is too vague.

- We have removed this part and added the detailed explanation of comparison.
* * *
250: This wavenumber seems too precise

- We have amended the wavenumber range.
* * *
Figure 5 would be easier to interpret if the points were colored on a linear scale with distance from the inlet boundary

- We re-plot turbulence energy spectra with linear scale of distance from the inlet boundary.
* * *
251: To say that these spectra follow a -5/3 slope feels like a stretch. Also, are you analyzing wavenumbers that are too large here for your resolution? It would be helpful to have marked the transition between resolved and sub-grid turbulence.

- With new figure for energy spectra, we have amended this explanation with grid resolution and turbulence scale.

255: How do these factors suppress turbulence?

- In the revised manuscript, we have removed this phrase.
* * *
270: What is the origin of these baroclinic eddies? Are they convective instabilities associated with salt flux or something else?

- This baroclinic eddies are caused by the gradient of meridional velocity and density difference between local salt maximum and PISW. We have added this explanation.
* * *
280: I think it would be better to say "caused by melting"

- As the reviewer mentioned, we have amended this phrase.
* * *
290-294: I found these sentences confusing. Do we only know melt rates in a basal channel? How can your thermal driving values be so different from observations?

- In this study, we only consider the effect of sub-ice shelf plume, whereas observations include the effects of Antarctic surface water and sub-ice shelf plume. In the revised manuscript, we have amended these phrases to clarify the difference between LES results and observations.
* * *
Figure 8: it's hard to tell the difference between u and v lines.

- In the revised manuscript, we have amended this figure with another shape of v.
* * *
296: This doesn't look like a high-speed current to me.

- In this part, we have amended this phrase without 'high'.
* * *
Figure 9: The Ekman length scale looks super large.

- Ekman length can be identified in meridional velocity profile of Figure 8. This large scale of momentum fluxes represents effects of sub-ice shelf plume and Ekman layer.
* * *
Figure S2: These differences in the Ro radius don't appear to be large enough to explain differences in flow heterogeneity.

- Main parameter for heterogeneity of freezing is the PISW layer and inner overturning cell near

the ice front. We have amended this phrase for Figure S2 to clarify this.

310: this makes it sound like buoyancy and momentum are directly related rather than indirectly related.

- This phrase was not clear. We have amended this to clarity what we meant.

312: It's unclear to me how you determine the PISW top and why it wouldn't be coincident with the ice base.

313: I was expecting the 5% heat flux to determine the IOBL bottom.

- In the previous manuscript, the notation for IOBL top, PISW top was confusing. It is IOBL bottom and PISW bottom, not the top. We have amended these terms.

325: I would have expected you to bring your study into this paragraph. You have a HSSW-dominated regime and yet only ice-shelf melting and not freezing.

325-344: I think you spend too much time reiterating topics already laid out in the introduction.

- In the revised manuscript, we have focused on the interpretation of our results and its discussion with various limitations.

345: oceanic region >> open ocean?

- As the reviewer mentioned, we have amended this phrase.

346: the way this is worded it makes it sound like the far-field values are derived from the interfacial values rather than vice versa

- As the reviewer mentioned, this phrase was unclear. We have amended this phrase.

348: what do you mean by trend here?

- We have amended this phrase with specific physics.

353: "change in plume characteristics" change with what? Maybe just that inclusion of frazil dynamics changes plume characteristics, which requires rephrasing this sentence.

- We have added the paragraph for the relationship between the inclusion of frazil ice dynamic and plume change.

358: I would think that heat and salt exchange would be less vigorous in the stably stratified IOBL than in a convecting ML. I imagine that some readers would be thinking the same and it would be helpful to address why this is the case.

- We have removed this phrase.

360: I don't understand how you get circulation cells over the same depths that the velocity is set to 0. I don't remember this from the Methods either. Is this text correct? Is this only an initial condition, because here it sounds like it's fixed for all time?

- Our description for top boundary was incorrect. We set zero velocity for initial condition and impose Neumann boundary condition (gradient is zero) for the momentum at top boundary. We have amended this explanation in methodology part.

364: Again, I don't see evidence for high speed currents

- We change this to "relatively high speed current".

367: "Assumed" >> "calculated"

- As the reviewer mentioned, we have amended this phrase.

368: "This denotes that…" means that the location of the salt maximum would indicate driving forces and I don't think you intend to say that.

- We have removed this phrase.

370: The wedge mechanism needs more introduction.

- We have added the introduction for freshwater wedge and additional discussion.

371: I don't think that the existence of an Ekman layer needs to be brought up in the discussion, as it is expected.

- We agree with reviewer's comment. We have removed the part for Ekman layer.

376: It's unclear whether the baroclinic eddies mentioned here are the same or different (i.e., in scale) from the baroclinic eddies at the Rossby radius.

- Because the comparison of baroclinic eddy is difficult, we have removed the part for this.

378: This paragraph needs to be split. You go from talking about Ri_f to a list of model limitations.

- We have amended whole discussion section with specific topics. We have moved Ri_f part in model limitations.

379: I don't know what you mean by "turbulence" here (what features?) or where in the column you're computing the Ri_f and stratification.

- In this part, we meant the relationship between stratification and turbulent mixing. We have amended this phrase to clarify this.

381: It's unclear what you mean by negative feedback here, though I assume you're talking about the degree of stratification generated by enhanced buoyancy fluxes. It should be clarified for the reader.

- As the reviewer mentioned, we have amended this phrase to clarify what we meant.

384: "This limitation can be solved using a dynamic SGS model" implies that you didn't use a dynamic SGS model. This sentence should be edited.

- We have removed this phrase and added the phrase for investigation of model constants for near wall physics.

385: What is the "claimed mechanism"?

- We have removed this term.

385: When you say "fill the gaps" I'm picturing a physics-informed interpolation or state estimate whereas here it's more of an extrapolation from one point at the ice-shelf front into the cavity. Please rephrase.

- As the reviewer mentioned, we have amended this phrase to clarify what we meant.
* * *
387: I appreciate the clarification in the parenthetical. However, it is still awkward to say "and their feedback."

- As the reviewer mentioned, we have amended this phrase to clarify what we meant.
* * *
397: inlet >> inflow boundary condition

- As the reviewer mentioned, we have amended this phrase.
* * *
399: "fluctuation" is ambiguous because it could be turbulent fluctuations or fluctuations in the mean flow.

- As the reviewer mentioned, we have amended this phrase (fluctuation -> variance).
* * *
403: "agree well" I think this sentence should acknowledge the primary misfit(s) with observations.

- In paragraph for Figure 5, we have discussed primary misfits between observation and LES results.
* * *
412: This sentence is unclear to me. The 4 turbulence cases should have different levels of shear yet similar stratification so how does this indicate that "the stratified forcing by PISW varies according to the flow shear caused by turbulence"?

- We have removed this phrase.
* * *
413: What exactly should be investigated further? Controls on stratification of the IOBL?

- We have removed this phrase.
* * *
**Anonymous Referee #2**

General comments:

This paper reports on large-eddy simulations of an ice shelf-edge region, inspired by observations made in front of Nansen Ice Shelf, Antarctica. The simulations are run with an idealised geometry

and boundary/initial conditions for temperature, salinity and velocity that are inspired by observations. The velocity boundary condition beneath the ice shelf is varied to simulate four different regimes with varying levels of turbulence. The velocity profile between the ice shelf and the continental shelf below is modelled using a power-law velocity profile, where the power-law relationship is varied to simulate varying degrees of turbulence. A three-equation model is used to model the basal melting of the ice shelf, where changes in ice shelf geometry, volume flux input and lateral ice shelf melt are excluded. In the open- ocean portion of the simulation domain, a rigid sea-ice lid is assumed, where the freezing rates are calculated using a three-equation model. Constant heat and salt exchange coefficients are used for the sea-ice region and varying coefficients, calculated using Monin-Obukhov similarity theory, are used in the ice shelf region (this needs to be clarified, as I may have got this wrong?). The high turbulence simulations show an increased basal melt rate and a stronger Ekman layer, resulting in a modified circulation pattern in the open-ocean region.

- All your summary is correct, except a rigid sea-ice lid. Boundary condition and explanation in previous manuscript was not clear. We have amended this part to clarify this.

I found these simulations to be interesting and I think this paper is of interest to the community. Some changes and clarifications need to be made however before I would be comfortable with publication. Specifically, I think further reference to the existing literature needs to be made. I think the ice front region is a particularly interesting region, not least because it is relatively easy to observe compared with the grounding line. Your simulations are relatively idealised, but I think you should still be able to link your work to previous work in the ice shelf front region, even if the conclusion is that elements of your simulations make comparison difficult. Most importantly, the reader needs to be left with an understanding of what the implications of your simulations are for studies of the ice shelf region. In Garabato et al. (2017), they make the point that the intrusion depth of the ice shelf plume has important ramifications for the effect of ice shelf melt on the Southern Ocean (specifically in simulations). I think motivating your study with this scientific question (or some other broad question) would help clarify the point of your work to the reader.

I have a series of other clarifications and edits to the text that I would like to see which are listed below:

1. It needs to be clear that you are studying an ice front throughout the abstract and the introduction. The geometry of your situation is key to the physics of interest, so must avoid trying to say too much about generic IOBL plumes. i.e. I would revise the sentence in the abstract:

"In this study, we utilize a large-eddy simulation to investigate the role of the turbulence within the IOBL flow with sub-ice shelf plume"

To something involving explicitly focused on ocean dynamics at the edge of an ice shelf.

- As the reviewer mentioned, we have focused on ocean dynamics and its structure near the ice front. We have amended this phrase to clarify this.

Modified phrase in the revised manuscript:

In this study, we utilize a large-eddy simulation (LES) model to investigate the role of turbulence within the IOBL flow with a sub-ice shelf plume and a coherent structure of the ocean dynamics near the ice front.
* * *
2. This line in the abstract:

"This demonstrates that the larger baroclinic eddies enforces heterogeneous distribution of positively buoyant meltwater upwelling"

Is confusing to me. How does a larger melt rate demonstrate that there is a heterogeneous distribution of meltwater upwelling? Surely the melt rate could be homogeneous but larger? This point about heterogenous/homogenous response needs to be explained more fully in the text (or excluded, as I am not totally sure what it adds to the conclusions of the paper).

- As the reviewer mentioned, this phrase was not clear. We have removed this phrase in abstract and conclusions.
* * *
3. Lines 32-35 "Shear force generated by tidal mixing and the thermohaline process during sea ice formation are the basal melting driving forces in cold water cavity (e.g., high salinity shelf water), whereas the intrusion of circumpolar deep water (CDW), which is the water well above the local freezing temperature, is the main driving force for basal melting in the warm-water cavity (Davis and Nicholls, 2019; Jacobs et al., 1992; Yoon et al., 2020)."

Needs to be split up into two sentences maybe. One saying shear forces drive turbulent mixing of T and S through the IOBL, and another saying that shear is generated by tidal mixing or by circulation, where the source of temperature and salinity mixed up to the ice base is HSSW or CDW

- In the revised manuscript, previous phrases have been separated to two phrases for warm water cavity, cold water cavity, and its driving forces.

Modified phrases in the revised manuscript:

In the cold-water cavity, shear forces generated by the tides and brine rejection during the sea ice formation (e.g., high salinity shelf water (HSSW)) are the driving forces that cause basal melting (Davis and Nicholls, 2019; Yoon et al., 2020). In contrast, the intrusion of circumpolar deep water and melt-driven circulation near the grounding line mainly cause basal melting in the warm-water cavities (Holland et al., 2020; Jacobs et al., 1992).
* * *
4. Line 67 : This would be a great opportunity to introduce the novelty of your geometry e.g.

However, applications of the LES to IOBL at sub-ice shelf environment are quite limited. The geometry and scales of ice-ocean interaction may be qualitatively different for an ice shelf, particularly when considering the ice front. Ice shelves typically have a thickness of 100s of metres compared with the $O(1\ m)$ scales of sea ice.

- In the revised manuscript, we have added the additional explanation for our geometry scale in this study.

Modified phrases:

However, studies on the application of LES to the IOBL under a sub-ice shelf environment are limited (Dinniman et al., 2016). This is because the geometry and scales of this ice–ocean interactions are qualitatively different for an ice shelf, particularly at the ice front.
* * *
5.  Line 68 : "In this study, we performed LES experiments for the IOBL and oceanic flow including freezing effect at sea surface and the basal melting process with neutrally buoyant sub-ice shelf plume near ice front."

Split into two sentences, first saying that you are studying the ice shelf plume at the ice front, then saying the effects you are including within your study.

- As the reviewer mentioned, we have amended this phrase. First phrase is for sub-ice shelf plume and second phrase is for parameterized melting and frazil ice formation.
* * *
6. Introduction: Paragraph on what would we expect of this flow? Cite observations of this near ice shelf region and the ideas that people use (i.e. Garabato instability work, ice front blocking work from Wahlin et al, 2020?)

- We have amended last paragraph in introduction to clarify what we expect in this study. For implication of this study, we have added phrases for the discussion with other observation works.
* * *
7.  Line 86: "To simulate the oceanic flow with refreezing"

maybe expand on this, melting is included as well as refreezing so that's worth noting

- As the reviewer mentioned, we have amended this phrase.
* * *
8.  Sa is confusing notationally, it might be better just as S

- As the reviewer mentioned, we have amended this notation.
* * *
9. Line 116: you've introduced the governing equations and the turbulent closure which is great. I think this paragraph could do with a sentence that summarises the key positives and drawbacks of your chosen sub-grid scale model. For instance, I don't imagine it works well for regions where the flow becomes laminar?

- As the reviewer mentioned, we have added the phrases for the important positives and drawbacks of our SGS model.

10. Elaboration is needed on the melt/freeze condition that you apply. My understanding is that you apply a three equation model with constant coefficients in the open- ocean/freezing region, and a three equation model with exchange coefficients calculated using Monin-Obukhov theory for the ice shelf region. Is this correct? If so you need to state this explicitly

- For the ice shelf region, we also used constant coefficients obtained from exchange coefficients of Vreugdenhil & Talyor (2019), not the calculated exchange coefficients (previous Eq. 16 have to be removed). We have added the phrases for exchange coefficients.

11. You include citations for your three-equation model values in the table, but I think mentioning your sources in the text would be beneficial for the reader

- As the reviewer mentioned, we have added the phrases for transfer coefficients.

12. Line 135: "where u* is the friction velocity which is calculated by the velocity at first node and roughness length", are you saying that you infer the friction velocity using a drag coefficient, using the relation $U^2 = C\_d u*^2$ ? If so, you should state this explicitly including your value for C_d and your source for that number. The true drag coefficient is defined using the vertical shear at the boundary; however, I don't think you would resolve those scales in your simulations, so I presume you are using a drag coefficient.

- In this study, we calculate friction velocity using logarithmic law of the wall [$u_* = 1/k \times u_1 \times \ln(z_1/z_0)$] with roughness length ($z_0 = 0.005$ m). We have amended this phrase to clarify this.

13. Line 157 : "Initial profiles were set in the variation range of vertical profiles of our 24 CTD and 23 LADCP observations conducted near the ice front of the NIS." I don't understand this sentence. Are you saying that the initial profiles are taken as a mean of the vertical profiles or a smoothed version? Can you detail the exact method for choosing your idealised profiles?

- As the reviewer mentioned, our explanation for initial profile of potential temperature and salinity was insufficient. We have elaborated information of initial profiles.

14. Line 158-159: "The outlet boundary condition was determined to match the radiation boundary condition (extrapolation)", could you elaborate further on this boundary condition? I do not understand which radiation you are referring to and what the form of the boundary condition is? This is important for determining the utility of your inferred circulation.

- In the previous manuscript, the explanation for radiation boundary condition was not insufficient. We have added the phrase for additional explanation of outlet boundary condition.

15. Line 160-162: Your Dirichlet boundary condition for velocity implies that you have a rigid lid. Your satellite imagery shows a region of ice-free ocean at the edge of the ice shelf. You have assumed essentially that it is ice-covered and the ice does not move (similar to land-fast sea ice). I do not think this is necessarily a problem, but it should be pointed out when you introduce your boundary condition. Are you simulating a winter-version of this ice front region?

- In the previous manuscript, our explanation for momentum boundary condition at top boundary (sea surface) was incorrect. We imposed Neumann boundary condition for momentum at top boundary, considering open ocean in polynya. We have amended this in methodology section.

Modified phrase:

The cyclic boundary condition was applied to lateral boundaries, whereas the Neumann boundary condition for momentum was imposed on the top layer.

16. Line 221: "Below 400 m depth, a well-stratification features appear in salinity distribution." I do not know what this sentence refers to, please clarify.

- To clarify what we meant, we have added the isopycnal lines and the phrase of explanation for this.

17. Line 246 : "it is necessary to confirm that the turbulence characteristics of the LES result are similar to the turbulence characteristics of inertial subrange in which energy cascading occurred with few dissipations", I understand most of this sentence but I do not understand the last three words. Are you just saying it is necessary to confirm that the resolved turbulence in your model follows an inertial scaling?

- As the reviewer mentioned, we have amended this phrase to clarify what we meant (without last three words).

18. You should include details of how you calculated the 1D energy spectra, perhaps including the

equation for your calculation. Specifically, I'm wondering whether it's calculated along a single line in the y-direction for instance? I'm slightly confused as to why you have so many fewer points at the high wavenumber end of your plot than at the low wavenumber end. Usually, power spectra show the opposite trend, as you have more points to evaluate small wavelengths with, you get a higher density of points at the high wavenumber end. If you tried calculating your spectra using a numerical method such as the Welch method, you might also get a smoother result, as currently it's difficult to determine whether the presented spectra do indeed show a - 5/3 slope or not. Also, I don't think it's a problem if they do not show a -5/3 spectrum, as you are simulating an anisotropic flow, so you might not expect a classic inertial subrange. Your SGS model may assume homogeneous isotropic turbulence, but your resolved turbulence does not need to.

- As the reviewer mentioned, energy spectra at the high wavenumber have many points than that at the low wavenumber. In the previous Figure for energy spectra, we used different wavenumber range with incorrect interval. However, spectra trend is similar to the previous plot. In the revised manuscript, we have added the explanation for calculating energy spectra and re-plotted the energy spectra distribution with correct wavenumber range.

New Figure 6. One-dimensional turbulence energy spectra:

[Figure]

19. Line 317 : "Negative heat flux at 320–400 m depths denotes that some of the entrained heat by the intrusion of the outer ocean is transferred to the downward direction." I don't understand this argument. In your temperature and salinity plots, the profiles seem to be increasing with depth in the sub-ice shelf region, so why would the heat flux be negative? I think this needs further explanation

- Because previous sentence was not clear, we have removed this phrase. We have added the phrases for heat entrainment and cooling effect of PISW with convective scale.

20. Your first discussion section seems more like introduction than discussion to me. Your results

aren't discussed, rather the broad field and approach is discussed.

- In the revised manuscript, we have focused on our results and its discussion with various limitations.

---

## Author Response (AR4)

Response to comments

We wish to thank anonymous reviewer for their valuable comments, which will help us to improve our manuscript. We addressed each of the comments in turn below. Our responses are colored by green.
* * *
**Anonymous Referee #1**

- General comments

The authors have made some significant improvements on the previous iteration of this article, and I think it could be close to being of publishable quality. My main reservations are still the ways in which this study links to other studies of the ice shelf edge specifically. The paragraph discussing the work of Garabato et al. (2017) and Malyarenko et al. (2018) is good, but these two works should be discussed in detail in your introduction as they are very relevant to the situation you are describing, and I think your work links with those studies very well. You mention that your work agrees with the picture presented by Garabato et al. (2017), whereby centrifugal instability causes turbulent mixing which changes the settling depth of the ice shelf plume. I am not totally convinced that you are seeing this mechanism, so I suggest that you assess the potential instability from your simulations (i.e. look at the vorticity and compare it to the criteria for centrifugal instability). You could also check the Potential Vorticity to see if you are likely to see any symmetric instability which was ruled out by Garabato et al. (2017). You could also show a plot of the spatial variability of the dissipation rate, or the TKE, to demonstrate whether enhanced turbulence is associated with the plume reaching the ice shelf edge. I think you should expand on how these simulations would change if you included winds, with reference to Malyarenko et al. (2018) who discuss this scenario.

- As the reviewer mentioned, we have discussed observational studies on the frontal region in the introduction of the revised manuscript.

- Moreover, we have added the analysis for relative vorticity and the criteria for potential instability in the supplementary material (Figure S6, S7). We can observe the symmetric instability as well as centrifugal instability.

- We have expanded the discussion for the situation with wind effects (for the katabatic wind case).

**Modified or added parts:**

**Introduction:**

Observational efforts of meltwater behavior and ocean circulation near the frontal region of the ice shelf demonstrate various mechanisms at different locations around Antarctica. In the frontal region of the Pine island ice shelf, Garabato et al. (2017) revealed that the ascent of the meltwater outflow causes vigorous lateral export, affecting the settling of meltwater at depth. The intrusion of relatively warm surface waters and high basal melting near the ice shelf front was observed in the Ross ice shelf (Hogan et al., 2011). Moreover, Malyarenko et al. (2018) suggested

the existence of a "wedge" of fresher water in the Western Ross sea and it is formed from meltwater near the ice shelf front.

**Supplementary material:**

[Figure]

Figure S6. xz contours of z-direction, relative vorticity in the cases with weak turbulence (n=3) and strong turbulence (n=7). Positive values represent the region for the symmetric instability (vertical shear), whereas negative values represent the region for the centrifugal instability (lateral shear) in well-stratified fluid ($N^2 > 0$). Gravitational instability occur at the region of concentrated salt flux by sea ice formation ($N^2 < 0$). PISW upwelling right after the ice front can be classified to gravitational instability (inset profile of Figure S7a).

[Figure]

**Figure S7. (a) Vertical profiles of angle of balanced Richardson number to identify the type of possible instability in the whole ocean region. Inset figure in (a) is for that in the frontal region from the ice front to 24 m. Vertical profiles of (b) buoyancy and (c) flow shear terms of Richardson number in the cases with weak turbulence (n=3) and strong turbulence (n=7).**

**Discussion:**

Differ to centrifugal instability in the Pine island ice shelf, we can demonstrate the symmetric instability in this study (Figure S6 and S7). This difference is caused by the different directions of the current near the sea surface and the exclusion of the katabatic wind effect.

In Terra Nova Bay, the northeastward katabatic wind is dominant and it drives the along-front current (Guest, 2021; Malyarenko et al., 2019). If we included the wind effect at the sea surface, the horizontal mixing may be enhanced, advecting the fresh meltwater of the PISW layer near the ice front to the open sea. In terms of overturning cells, the strength and horizontal scale of the outer overturning cell may decrease, as the wind stress reduces the shear stress between the sea surface and the sub-ice shelf plume.
* * *
I have a series of smaller points which I also think need to be addressed:

4. "Coherent structure of the ocean dynamics near the ice front" needs rephrasing, I think this is not the correct use of the phrase coherent structure

- We have amended this phrase without coherent structure part (line 11-13).

5. Change to "to simulate the varying turbulent intensities"

- We have amended this phrase as the reviewer mentioned (line 13).
* * *
23. "Frazil ice" is misused throughout the manuscript. Frazil ice refers to ice that forms due to the supercooling of water, where the water pressure has increased and so the freezing temperature has increased, so ice begins to form within water that is below its own freezing point. I think you mean to refer to sea-ice formation, or simply generic freezing.

- We have changed "frazil ice" to "sea-ice formation" throughout the manuscript.
* * *
52. "well-mixed feature (20–30 m) of the temperature and salinity induced by a strong tidal forcing and a weak stratification structure was observed" needs rephrasing e.g. "a well-mixed boundary layer was observed in both temperature and salinity, induced by a strong tidal forcing and a weak stratification. A moderate melt rate was observed, despite the low thermal driving, due to the observed shear-driven turbulence."

- We have amended this phrase to clarify what we meant.

**Modified phrases (line 51-53):**

In the Larsen C ice shelf which is a cold-water cavity, a well-mixed boundary layer (20–30 m) was observed in both temperature and salinity, induced by a strong tidal forcing and a weak stratification. A moderate melt rate was observed, despite the low thermal driving due to the observed shear-driven turbulence (Davis and Nicholls, 2019).
* * *
79. You have not used Monin-Obukhov similarity theory in your set up – just the standard three equation model. Monin-Obukhov theory would give a correction to the dependence of heat and salt flux on the difference between far-field temperatures/salinities and those at the ice base, as discussed in Vreugdenhil et al (2019). You have only included a linear dependence for your heat and salt fluxes, so I would not term that Monin-Obukhov similarity theory.

- We have amended this term to surface fluxes.

**Modified phrase (line 84-85):**

To include thermohaline effect by the sea-ice formation at sea surface and basal melting at the ice-shelf base, surface fluxes in both temperature and salinity were used.

166. 1/Gamma_T, 1/Gamma_S, usually these are defined as Gamma_T and Gamma_S, why have you used the inverse?

- As the reviewer mentioned, these nomenclatures were confusing. We have amended these terms without the inverse.

**Modified phrases (line 172-173):**

$\Gamma_\theta$ and $\Gamma_S$ are the non-dimensional transfer coefficients of heat and salt, respectively, determined from the near-wall physics.
* * *
270. Your discussion of the 'baroclinic eddy' needs more clarity. I think you could remove the whole paragraph starting from "this PISW layer blocks an intrusion of the outer ocean" to line 274.

- As the reviewer mentioned, the previous discussion of the baroclinic eddy was unclear. We have removed this part and rephrased this.

274. A comparison of the scale of your spatial patterns to the Rossby radius of deformation is useful, but you should first define the Rossby radius of deformation and make it clear how you have calculated it.

- We have added the definition of Rossby radius of deformation and how we have calculate this.

**Modified phrases (line 275-280):**

However, heterogeneous patterns of the freezing rate are observed in the strong turbulence case, because the PISW layer near the ice front is wide with a weakened inner overturning cell, permitting the larger baroclinic disturbance caused by sloped isopycnals. This heterogeneous pattern of the freezing rate is comparable to the disturbance scale (2,066 m), as identified from the Rossby radius of deformation which represents the length scale the rotation effect is dominant. This scale is obtained by depth-averaged buoyancy frequency and depth between the sea surface and IOBL bottom.
* * *
Figure 6. I think you should plot lines rather than a scatter, to better show the power spectra. You should also list in the caption how exactly these spectra were calculated i.e. was it a single point velocity measurement through time? Or was it a spatial assessment of the entire velocity field? The spectra are very noisy, so it's difficult to see the slope – if you average spectra across multiple times, or some spatial average, then you will get smoother spectra which may be more useful.

- These one-dimensional spectra were obtained by a spatial assessment (meridional direction) of the time-averaged velocity field. Through more spatial averaging, we can get smoother spectra that can indicate its slope. We have added the explanation for the calculation process in the caption of the revised manuscript.

Figure 6. Your second plot seems to show shallower slopes than k^(-5/3)? Is this a k^(-1) slope? Please include the slope on this plot. If it is k^(-1) then that could be indicative of a Batchelor style regime rather than an inertial subrange, which would be worth commenting on.

- As the reviewer mentioned, power spectra in the high wavenumber regime are close to the $k^{-1}$ slope, representing the Batchelor regime rather than an inertial subrange. We have added the $k^{-1}$ slope in new Figure 6.

**Modified Figure 6:**

[Figure]

**Figure 6: One-dimensional turbulence energy spectra at a depth of 291 m at the PISW within the IOBL. (a) n = 3 and (b) n = 7. Different shapes and colors represent the values at different zonal distances: 400, 800, 1200, 1600, 2000 and 2400 m. These power spectra are obtained by y-direction (meridional direction) spatial assessment of time-averaged velocity at each x location. The –5/3 slope (Kolmogorov scaling) represents the regime of inertial subrange, whereas –1 slope (Batchelor) represents viscous-convective range in high-Schmidt number.**

344. I think the way you are calculating thermal driving is different from the method given in Wray (2019). You have temperatures and salinities taken from ocean observations, so that should not give a significantly different thermal driving. I imagine that the main differences between your simulations and the observations in terms of melt rate is associated with the lack of winds in your simulations, but there are so many idealisations in your simulations that could be the cause of this disparity.

- In the study of Wray (2019), he used the ocean temperature of -1.86 ℃ and freezing temperature at 140 m depth to obtain thermal driving (0.14 ℃). Whereas, we used the ocean temperature of - 2.06 ℃ (sub-ice shelf plume) and freezing temperature at -2.116 ℃ at 280 m depth to obtain thermal driving (0.056 ℃). We have demonstrated this difference in ocean temperature was due to intrusion and interaction of surface water due to wind and tidal effects (line 348-350).

**Modified phrases:**

However, the melt rate in our LES results is comparable to the observed melt rate, considering the difference in the thermal driving (0.056 °C (2.116 °C – 2.06 °C) in our LES simulations and

0.14 °C (2.0 °C – 1.86 °C) in the study of Wray (2019)). In this study, only the effect of sub-ice shelf plume (formed by HSSW) was considered, but the observations included the effects of relatively warm Antarctic surface water and sub-ice shelf plume, resulting in a difference in the thermal driving and melt rates. If the melt rate in this study is assumed to be 0.12 m yr$^{-1}$ (averaged value of 0.092 and 0.153), we can estimate that 12–25 % of the total basal melting near the NIS front is due to sub-ice shelf plumes. The rest portion is related to surface water intrusion with wind stress and tide effects.

---

## Author Response (AR5)

Response to comments

We wish to thank editor for their valuable comments, which will help us to improve readability of our manuscript. We addressed each of the comments in turn below. Our responses are colored by green.
* * *
**Editor comments**

- General comments

Thank you for your submission of a revised version of the manuscript on Large-eddy simulations of the IOBL near the ice front of Nansen Ice Shelf, Antarctica. You have addressed the reviewer's comments by supplying additional details in the main text, as well as new figures in the supplementary material. Thank you for these new contributions.

Based on my own reading of the latest version of the manuscript, I believe significant additional improvements are still required to elevate your manuscript to publishable quality. I have attached a file with a list of further comments and suggestions for improvement, in particular to help improve the readability of the manuscript. At present, significant portions of text, including the abstract and introduction, lack coherence and often use poorly defined terminology, which make it difficult to follow the narrative or glean the key messages. Conclusions are not always firmly supported by the results. I think these shortcomings can be addressed without the need for additional experiments or figures, and I therefore welcome a revised version of your work, which takes these suggestions and comments into account.

- As the editor mentioned, there were shortcomings in previous manuscript. Based on comments and suggestions, we have rearranged the structure of manuscript and rephrased the shortcoming parts to avoid confusion.

- There were many comments from marked pdf file. For the crucial comments, we addressed each of the comments with our responses.
* * *
- Abstract

12-13 It is unclear how you define 'sub-ice shelf plume', and its relation to the IOBL. To avoid confusion, consider omitting this sentence here.

- We have omitted the part for "sub-ice shelf plume" to avoid confusion.

16-18 This sentence seems out of place and I'm not sure these details should be included in the abstract.

- We have removed these details of this sentence.

26-28 Readers might find this sentence confusing, because you have not defined what you mean

by ice shelf plume. So far I assumed you would be simulating the ice shelf plume in your model. If this is not the case, how do you 'parameterize' the plume?

- As the editor mentioned, we simulated the ice shelf plume in our model. This sentence can be confusing for readers. We have removed this sentence.
* * *
- Introduction

34-37 I think this paragraph is better placed after line 45. First you introduce the ice shelf-ocean system, explain that this is a complex environment, but that observations could help; then you point out that observations are difficult to obtain; then that ocean models can help improve our understanding.

- As the editor mentioned, we moved and merged this paragraph to observational effort part.

58 The concept of a sub ice-shelf plume is used heavily throughout the manuscript, but has not been defined. In particular, how does it relate to the IOBL? Here it sounds as if they are 2 separate entities, but isn't the IOBL a part of the plume? Please make it abundantly clear what you mean by 'sub ice-shelf plume' and how it distinguishes from the IOBL.

- We have added the explanation for the concept the IOBL and sub-ice shelf plume.

62 You need to make abundantly clear early on (maybe even in the abstract), that you focus on the dynamics in the vicinity (within kilometers) of the ice front, and that you do not include the grounding line in your simulations. Otherwise, readers will assume/expect you are doing cavity-wide simulations, and won't know you are only looking at ice-front dynamics until you describe the experiments in the next section.

- We have added the paragraph for the frontal region of the ice shelf.

72 I think before this paragraph you need to give a short description of what LES are, and how they have been used to study ice-ocean boundary processes so far.

- We have added the paragraph for LES description and its relationship with IOBL processes.

77 I think lines 77-83 need to come earlier, where you motivate your focus on ice front dynamics.

- We have moved this part and added paragraph for the frontal region of the ice shelf.

83 So far it was unclear to me what the overall objectives of your work are. It would be nice to see some of the next paragraph (L83-85, L90-95) earlier in the introduction, potentially even after line 45.

- We have rearranged the introduction to clarify the importance and objectives of this study.
* * *
- Methodology

153 Are you not trying to model the IOBL, and if so, would ambient conditions not need to be set away from this layer, rather than 'within' the IOBL?

- As the editor mentioned, "within the IOBL" was not clear for describing ambient conditions. We rephrased this part.

155 in situ: please refer to Sect 2.1, Fig 1 and strongly reiterate that measurements are taken in front of the ice shelf, not below the ice shelf.

- In revised manuscript, we have reiterated that measurements are taken in front of the ice shelf.

185 A general comment about lines 185-190: It might be better to move these to the beginning of the section, so you first describe your domain, carefully name/identify the different (types of) boundary, and then describe the conditions you impose at each boundary.

197 again I think the description of the ice shelf geometry needs to come earlier in your description of the domain geometry and before you introduce the various boundary conditions.

In revised manuscript, we have rearranged the methodology section as the editor mentioned.
* * *
- Results

222 I think this can be removed here, and discussed in section 4, when the reader has a better understanding of experiments and results in Fig9 have been presented.

- We moved this part for IOBL depth after Fig 9.

229 It seems some periodicity also exists before 14t*. Can you comment?

- Your comment is correct. We have amended this phrase to clarify that averaged friction velocity had a temporal convergence after 14 $t^*$.

242 I don't understand how this follows from the preceding discussion. Can you please elaborate more?

- Main driving forces for inner cell were buoyancy forces by PISW upwelling and downwelling of local salt maximum. That is way inner cell is stretched in the vertical direction. We have amended this phrase to clarify this.

249 Melt rates are <0.3m/yr. Is this really 'vigorous'? Please avoid subjective statements.

- We removed this statement.

256 By 'observed' you mean 'simulated'? But isn't the zonal velocity imposed as a boundary condition? How is this a diagnostic model result?

- Although we imposed the boundary condition for zonal velocity, the flow gradually develops to IOBL flow as it moves away from the inflow boundary. So, we used the expression of "observe".

306 'Similar to' needs more nuance. You need to point here that you imposed observation-constrained boundary conditions (refer to the dashed lines in Fig5). Any deviation of model data from the dashed line is due to model physics, which tend to the observations for T, but move further away from observations for S. So there is a nuanced picture here, where you are able to match some observations, but unable to explain other features, such as S and crucially, the core of cold water at -100m.

314 General comment about Section 3.2: It seems like a best match between modeled and observed temperature is found for low turbulence numbers n, whereas a best match for salinity is found for high turbulence numbers. How do you reconcile these results, and does it mean there is no optimal value for n that best describes both observed profiles of T and S? If so, 1) can you really conclude that LES results are 'similar' to in situ observations, and 2) what is the role of n in better simulating IOBL processes if no value of n provides an optimal match with observations?

- These LES results resolve *in situ* oceanic circulation and thermohaline properties, but LES results had a difference in magnitude quantitatively. This difference was caused by initial condition, boundary conditions, model physics, and thermohaline effects of melting and sea-ice formation. To correct this, additional sensitivity studies for each condition and *in situ* observation for sub-ice shelf environment will be needed.

- The reason we used turbulence number n is that we cannot obtain *in situ* profile of sub-ice shelf, not the optimal match to observations 1 km away from ice shelf front.

- In this study, we cannot diagnose what parameters cause this difference between LES results and observation. But we can resolve the oceanic environments such as overturning cells and PISW upwelling. Based on this, we have amended "similar to" to "are consistent with".

L315-329 Consider adding this as a separate section, or move to the discussion section.

- Because this paragraph is to examine resolving turbulence and its spectra of LES results, this is about the validation of turbulence resolving. So, we have not moved this paragraph.

354 This is VERY speculative, as you are mixing (incomplete) simulations with (uncertain) observations. Your simulations don't really allow you make this conclusion unless you include effects such as a non-trivial ice geometry, non-trivial bathymetry, full development of the boundary layer and sub-shelf plume, 3D ocean circulation in the cavity etc. I would argue that a full cavity model is needed in order to be able to partition the melt resulting from frontal dynamics, and melt occurring beneath the remainder of the ice shelf.

- We agree with editor's comment. We have removed this phrase to avoid confusing.

369-375 I'm unsure why you introduce a metric for the IOBL bottom here, and how it contributes to the discussion. Can you remove L369-375, or clarify why this is important?

- IOBL bottom and the depth of maximum momentum fluxes are highly correlated to heat entrainment by turbulence. So, we have to determine the IOBL bottom here. We have amended this part to clarify why it is important.

- Discussions & Conclusions

386 I think this belongs in the introduction.

- We have moved this part to introduction section.

435 L435 Do you have any specific cavity observations in mind? Why not simulate a full cavity geometry (including the grounding line) and only set boundary conditions at the open ocean end, for which more observations exist?

- Because this LES model needs to fine grid scale to resolve the SGS turbulence, it is hard to simulate a full cavity geometry. Vertical profile of how-water drilling can be helpful for composing the boundary conditions of LES.

447 what is the 'lateral side' of the ice shelf?

- It is vertical side of the ice shelf front. We have amended this.

488 I think this is an overstatement, since you don't present observations of T, S, LDCP data for the inner overturning cell. Perhaps say 'are consistent with'

- As the editor mentioned, it is overstatement. We have amended this part as editor's comments.